# Decentralized primal-dual actor-critic with entropy regularization for safe multi-agent reinforcement learning

## Abstract

We investigate the decentralized safe multi-agent reinforcement learning (MARL) problem based on homogeneous multi-agent systems, where agents aim to maximize the team-average return and the joint policy's entropy, while satisfying safety constraints associated to the cumulative team-average cost. A mathematical model referred to as a homogeneous constrained Markov game is formally characterized, based on which policy sharing provably preserves the optimality of our safe MARL problem. An on-policy decentralized primal-dual actor-critic algorithm is then proposed, where agents utilize both local gradient updates and consensus updates to learn local policies, without the requirement for a centralized trainer. Asymptotic convergence is proven using multi-timescale stochastic approximation theory under standard assumptions. Thereafter, a practical off-policy version of the proposed algorithm is developed based on the deep reinforcement learning training architecture. The effectiveness of our practical algorithm is demonstrated through comparisons with solid baselines on three safety-aware multi-robot coordination tasks in continuous action spaces.

## 1 Introduction

Cooperative multi-agent reinforcement learning (MARL) involves multiple agents operating within a shared environment, which learn to make sequential decisions that optimize a common objective. Over the past few years, numerous efficient cooperative MARL algorithms have been developed following the centralized-training (CT) paradigm (Kuba et al., 2022; Liu et al., 2024). These algorithms address the non-stationarity issue by training centralized critics, which also maintain scalability in coordination through the use of decentralized policies. Moreover, the policy sharing mechanism is widely used in CT-based algorithms for homogeneous agents, since it empirically enhances learning scalability and efficiency (Gupta et al., 2017; Liu et al., 2019) and has been proven to preserve the optimality of cooperative MARL problems (Chen et al., 2022). Nevertheless, in many real-world scenarios, it requires centralized or all-to-all communication to achieve centralized training, which makes CT-based algorithms impractical when communication resources are limited.

Decentralized MARL algorithms aim to solve the cooperative MARL problem under mild communication conditions (Zhang et al., 2019). Under the decentralized training setting, there only exists a possibly time-varying and sparse communication network among agents. Each agent learns its policy based on local experiences and necessary information shared by its neighbors, such as network parameters (Zhang et al., 2018) and local state-action pairs (Qu et al., 2022). Early works in this field mainly focus on designing convergent decentralized algorithms under standard assumptions. Based on the theoretical results, recent works further showcase the potential of applying decentralized algorithms to challenging MARL tasks involving many decision-makers, such as multi-robot coordination (Chen et al., 2022; Hu et al., 2024) and traffic control (Du et al., 2022; Ma et al., 2024). Realizing that agents in the real world can be safety-critical systems, researchers have paid increasing attention to the decentralized safe MARL problem, where the learned policies should meet specific safety constraints. Despite that existing approaches have demonstrated promising performance on safe MARL tasks with discrete action spaces (Lu et al., 2021; Ying et al., 2023b), it still remains a challenge to design efficient decentralized algorithms for continuous safe MARL tasks.

**Contributions.** We develop decentralized safe MARL methods for networked homogeneous multi-agent systems. A decentralized safe MARL problem is formulated, which uses the entropy regularization mechanism to address sample efficiency issues in high-dimensional spaces. The optimality of our problem under the policy sharing setting is studied, based on which decentralized safe MARL algorithms are proposed. The main contributions of this work are summarized as follows:

- A subclass of the constrained Markov game (MG) (Gu et al., 2023) is characterized, which extends the model proposed in Chen et al. (2022) under the safe MARL setting. Based on this model, a decentralized safe MARL problem is formulated employing an entropy regularizer in the objective function design, where policy sharing is proven to preserve optimality and safety.

- An on-policy decentralized primal-dual actor-critic algorithm is proposed, where a novel decentralized dual variable update step is designed to deal with the centralized constraint. Asymptotic convergence of the proposed algorithm is established based on multi-timescale stochastic approximation theory under standard assumptions.

- Compared with existing works on decentralized safe MARL (Lu et al., 2021; Ying et al., 2023b), a practical off-policy decentralized algorithm is further proposed based on the deep reinforcement learning (DRL) training architecture, which can effectively deal with continuous spaces. Simulation results on safety-aware continuous multi-robot coordination tasks demonstrate the effectiveness of our practical decentralized algorithm.

**Related Work.** Decentralized MARL algorithms solve the cooperative MARL problem based on a possibly time-varying and sparse communication network. Based on the use of the communication network, existing decentralized algorithms can be divided into two categories: *communication for parameter information* and *communication for state-action information*. In the former category, decentralized algorithms usually assume the availability of the global state due to the coupled state transition function, where agents locally exchange parameter information over the communication network to estimate global value functions (Zhang et al., 2018; Suttle et al., 2020; Ye et al., 2024). These algorithms obtain theoretical convergence under practical communication conditions. Note that some practical decentralized MARL algorithms have been proposed by combining theoretical decentralized algorithms with DRL (Chen et al., 2022; Hu et al., 2024), achieving comparable performance to solid CT-based MARL baselines. In the latter category, decentralized algorithms rely on some spatial correlation decay assumptions of weakly coupled Markov decision processes (Qu et al., 2022; Ying et al., 2023a). In these algorithms, each agent replaces the global state and action with local state-action pairs received from its $k$-hop neighbors in its critic network during training. Our algorithm is more pertinent to the former, where we assume that the global state and action information is available to each agent. It is worth pointing out that the communication mechanism employed in the latter can be effectively incorporated into our algorithm under the local observation setting (see Appendix J.5 for more details).

Safe MARL has seen a surge of interest in recent years. Cai et al. (2021) combine the well-known CT-based algorithm MADDPG (Lowe et al., 2017) with decentralized control barrier function (CBF) shields to achieve safe exploration, which requires accurate system model of agents when designing CBF conditions. Gu et al. (2023) develop a model-free safe multi-agent policy iteration algorithm based on the multi-agent trust region learning theory (Kuba et al., 2022), which attains monotonic improvement in reward and satisfaction of safety constraints at every iteration. Note that centralized units are required by these algorithms during the training process. Lu et al. (2021) propose the first decentralized safe MARL algorithm for networked multi-agent systems, which employs a primal-dual framework to search for the saddle point associated to reward and cost. However, each agent must maintain and share a copy of the global policy, which may not be preferred in privacy-sensitive applications and could result in scalability issues. Moreover, the learning performance of this algorithm can be severely limited by the vanilla policy gradient method in high-dimensional spaces (Lillicrap et al., 2016). Following the similar decentralized setting as Qu et al. (2022), Ying et al. (2023b) propose a convergent primal-dual actor-critic algorithm for safe MARL, which utilizes shadow rewards to deal with the general utility setting. Nevertheless, this algorithm inevitably faces challenges in estimating local state-action occupancy measures in continuous spaces.

**Notations.** Let $\mathcal{N} = \{1, \ldots, N\}$. Let $y = (y_1, \ldots, y_N)$ be an ordered list and $M = [m_1, \ldots, m_N]$ be a permutation satisfying $m_i \in \mathcal{N}$ and $m_p \neq m_q$ if $p \neq q$, $\forall p, q \in \mathcal{N}$. Then, a permutation of $y$ under $M$ can be represented by $My = (y_{m_1}, \ldots, y_{m_N})$. Let $\mathcal{M}$ be the set containing all possible

$M$. Denote $|\mathcal{P}|$ as the cardinality of a finite set $\mathcal{P}$. Denote $\otimes$ as the Kronecker product. Let $\mathbb{1}$ and $I$ be respectively the all-one vector and the identity matrix with proper dimensions.

## 2 SAFE MARL WITH DECENTRALIZED AGENTS

### 2.1 HOMOGENEOUS CONSTRAINED MARKOV GAME

We consider a constrained MG $\langle N, \mathcal{S}, \mathcal{A}, P, R, C, \gamma \rangle$ which contains $N$ agents indexed by $i \in \mathcal{N}$, where $\mathcal{S}$ and $\mathcal{A} = \prod_{i=1}^{N} \mathcal{A}_i$ are respectively finite state and action spaces, $P : \mathcal{S} \times \mathcal{A} \times \mathcal{S} \to [0, 1]$ is a state transition function, $R = \{R_i\}_{i \in \mathcal{N}}$ and $C = \{C_i\}_{i \in \mathcal{N}}$ are respectively reward and cost functions with $R_i, C_i : \mathcal{S} \times \mathcal{A} \to \mathbb{R}$ for any $i \in \mathcal{N}$, and $\gamma \in [0, 1)$ is a discount factor. Denote $\pi_i : \mathcal{S} \times \mathcal{A}_i \to [0, 1]$ as the local policy of agent $i$. At time step $t$, every agent $i$ executes an action $a_{i,t} \in \mathcal{A}_i$ sampled from $\pi_i$ based on the current state $s_t$. Then, the constrained MG shifts to $s_{t+1}$ at the next time step, and each agent $i$ receives a reward $r_{i,t+1}$ and a cost $c_{i,t+1}$ satisfying $R_i(s_t, a_t) = \mathbb{E}[r_{i,t+1}|s_t, a_t]$ and $C_i(s_t, a_t) = \mathbb{E}[c_{i,t+1}|s_t, a_t], \forall i \in \mathcal{N}$. When agents are homogeneous, we can further characterize a subclass of constrained MGs, referred to as homogeneous constrained MGs, which is defined as follows:

**Definition 1.** A constrained MG $\langle N, \mathcal{S}, \mathcal{A}, P, R, C, \gamma \rangle$ is homogeneous if

  (i) The local action space is homogeneous to all agents, i.e., $\mathcal{A}_i = \mathcal{A}_j, \forall i, j \in \mathcal{N}$. The state space can be decomposed into $N$ homogeneous local state spaces, i.e., $s = (s_1, \ldots, s_N) \in \mathcal{S} = \mathcal{S}_1 \times \cdots \times \mathcal{S}_N$ with $\mathcal{S}_i = \mathcal{S}_j, \forall i, j \in \mathcal{N}$, where $s_i \in \mathcal{S}_i, \forall i \in \mathcal{N}$.

  (ii) Both the joint reward and cost functions are permutation preserving, and the state transition function is permutation invariant, i.e., for any $M \in \mathcal{M}$, $s_t = (s_{1,t}, \ldots, s_{N,t}) \in \mathcal{S}$, and $a_t = (a_{1,t}, \ldots, a_{N,t}) \in \mathcal{A}$, it holds

$$Z(Ms_t, Ma_t) = MZ(s_t, a_t), \quad P(s_{t+1}|s_t, a_t) = P(Ms_{t+1}|Ms_t, Ma_t),$$

  where $Z(s_t, a_t) = (Z_1(s_t, a_t), \ldots, Z_N(s_t, a_t))$ for any $Z \in \{R, C\}$.

  (iii) Each agent $i \in \mathcal{N}$ determines its observation through a bijective function $o_i : \mathcal{S} \to \mathcal{O}$, where the observation space $\mathcal{O}$ is homogeneous to all agents. Furthermore, the observation functions are permutation preserving, i.e., for any $s \in \mathcal{S}$ and $M \in \mathcal{M}$, it holds

$$(o_1(Ms), \ldots, o_N(Ms)) = M(o_1(s), \ldots, o_N(s)).$$

The homogeneous constrained MG given in Definition 1 is a natural extension of the homogeneous MG established in Chen et al. (2022) under the safe MARL setting. An illustrative example of this model is given in Appendix A.1, and some practical examples are given in Appendix A.2.

### 2.2 PROBLEM FORMULATION

We consider a decentralized safe MARL problem over homogeneous constrained MGs. Following the decentralized setting in Zhang et al. (2018); Chen et al. (2022), we assume that neither centralized nor all-to-all communication is applicable in our problem. Instead, there exists a time-varying sparse communication network among agents, characterized by an undirected graph $\mathcal{G}_t = (\mathcal{N}, \mathcal{E}_t)$, where $\mathcal{E}_t \subseteq \{(m, n) : m, n \in \mathcal{N}, m \neq n\}$ denotes the edge set. Agents $m$ and $n$ can share information at time step $t$ if $(m, n) \in \mathcal{E}_t$. In addition, each agent can observe the global state and the joint action, but it only has access to its local policy and reward information. Let $\pi(\cdot|s) = \prod_{i=1}^{N} \pi_i(\cdot|s)$ with $\Pi$ being the set of all possible $\pi$. To incentivize exploration, the entropy regularization mechanism is employed (Geist et al., 2019). Denote $V_\pi^r(s) = \mathbb{E}\left[\sum_{t=0}^{\infty} \gamma^t (\bar{r}_{t+1} + \alpha \mathcal{H}(\pi(\cdot|s_t)))|s_0 = s\right]$ as the value function associated to the reward, where $\bar{r}_{t+1} = \frac{1}{N} \sum_{i \in \mathcal{N}} r_{i,t+1}$, $\alpha > 0$, and $\mathcal{H}(\pi(\cdot|s)) = -\sum_{a \in \mathcal{A}} \pi(a|s) \log(\pi(a|s))$ is the entropy functional. Let $V_\pi^c(s) = \mathbb{E}\left[\sum_{t=0}^{\infty} \gamma^t \bar{c}_{t+1}|s_0 = s\right]$ be the value function associated to the cost, where $\bar{c}_{t+1} = \frac{1}{N} \sum_{i \in \mathcal{N}} c_{i,t+1}$. Denote $\rho$ as the known initial state distribution over $\mathcal{S}$. Given a threshold $b$, the target of the agents is to collaboratively learn an optimal joint policy $\pi^*$ for the following constrained optimization problem:

$$\max_{\pi \in \Pi} \quad J^r(\pi) = \mathbb{E}_{s \sim \rho}\left[V_\pi^r(s)\right], \quad \text{s.t.} \quad J^c(\pi) = \mathbb{E}_{s \sim \rho}[V_\pi^c(s)] \leq b. \tag{1}$$

The method proposed in this paper can generalize to the case of multiple constraints directly, and it is reasonable to assume that all agents have the same number of cost functions due to homogeneity. Compared with existing works on decentralized safe MARL, the constraint considered in this work is centralized, which includes the costs of all agents. This poses a huge challenge for the decentralized safe MARL algorithm design.

# 3 DECENTRALIZED PRIMAL-DUAL ACTOR-CRITIC ALGORITHM DESIGN

The observation function introduced in condition (iii) of Definition 1 inspires us to further design the local policy $\pi_i$ as $\pi_{i,o}(\cdot|o_i(s))$. Denote $\pi_o(\cdot|s) = \prod_{i=1}^{N} \pi_{i,o}(\cdot|o_i(s))$ with $\Pi_o$ being the set of all possible $\pi_o$.

**Theorem 1.** *In homogeneous constrained MGs, suppose $\pi^* \in \Pi$ is the optimal joint policy for the constrained optimization problem (1). Then, there exists an optimal joint policy $\pi_o^* = \prod_{i=1}^{N} \pi_{i,o}^* \in \Pi_o$ with $\pi_{i,o}^* = \pi_{j,o}^*$, $\forall i, j \in \mathcal{N}$, which satisfies $J^r(\pi_o^*) = J^r(\pi^*)$ and $J^c(\pi_o^*) = J^c(\pi^*) \leq b$.*

The proof for Theorem 1 is given in Appendix A.3. Theorem 1 clearly indicates that we can consider the constrained optimization problem (1) on the set $\Pi_o$, and policy sharing of observation-based local policies does not harm the optimality of (1) in homogeneous constrained MGs. This result justifies the use of the policy sharing mechanism in the safe MARL algorithm design for the first time. In the remainder of this paper, we focus on learning observation-based local policies to solve (1).

Let $\pi_{i,\theta_i}$ be the parameterized policy of $\pi_{i,o}$ with $\theta_i \in \Theta_i$. Let $\pi_\theta = \prod_{i=1}^{N} \pi_{i,\theta_i}$ be the parameterized joint policy with $\theta = [\theta_1^T, \ldots, \theta_N^T]^T \in \Theta$, where $\Theta = \prod_{i=1}^{N} \Theta_i$ is compact. Let $\Pi_\Theta$ be the set including all $\pi_\theta$. For simplicity of notation, denote $V_\theta^r$ and $V_\theta^c$ as the value functions of $\pi_\theta$ associated to the reward and the cost, respectively. According to Cayci et al. (2021) and Sutton et al. (1999), the action-value functions of $\pi_\theta$ associated to the reward and the cost take the forms $Q_\theta^r(s,a) = \bar{R}(s,a) + \gamma \mathbb{E}_{s'}[V_\theta^r(s')]$ and $Q_\theta^c(s,a) = \bar{C}(s,a) + \gamma \mathbb{E}_{s'}[V_\theta^c(s')]$, respectively, where $\bar{R}(s,a) = \mathbb{E}[\bar{r}_{t+1}|s_t = s, a_t = a]$ and $\bar{C}(s,a) = \mathbb{E}[\bar{c}_{t+1}|s_t = s, a_t = a]$. Then, the constrained optimization problem (1) can be equivalently represented by

$$\max_{\theta \in \Theta} \quad J^r(\theta) = (1-\gamma)\mathbb{E}_{s\sim\rho}\left[V_\theta^r(s)\right], \quad \text{s.t.} \quad J^c(\theta) = (1-\gamma)\mathbb{E}_{s\sim\rho}[V_\theta^c(s)] \leq (1-\gamma)b. \quad (2)$$

Note that both the objective function and the constraint in (2) are non-convex, making this problem difficult to solve even in a centralized fashion. Hence, we employ the primal-dual method to obtain approximate solutions of (2). Define the Lagrangian associated to (2) as

$$L(\theta, \lambda) = J^r(\theta) - \lambda(J^c(\theta) - (1-\gamma)b), \quad (3)$$

where $\lambda \geq 0$ is the Lagrangian multiplier. Define the dual function as $f_d(\lambda) = \max_{\theta \in \Theta} L(\theta, \lambda)$. Then, the dual problem of (2) takes the form

$$\min_{\lambda \geq 0} f_d(\lambda) = \min_{\lambda \geq 0} \max_{\theta \in \Theta} L(\theta, \lambda). \quad (4)$$

The solution to (4) determines the tightest upper bound for the primal problem (2) (Boyd & Vandenberghe, 2014). Provided that the dual gap is small, the optimal parameter $\theta^*$ obtained from (4) is close to that of the primal problem (2) (Paternain et al., 2023).

Let $\mathbb{P}_\theta(s_t)$ be the marginal state distribution with respect to (w.r.t.) $\pi_\theta$, then the visitation measure of a state $s \in \mathcal{S}$ is represented by $d_\theta(s) = (1-\gamma)\sum_{t=0}^{\infty} \gamma^t \mathbb{P}_\theta(s_t = s)$. Given the Lagrangian (3), we first establish a policy gradient theorem for safe MARL and provide its proof in Appendix B.

**Theorem 2.** *Suppose that $\pi_{i,\theta_i}$ is continuously differentiable w.r.t. $\theta_i$ over $\Theta_i$ for any $i \in \mathcal{N}$, $s \in \mathcal{S}$, and $a_i \in \mathcal{A}_i$. Let $A_\theta^\lambda(s,a) = Q_\theta^r(s,a) - \alpha \log(\pi_\theta(a|s)) - \lambda Q_\theta^c(s,a)$. Then, the gradient of $L(\theta, \lambda)$ w.r.t. $\theta_i$ takes the form*

$$\nabla_{\theta_i} L(\theta, \lambda) = \mathbb{E}_{s\sim d_\theta, a\sim\pi_\theta}[A_\theta^\lambda(s,a)\nabla_{\theta_i} \log(\pi_{i,\theta_i}(a_i|o_i(s)))]. \quad (5)$$

We then propose a decentralized actor-critic algorithm based on the policy gradient formula (5). For any $z \in \{r, c\}$, the action-value function $Q_\theta^z(\cdot, \cdot)$ is approximated by a family of critic approximators $Q^z(\cdot, \cdot; \omega^z)$ parameterized by $\omega^z \in \mathbb{R}^{K_z}$ with $K_z \ll |\mathcal{S}| \times |\mathcal{A}|$, and each agent $i \in \mathcal{N}$ estimates

$Q_\theta^z(\cdot, \cdot)$ with $Q^z(\cdot, \cdot; \omega_i^z)$, where $\omega_i^z$ is maintained locally. Denote $W_t = [w_t(i, j)]_{N \times N}$ as the weight matrix associated to $\mathcal{G}_t$, which satisfies $w_t(i, j) \geq 0$ for any $i, j \in \mathcal{N}$, and $w_t(i, j) = 0$ if $(i, j) \notin \mathcal{E}_t$. Denote $\mathcal{N}_{i,t} = \{j : (i, j) \in \mathcal{E}_t\}$. Once a tuple $(s_t, a_t, r_{i,t+1}, c_{i,t+1}, s_{t+1}, a_{t+1})$ is collected from the environment by agent $i \in \mathcal{N}$, its critic parameters are updated by

$$\tilde{\omega}_{i,t+1}^z = \omega_{i,t}^z + \beta_{\omega,t} \delta_{i,t}^z \nabla_{\omega^z} Q_t^z(\omega_{i,t}^z), \quad \omega_{i,t+1}^z = \sum_{j \in \mathcal{N}_{i,t}} w_t(i, j) \tilde{\omega}_{j,t+1}^z, \quad \forall z \in \{r, c\}, \quad (6)$$

where $\beta_{\omega,t} > 0$ is the stepsize, $Q_t^z(\omega_{i,t}^z) = Q^z(s_t, a_t; \omega_{i,t}^z)$, $\delta_{i,t}^c = c_{i,t+1} + \gamma Q_{t+1}^c(\omega_{i,t}^c) - Q_t^c(\omega_{i,t}^c)$ and $\delta_{i,t}^r = r_{i,t+1} + \gamma(Q_{t+1}^r(\omega_{i,t}^r) - N\alpha \log(\pi_{i,\theta_{i,t}}(a_{i,t+1}|o_i(s_{t+1})))) - Q_t^r(\omega_{i,t}^r)$ are local temporal difference (TD) errors. In (6), the critic parameters of each agent $i \in \mathcal{N}$ are first updated based on local reward, cost, and policy information, and are then processed through the consensus update using critic parameters shared by neighboring agents. This allows each agent to update its critic parameters in a decentralized manner. Compared to existing decentralized MARL algorithms (Zhang et al., 2018; Chen et al., 2022; Hu et al., 2024), each agent in our algorithm additionally maintains a critic for $Q_\theta^c$, which will be employed in the policy update.

For the actor parameter update, we assume that all agents share the same policy class from Theorem 1, such that we have $\Theta_1 = \cdots = \Theta_N \subseteq \mathbb{R}^m$. We then define a copy operator $[\cdot] : \mathbb{R}^q \to \mathbb{R}^{Nq}$, which satisfies $[v] = \mathbb{1} \otimes v$ for any $v \in \mathbb{R}^q$, where $\mathbb{1} \in \mathbb{R}^N$ and $q$ is an arbitrary positive integer. Based on (5), the actor parameter of agent $i$ is updated by

$$\tilde{\theta}_{i,t+1} = \theta_{i,t} + \beta_{\theta,t} N \eta_{i,t} \psi_{i,t}, \quad \theta_{i,t+1} = \sum_{j \in \mathcal{N}_{i,t}} w_t(i, j) \tilde{\theta}_{j,t+1}, \quad (7)$$

where $\beta_{\theta,t} > 0$ is the stepsize for the actor, $\eta_{i,t} = Q_t^r(\omega_{i,t}^r) - \alpha \log(\pi_{[\theta_{i,t}]}(a_t|s_t)) - \lambda_{i,t} Q_t^c(\omega_{i,t}^c)$, $\psi_{i,t} = \nabla_{\theta_i} \log(\pi_{i,\theta_{i,t}}(a_{i,t}|o_i(s_t)))$, and $\lambda_{i,t}$ is the local Lagrangian multiplier maintained by agent $i$. Inspired by Theorem 1, the policy consensus step is incorporated into (7), which allows us to estimate the entropy regularization term in (5) with $\log(\pi_{[\theta_{i,t}]}(a_t|s_t)) = \sum_{j=1}^N \log(\pi_{i,\theta_{i,t}}(a_{j,t}|o_j(s_t)))$. Note that $o_j(s_t)$ is available to agent $i$ based on the permuted observation $o_i(Ms_t)$ with $m_i = j$ using condition (iii) in Definition 1, so that the term $\log(\pi_{[\theta_{i,t}]}(a_t|s_t))$ can be calculated locally. As a result, each agent can update its actor parameter in a decentralized manner.

Finally, each agent $i$ updates its dual variable by

$$\tilde{\lambda}_{i,t+1} = \Gamma_\lambda \left[ \lambda_{i,t} - \beta_{\lambda,t} \mathbb{E}_{\tilde{s} \sim \rho, \tilde{a} \sim \pi_{[\theta_{i,t}]}}[b - Q^c(\tilde{s}, \tilde{a}; \omega_{i,t}^c)] \right], \quad \lambda_{i,t+1} = \sum_{j \in \mathcal{N}_{i,t}} w_t(i, j) \tilde{\lambda}_{j,t+1}, \quad (8)$$

where $\beta_{\lambda,t} > 0$ is the stepsize, and $\Gamma_\lambda$ projects any scalar onto the set $[0, \lambda_{\max}]$ satisfying $\lambda_{\max} > 0$. For each agent $i \in \mathcal{N}$, $\tilde{s}$ is sampled from the known initial state distribution $\rho$, and every $\tilde{a}_i$ in $\tilde{a} = (\tilde{a}_1, \ldots, \tilde{a}_N)$ is sampled from $\pi_{i,\theta_{i,t}}(\cdot|o_i(M\tilde{s}))$ with $m_i = j, \forall j \in \mathcal{N}$. Following the similar analysis for the actor update, (8) can also be executed by each agent in a decentralized manner.

## 4 CONVERGENCE ANALYSIS

In this section, the convergence of the proposed decentralized actor-critic algorithm (6)-(8) is analyzed based on multi-timescale stochastic approximation theory. We begin by introducing standard assumptions taken from existing decentralized MARL works, where detailed discussions on these assumptions can be found in Appendix C.

**Assumption 1.** For any $s \in \mathcal{S}$, $a_i \in \mathcal{A}_i$ and $i \in \mathcal{N}$, $\pi_{i,\theta_i}(a_i|o_i(s))$ is continuously differentiable w.r.t. $\theta_i$, which satisfies $\pi_{i,\theta_i}(a_i|o_i(s)) \geq \kappa$ for some $\kappa > 0$. Let $P_\theta$ be the state transition matrix of the Markov chain $\{s_t\}_{t \geq 0}$ w.r.t. $\pi_\theta$ for any $\theta \in \Theta$, such that $P_\theta(s'|s) = \sum_{a \in \mathcal{A}} \pi_\theta(a|s) P(s'|s, a)$, $\forall s, s' \in \mathcal{S}$. The Markov chain $\{s_t\}_{t \geq 0}$ is irreducible and aperiodic under any policy $\pi_\theta$.

**Assumption 2.** The weight matrix sequence $\{W_t\}_{t \geq 0}$ satisfies the following conditions: (i) $W_t$ is row stochastic and $\mathbb{E}[W_t]$ is column stochastic, $\forall t \geq 0$, i.e., $W_t \mathbb{1} = \mathbb{1}$ and $\mathbb{1}^T \mathbb{E}[W_t] = \mathbb{1}^T$. (ii) The spectral norm of $\mathbb{E}[W_t^T(I - \mathbb{1}\mathbb{1}^T/N)W_t]$ is strictly smaller than one. (iii) $W_t$ is conditionally independent of $r_{i,t+1}$ and $c_{i,t+1}$ for any $i \in \mathcal{N}$ provided the $\sigma$-algebra generated by the random variables before time $t$.

**Assumption 3.** The instantaneous reward $r_{i,t+1}$ and cost $c_{i,t+1}$ are uniformly bounded for any agent $i \in \mathcal{N}$ and $t \geq 0$.

**Assumption 4.** $Q_\theta^z(s,a)$ is approximated by linear critic functions $Q^z(s,a;\omega^z) = \phi^z(s,a)^T \omega^z$ for any $z \in \{r,c\}$. The feature vectors $\phi^z(s,a)$ are uniformly bounded for any $s \in \mathcal{S}$ and $a \in \mathcal{A}$, and the feature matrix $\Phi^z \in \mathbb{R}^{N_s^a \times K_z}$ has full column rank, where $N_s^a = |\mathcal{S}| \times |\mathcal{A}|$.

**Assumption 5.** Stepsize sequences $\{\beta_{\omega,t}\}_{t\geq0}$, $\{\beta_{\theta,t}\}_{t\geq0}$ and $\{\beta_{\lambda,t}\}_{t\geq0}$ satisfy $\beta_{\omega,t}, \beta_{\theta,t}, \beta_{\lambda,t} > 0$, $\sum_{t=0}^\infty \beta_{\omega,t} = \infty$, $\sum_{t=0}^\infty \beta_{\theta,t} = \infty$, $\sum_{t=0}^\infty \beta_{\lambda,t} = \infty$, $\sum_{t=0}^\infty \beta_{\omega,t}^2 + \beta_{\theta,t}^2 + \beta_{\lambda,t}^2 < \infty$, $\beta_{\theta,t} = o(\beta_{\omega,t})$, $\beta_{\lambda,t} = o(\beta_{\theta,t})$, and $\lim_{t\to\infty} \beta_{\omega,t+1}\beta_{\omega,t}^{-1} = \lim_{t\to\infty} \beta_{\theta,t+1}\beta_{\theta,t}^{-1} = \lim_{t\to\infty} \beta_{\lambda,t+1}\beta_{\lambda,t}^{-1} = 1$.

**Assumption 6.** The critic update is stable almost surely (a.s.) for any $i \in \mathcal{N}$, i.e., $\sup_{t\to\infty} \|\omega_{i,t}^z\| < \infty$ a.s. for any $z \in \{r,c\}$. For the actor update, $\{\theta_{i,t}\}_{t\geq0}$ belongs to a compact set in $\Theta_i$ for any $i \in \mathcal{N}$ and $t \geq 0$.

**Convergence of the critic.** Assumption 5 indicates that the critic parameters update at the fastest timescale, which allows us to analyze their convergence under fixed $\theta$ and $\bar{\lambda} = [\lambda_1, \ldots, \lambda_N]^T$ based on the multi-timescale stochastic approximation theory (Borkar, 2008). By abuse of notation, let $P_\theta$ be the transition matrix of the state-action pairs with $P_\theta(s',a'|s,a) = P(s'|s,a)\pi_\theta(a'|s')$. Based on Assumption 1, denote the stationary distribution of each state $s \in \mathcal{S}$ by $\nu_\theta(s)$, satisfying $\nu_\theta(s) = \lim_{t\to\infty} \mathbb{P}_\theta(s_t = s)$, based on which we define a stationary distribution matrix as $D_\theta^{s,a} = \text{diag}[\nu_\theta(s)\pi_\theta(a|s), s \in \mathcal{S}, a \in \mathcal{A}]$. Then, we denote $\bar{R} = \text{col}[\bar{R}(s,a), s \in \mathcal{S}, a \in \mathcal{A}] \in \mathbb{R}^{N_s^a}$, $\bar{C} = \text{col}[\bar{C}(s,a), s \in \mathcal{S}, a \in \mathcal{A}] \in \mathbb{R}^{N_s^a}$, and $\Omega_\theta = \text{col}[\alpha\log(\pi_\theta(a|s)), s \in \mathcal{S}, a \in \mathcal{A}] \in \mathbb{R}^{N_s^a}$. For any vector $Q = \text{col}[Q(s,a), s \in \mathcal{S}, a \in \mathcal{A}] \in \mathbb{R}^{N_s^a}$, we can define two operators $\mathcal{T}_\theta^r, \mathcal{T}_\theta^c : \mathbb{R}^{N_s^a} \to \mathbb{R}^{N_s^a}$, which respectively take the forms

$$\mathcal{T}_\theta^r[\bar{Q}] = \bar{R} + \gamma P_\theta(\bar{Q} - \Omega_\theta), \quad \mathcal{T}_\theta^c[\bar{Q}] = \bar{C} + \gamma P_\theta \bar{Q}. \tag{9}$$

**Theorem 3.** *Under Assumptions 1-6, for any policy $\pi_\theta$ with the sequences $\{\omega_{i,t}^r\}_{t\geq0}$ and $\{\omega_{i,t}^c\}_{t\geq0}$ generated by (6), it satisfies $\lim_{t\to\infty} \omega_{i,t}^z = \omega_\theta^z$ a.s. for any $i \in \mathcal{N}$, where*

$$(\Phi^z)^T D_\theta^{s,a} (\mathcal{T}_\theta^z[\Phi^z \omega_\theta^z] - \Phi^z \omega_\theta^z) = 0, \quad \forall z \in \{r,c\}. \tag{10}$$

The proof of Theorem 3 can be found in Appendix D. Note that the learned critic parameters $\omega_\theta^r$ and $\omega_\theta^c$ in (10) correspond to the Mean Square Projected Bellman Error (MSPBE) minimizers respectively associated to $Q_\theta^r$ and $Q_\theta^c$ (Zhang et al., 2018). This theorem indicates that each agent in our algorithm can learn good approximators for the global action-value functions using parameter information from its neighbors only.

**Convergence of the actor.** We then show the convergence of the actor parameters under fixed $\bar{\lambda}$. Denote $\eta_{i,t,\theta}^{\lambda_i} = (\phi_t^r)^T \omega_\theta^r - \alpha\log(\pi_{[\theta_i]}(a_t|s_t)) - \lambda_i(\phi_t^c)^T \omega_\theta^c$ and $\psi_{i,t,\theta} = \nabla_{\theta_i}\log(\pi_{i,\theta_i}(a_{i,t}|o_i(s_t)))$, where $\phi_t^z = \phi^z(s_t, a_t) \in \mathbb{R}^{K_z}$ for any $z \in \{r,c\}$.

**Theorem 4.** *Under Assumptions 1-6, for any fixed $\bar{\lambda}$, with the sequences $\{\theta_{i,t}\}_{t\geq0}$ generated by (7), we have $\lim_{t\to\infty} \theta_{i,t} = \hat{\theta}_{\bar{\lambda}}$ a.s. for any $i \in \mathcal{N}$, where $\hat{\theta}_{\bar{\lambda}}$ is a point in the set of asymptotically stable equilibria of*

$$\dot{\hat{\theta}} = \mathbb{E}_{s_t \sim d_{[\hat{\theta}]}, a_t \sim \pi_{[\hat{\theta}]}} \left[ \sum_{i\in\mathcal{N}} \eta_{i,t,[\hat{\theta}]}^{\lambda_i} \psi_{i,t,[\hat{\theta}]} \right]. \tag{11}$$

The proof of Theorem 4 can be found in Appendix E. Note that the ordinary differential equation (ODE) (11) is different from those in Chen et al. (2022); Hu et al. (2024), which additionally contains the terms w.r.t. $\bar{\lambda}$ and $Q^c$ due to that all agents aim to maximize the Lagrangian (3) rather than the objective function in (2).

**Convergence of the dual variable.** Based on the projection operator $\Gamma_\lambda$ in (8), we define an operator $\hat{\Gamma}_\lambda$ as $\hat{\Gamma}_\lambda[f(\lambda)] = \lim_{\eta\to0^+} \{\Gamma_\lambda[\lambda + \eta f(\lambda)] - \lambda\}/\eta$, where $\lambda \in [0, \lambda_{\max}]$ and $f : [0, \lambda_{\max}] \to \mathbb{R}$ is a continuous function. We then introduce an additional assumption from Bhatnagar (2010) for the convergence analysis of the dual variables.

**Assumption 7.** For any dual variable vector $\bar{\lambda}$, the convergent point $\hat{\theta}_{\bar{\lambda}}$ of (11) is continuous in $\bar{\lambda}$.

**Theorem 5.** *Under Assumptions 1-7, for the sequences $\{\lambda_{i,t}\}_{t\geq0}$ generated by (8), it satisfies $\lim_{t\to\infty} \lambda_{i,t} = \lambda^*$ a.s. for any $i \in \mathcal{N}$, where $\lambda^*$ is a point in the set of asymptotically stable equilibria of*

$$\dot{\lambda} = \hat{\Gamma}_\lambda \left[ \mathbb{E}_{\tilde{s}\sim\rho, \tilde{a}\sim\pi_{[\hat{\theta}_{[\lambda]}]}}[Q^c(\tilde{s}, \tilde{a}; \omega_{[\hat{\theta}_{[\lambda]}]}^c) - b] \right]. \tag{12}$$

The proof of Theorem 5 is given in Appendix F. We then analyze the constraint satisfaction for the learned policy $\pi_{[\hat{\theta}_{[\lambda^*]}]}$. Let $\Lambda = \{\lambda : \hat{\Gamma}_\lambda[\mathbb{E}_{\tilde{s}\sim\rho,\tilde{a}\sim\pi_{[\hat{\theta}_{[\lambda]}]}}[Q^c(\tilde{s},\tilde{a};\omega^c_{[\hat{\theta}_{[\lambda]}]}) - b]] = 0, \lambda \in [0, \lambda_{\max}]\}$, and $\hat{\Lambda} = \{\lambda : \hat{\Gamma}_\lambda[\mathbb{E}_{\tilde{s}\sim\rho,\tilde{a}\sim\pi_{[\hat{\theta}_{[\lambda]}]}}[Q^c(\tilde{s},\tilde{a};\omega^c_{[\hat{\theta}_{[\lambda]}]}) - b]] = 0, \lambda \in [0, \lambda_{\max}]\}$.

**Proposition 1.** *For any $\lambda^* \in \hat{\Lambda}$, we have $\mathbb{E}_{\tilde{s}\sim\rho,\tilde{a}\sim\pi_{[\hat{\theta}_{[\lambda^*]}]}}[Q^c(\tilde{s},\tilde{a};\omega^c_{[\hat{\theta}_{[\lambda^*]}]})] \leq b$.*

**Proposition 2.** *For any $\lambda^* \in \Lambda$, if it satisfies $\mathbb{E}_{\tilde{s}\sim\rho,\tilde{a}\sim\pi_{[\hat{\theta}_{[\lambda^*]}]}}[Q^c(\tilde{s},\tilde{a};\omega^c_{[\hat{\theta}_{[\lambda^*]}]})] < b$, then we have $\lambda^* = 0$.*

The proofs of Propositions 1 and 2 can be found in Appendix G. It is indicated in Proposition 1 that the safety constraint in (2) can be approximately satisfied by $\pi_{[\hat{\theta}_{[\lambda^*]}]}$ when $\lambda^* \in \hat{\Lambda}$, where the value function associated to the cost in (2) is estimated using the critic approximator based on the Bellman equation (Sutton & Barto, 2018). In practice, we can select a sufficiently large $\lambda_{\max}$ to ensure that the learned $\lambda^*$ belongs to $\hat{\Lambda}$. Proposition 2 demonstrates that $\lambda^* = 0$ when the approximated safety constraint is strictly satisfied. In this case, the constrained MG problem (2) will reduce to a regular MG problem described by $\max_{\theta\in\Theta} J^r(\theta)$ approximately (Bhatnagar, 2010). Apart from the theoretical analysis, we also empirically evaluate the convergence of our decentralized algorithm on a toy experiment, where the simulation results can be found in Appendix H.

## 5 PRACTICAL ALGORITHM DESIGN

Even though the decentralized algorithm proposed in Section 3 is theoretically convergent, the performance of this algorithm can be severely limited by the standard assumptions, such as finite state and action space setting, the linear critic approximator and the decreasing learning rate. Note that this algorithm can also be sample inefficient due to the on-policy training architecture. To this end, we propose a practical decentralized algorithm by modifying the local update steps in (6)-(8) using DRL (Lillicrap et al., 2016).

In our practical algorithm, both the critic and the actor are modeled as neural networks (NNs). Each agent $i \in \mathcal{N}$ maintains a replay buffer $\mathcal{B}_i = \{(s_t, a_t, r_{i,t+1}, c_{i,t+1}, s_{t+1})\}_t$ and two target critic NNs denoted by $Q^z(\cdot,\cdot;\bar{\omega}^z_i)$ for $z \in \{r,c\}$. Let $o_{i,t} = o_i(s_t)$. Based on (6), we consider the following loss function for the local update of critic parameters:

$$J^z_Q(\omega^z_i) = \mathbb{E}_{(s_t,a_t,r_{i,t+1},c_{i,t+1},s_{t+1})\sim\mathcal{B}_i}\left[(Q^z(s_t,a_t;\omega^z_i) - y^z_i)^2\right], \quad \forall z \in \{r,c\}, \tag{13}$$

in which $y^r_i = r_{i,t+1} + \gamma(Q^r(s_{t+1},a_{t+1};\bar{\omega}^r_i) - N\alpha\log(\pi_{i,\theta_i}(a_{i,t+1}|o_{i,t+1})))$ and $y^c_i = c_{i,t+1} + \gamma Q^c(s_{t+1},a_{t+1};\bar{\omega}^c_i)$. Based on the consensus update of the actor parameters, each agent $i \in \mathcal{N}$ approximates the other agents' policies with its own policy, such that each $a_{j,t+1}$ in (13) is sampled from $\pi_{i,\theta_i}(\cdot|o_{j,t+1})$. Let $\hat{\nabla}_{\omega^z_i} J^z_Q(\omega^z_i)$ denote the stochastic gradient of (13) calculated using a batch of data $\mathcal{D}_i$ from $\mathcal{B}_i$. Based on (7), we consider the following loss function for the local update of the actor parameters:

$$J_\pi(\theta_i) = \mathbb{E}_{s_t\sim\mathcal{B}_i,a_t\sim\pi_\theta}\left[\alpha\log(\pi_\theta(a_t|s_t)) - Q^r(s_t,a_t;\omega^r_i) + \lambda_i Q^c(s_t,a_t;\omega^c_i)\right]. \tag{14}$$

Note that (14) indicates the similar parameter update direction as (7) with the only difference that the gradient of (14) is calculated based on the experiences stored in $\mathcal{B}_i$ rather than those collected in an on-policy manner, which is a common trick used in existing DRL algorithms (Lillicrap et al., 2016). In Appendix I.1, we provide a detailed analysis on the relationship between (7) and (14). Similar to the setting in (13), each agent $i \in \mathcal{N}$ samples the other agents' actions in (14) using its local policy due to the policy consensus. Let $\hat{\nabla}_{\theta_i} J_\pi(\theta_i)$ be the stochastic gradient of (14). For the local update of dual variables, we consider the following loss function based on (8):

$$J_D(\lambda_i) = \mathbb{E}_{(s_t,a_t)\sim\mathcal{B}_i}\left[\lambda_i(b - Q^c(s_t,a_t;\omega^c_i))\right]. \tag{15}$$

Here, the joint action $a_t$ is sampled from the replay buffer together with $s_t$, rather than being obtained from the current policy $\pi_{[\theta_i]}$. Note that this trick can effectively enforce constraint satisfaction, and has been employed in existing DRL-based off-policy primal-dual algorithms (Ray et al., 2019; Yang et al., 2021). Let $\hat{\nabla}_{\lambda_i} J_D(\lambda_i)$ be the stochastic gradient of (15). Finally, we employ the automatic entropy adjustment mechanism (Haarnoja et al., 2019) to balance exploration and exploitation during

training. Let $\alpha_i$ be the local temperature parameter of agent $i \in \mathcal{N}$. We consider the following loss function:

$$J(\alpha_i) = \mathbb{E}_{s_t \sim \mathcal{B}_i, a_{i,t} \sim \pi_{i,\theta_i}} \left[ -\alpha_i (\log(\pi_{i,\theta_i}(a_{i,t}|o_{i,t})) + \mathcal{H}_0) \right], \tag{16}$$

where $\mathcal{H}_0$ denotes the target policy entropy. Let $\hat{\nabla}_{\alpha_i} J(\alpha_i)$ be the stochastic gradient of (16). Recall that all the loss functions can be calculated by each agent using local information only. Hence, our practical algorithm can still maintain its decentralized training nature. Finally, the pseudocode of this algorithm, named decentralized primal-dual actor-critic with entropy regularization (DPDAC-ER), is shown in Appendix I.2.

## 6 EXPERIMENTS

**Environments.** We build three safety-aware swarm robotic tasks[1] to evaluate the proposed algorithm, which are *Aggregation*, *Swapping* and *Formation*. All the tasks are implemented in the classic Multi-Agent Particle Environment (Lowe et al., 2017), where all agents follow a discrete second-order dynamics model and move within a shared 2D space. Each task contains 10 agents, and the action space for each agent is continuous. Note that the tasks are visualized in Appendix J.1, which are also briefly described as follows. *Aggregation*: Each agent aims to aggregate at the origin. It receives a higher reward as its position gets closer to the origin, and incurs a penalty if it collides with other agents. There exists a hazardous area in the environment that covers the origin. Each agent $i$ incurs a cost of $c_{i,t+1} = 1$ if it enters this area, and $c_{i,t+1} = 0$ otherwise. *Swapping* (Hu et al., 2023): Each agent aims to move to the initial position of the agent located in the diagonal area. It receives a higher reward as it approaches the target position, and incurs a penalty if it collides with other agents. There exists velocity saturation constraints on the agents. Each agent $i$ incurs a cost of $c_{i,t+1} = 1$ if the 2-norm of its velocity exceeds a given threshold, and $c_{i,t+1} = 0$ otherwise. *Formation* (Agarwal et al., 2020): All agents aim to evenly distribute themselves along a circumference centered at their mean position. They share a team reward which increases as the total distance between their positions and the ideal formation positions decreases. There exists a landmark in the environment. The team cost function is defined as the distance between the agents' mean position and the landmark's position, which is shared by all agents.

**Baselines.** In our experiments, two CT-based off-policy MARL algorithms MASAC and MASAC-Lagrangian (MASAC-Lag) are chosen as baselines. MASAC (Willemsen et al., 2021) solely aims to maximize the total reward, without considering safety. MASAC-Lag extends SAC-Lagrangian (Ray et al., 2019; Yang et al., 2021) under the MARL setting, which also employs the primal-dual method for learning safe policies. In these CT-based algorithms, only one actor NN is built due to the homogeneity of agents, which is trained using the mean reward and cost directly. In addition, two decentralized MARL baselines named DPDAC and DAC-ER are incorporated. DPDAC is a variant of our algorithm, which doesn't employ the entropy regularizer, such that $\alpha = 0$. DAC-ER (Hu et al., 2024) reported state-of-the-art learning performance on multi-robot coordination tasks considering continuous action spaces. Similar to MASAC, this algorithm doesn't consider constraint satisfaction. In Appendix J.2, the differences between DPDAC-ER and the baselines are summarized, and the implementation details of these algorithms are introduced.

**Results.** The learning performance of all algorithms is evaluated over five independent trials across three tasks, where the smoothed learning curves are shown in Fig. 1. Note that the policies learned by DAC-ER and MASAC obtain the largest returns in all the tasks, but they are unsafe for largely violating the constraints. Our proposed algorithm DPDAC-ER demonstrates similar learning performance to MASAC-Lag in terms of both reward and cost, which also exhibits excellent learning stability across all the trials. Both the algorithms can converge to safe policies across all the tasks at the cost of relative lower returns. Note that DPDAC-ER outperforms MASAC-Lag in terms of reward in the Formation task. This result could be explained by that the agents in DPDAC-ER have different policies at the early stage of the training process, which helps to sample richer experiences for policy learning. It is worth pointing out that DPDAC has the worst learning stability among the algorithms, which fails to learn safe policies in the Formation task. This result can be attributed to the poor exploration capability of the vanilla policy gradient method in continuous spaces, which highlights the importance of incorporating the entropy regularization mechanism in our algorithm.

---

[1]https://github.com/ICLR2025anonymous/DPDAC-ER/

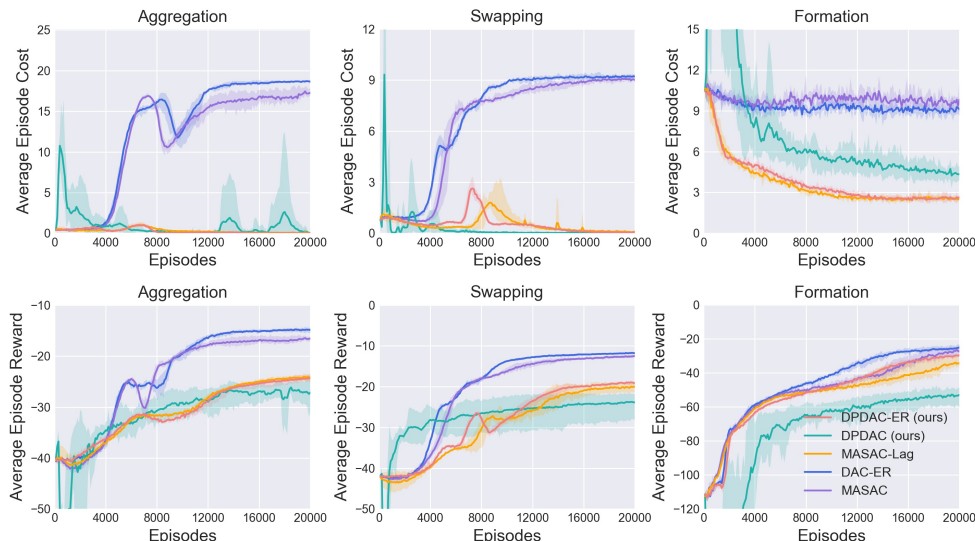

Figure 1: Learning curves of five algorithms across three tasks. The cost curves are displayed in the upper row (with lower values being better), while the reward curves are displayed in the bottom row (with higher values being better).

**Ablation: communication.** We then study the influence of communication networks on our decentralized algorithm. Apart from the sparse communication, we additionally consider two extreme communication scenarios: all-to-all communication and no communication. The learning curves of DPDAC-ER in all the scenarios are displayed in Fig. 5 in Appendix J.3. It can be observed that DPDAC-ER has the worst learning performance in the no communication scenario, which fails to learn safe policies in two tasks. This demonstrates the necessity for performing parameter consensus in our algorithm. Note that blindly increasing communication links doesn't improve learning performance of DPDAC-ER. This inspires us to consider sparse communication networks at the beginning when deploying this algorithm in practice.

**Ablation: constraints.** We also evaluate DPDAC-ER under different cost thresholds to study how it balances the trade-off between performance and safety. The learning curves can be found in Fig. 6 in Appendix J.4. We can learn that DPDAC-ER obtains higher returns at the end of training when the safety constraints become weaker, which demonstrates that the effectiveness of our algorithm can be maintained across different safety levels.

**Ablation: local observation.** We finally evaluate DPDAC-ER under the local observation setting, where the global state information is not available to each agent. We provide detailed modifications of our algorithms and experimental settings in Appendix J.5. The simulation results show that the modified version of our algorithm can maintain its learning performance under the local observation and decentralized training settings.

**Additional experiments.** In Appendix J.6, we further compare our algorithm with DAC-ER, which employs a reward-shaping mechanism to address safety constraints. The simulation results reveal that agents employing this method face challenges in balancing reward maximization and constraint satisfaction in most scenarios. Additionally, we compare DPDAC-ER with DPDAC in a customized 3D Formation task. The results reveal a significant decline in the learning stability of DPDAC-ER. In contrast, our algorithm maintains both sample efficiency and learning stability, highlighting the importance of incorporating the entropy regularization mechanism for high-dimensional tasks.

## 7 CONCLUSION

In this paper, a decentralized safe MARL problem for networked multi-agent systems has been investigated under the entropy-regularized setting. A subclass of constrained MGs considering homogeneous agents has been characterized, where policy sharing provably preserves both optimality

and safety. An on-policy decentralized primal-dual actor-critic algorithm has been proposed, which is asymptotically convergent under the linear critic assumption. For practical applications, a decentralized off-policy version of the proposed algorithm has been developed based on the DRL training architecture. Simulation results on three safety-aware continuous multi-robot tasks demonstrate the effectiveness of the proposed decentralized algorithm. Our future work aims to develop practical decentralized safe MARL algorithms under the local observation setting.

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

# A  APPENDIX FOR THE HOMOGENEOUS CONSTRAINED MG

## A.1  AN ILLUSTRATIVE EXAMPLE

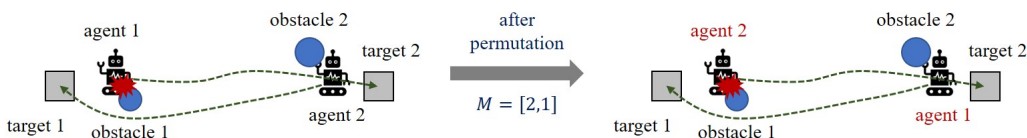

Figure 2: A safe position swapping task for two agents.

Consider a scenario where two agents control two homogeneous robots to perform a safe position-swapping task. For simplicity, the robots follow a first-order discrete dynamics model, and we denote $p_{L,t}$ ($p_{R,t}$) as the robot's position controlled by agent 1 (2) before permutation. As shown in Fig. 2, the environment also contains landmarks including two targets and two static obstacles, whose positions are denoted by $p_j^t$ and $p_k^o$, respectively, where $j, k \in \{1, 2\}$. Under the full observation setting, the local states at time $t$ are $s_{1,t} = (p_{L,t}, p_1^t, p_2^t, p_1^o, p_2^o)$ and $s_{2,t} = (p_{R,t}, p_1^t, p_2^t, p_1^o, p_2^o)$, and it is apparent that $\mathcal{S}_1 = \mathcal{S}_2$. The action of each agent is the feasible velocity command sent to its robot, such that $\mathcal{A}_1 = \mathcal{A}_2$ due to the homogeneity of robots. The reward for each agent is the negative of the distance between the robot it controls and its corresponding target, such that we have $R_1(s_t, a_t) = -\|p_{L,t} - p_2^t\|$ and $R_2(s_t, a_t) = -\|p_{R,t} - p_1^t\|$. The cost for each agent is an indicator function, which becomes 1 the robot it controls collides with other landmarks or agents. From Fig. 2, we have $C_1(s_t, a_t) = 1$ and $C_2(s_t, a_t) = 0$. Each agent's local observation includes the absolute position of the robot it controls and a list of the relative positions between the robot and landmarks, sorted by distance. Here, we have $o_1(s_t) = (p_{L,t}, p_{L,t} - p_1^t, p_{L,t} - p_2^t, p_{L,t} - p_1^o, p_{L,t} - p_2^o)$ and $o_2(s_t) = (p_{R,t}, p_{R,t} - p_2^t, p_{R,t} - p_1^t, p_{R,t} - p_2^o, p_{R,t} - p_1^o)$. After the permutation $M = [2, 1]$, we have $M(s_{1,t}, s_{2,t}) = (s_{2,t}, s_{1,t})$ and $M(a_{1,t}, a_{2,t}) = (a_{2,t}, a_{1,t})$ based on the definition of the permutation $M$, i.e., agent 1 (2) now controls the robot on the right (left) with the action $a_{2,t}$ ($a_{1,t}$). Due to that the robots are homogeneous, the state transition function is permutation invariant. We can obtain that $R_1(Ms_t, Ma_t) = R_2(s_t, a_t)$ and $R_2(Ms_t, Ma_t) = R_1(s_t, a_t)$ based on the definition of the reward. Thus, the permutation preserving property holds for the reward. Note that this property also holds for the cost due to that $C(Ms_t, Ma_t) = (0, 1) = MC(s_t, a_t)$. Finally, we have $o_1(Ms_t) = o_2(s_t)$ and $o_2(Ms_t) = o_1(s_t)$, which means that the permutation preserving property holds for the observation. As a result, this safe position swapping task is an example of the homogeneous constrained MG.

## A.2  PRACTICAL EXAMPLES

Apart from the illustrative example, we now show that practical multi-agent coordination tasks can also be examples of the homogeneous constrained MG, which are shown as follows:

**Leader-following flocking.** In this task, each follower agent must maintain specific distances from a moving leader and other agents. The local state of an individual agent is defined as the concatenation of its own system state and the leader's system state. The agent's observation includes its own system state and the relative system state information of both the other agents and the leader. An agent receives a higher reward when its distance to the leader approaches a predefined value, while incurring costs if its distances from other agents are either too large or too small.

**Data collection with UAVs.** In this task, a team of homogeneous UAVs operates in an urban environment to collect as much data as possible from several Internet of Things (IoT) devices. Each UAV is rewarded based on the amount of data collected within a given time interval and incurs a cost if it collides with other UAVs or enters no-fly zones. Each UAV's observation includes its absolute position, the relative positions of other UAVs, IoT devices, and no-fly zones, as well as the remaining data at each IoT device.

A.3   PROOF OF THEOREM 1

Based on condition (iii) in Definition 1, there exists a one-to-one mapping between $\Pi$ and $\Pi_o$ due to that the observation function $o_i$ is bijective for any $i \in \mathcal{N}$. Thus, for the optimal joint policy $\pi^* \in \Pi$, there exists an observation-based joint policy $\pi_o^* = \prod_{i=1}^{N} \pi_{i,o}^* \in \Pi_o$ satisfying $J^r(\pi_o^*) = J^r(\pi^*)$ and $J^c(\pi_o^*) = J^c(\pi^*) \leq b$. On the other hand, for any permutation $M = [m_1, \cdots, m_N] \in \mathcal{M}$ and $\pi_o = \prod_{i=1}^{N} \pi_{i,o} \in \Pi_o$, if it satisfies $\pi_{i,o}(\cdot|o_i(s)) = \pi_{j,o}(\cdot|o_j(Ms))$ with $i = m_j$ for any $s \in \mathcal{S}$ and $i \in \mathcal{N}$, then the entropy of $\pi_o$ is permutation invariant, i.e., $\mathcal{H}(\pi_o(\cdot|s)) = \mathcal{H}(\pi_o(\cdot|Ms))$. Thus, due to the permutation invariance of the state transition probability, the average reward, the average cost and the policy entropy, we have $\pi_{i,o}^*(\cdot|o_i(s)) = \pi_{j,o}^*(\cdot|o_j(Ms))$. Recall that $o_i(s) = o_j(Ms)$ from condition (iii) in Definition 1. We can obtain that $\pi_{i,o}^*(\cdot|o) = \pi_{j,o}^*(\cdot|o)$ for any $i \in \mathcal{N}$ and $o \in \mathcal{O}$. Note that the permutation $M \in \mathcal{M}$ can be arbitrary. Thus, we have $\pi_{1,o}^*(\cdot|o) = \cdots = \pi_{N,o}^*(\cdot|o)$, which completes the proof.

## B   PROOF OF THEOREM 2

For the Lagrangian (3), it holds

$$\nabla_\theta L(\theta, \lambda) = \nabla_\theta J^r(\theta) - \lambda \nabla_\theta J^c(\theta). \tag{17}$$

Then, based on the entropy-regularized policy gradient theorem (Cayci et al., 2021) and the vanilla policy gradient theorem (Sutton et al., 1999), we can directly obtain

$$\nabla_\theta J^r(\theta) = \mathbb{E}_{s \sim d_\theta, a \sim \pi_\theta}[(Q_\theta^r(s,a) - \alpha \log(\pi_\theta(a|s)))\nabla_\theta \log(\pi_\theta(a|s))], \tag{18}$$

$$\nabla_\theta J^c(\theta) = \mathbb{E}_{s \sim d_\theta, a \sim \pi_\theta}[Q_\theta^c(s,a)\nabla_\theta \log(\pi_\theta(a|s))]. \tag{19}$$

As a result, we have

$$\nabla_\theta L(\theta, \lambda) = \mathbb{E}_{s \sim d_\theta, a \sim \pi_\theta}[A_\theta^\lambda(s,a)\nabla_\theta \log(\pi_\theta(a|s))], \tag{20}$$

where $A_\theta^\lambda(s,a) = Q_\theta^r(s,a) - \alpha \log(\pi_\theta(a|s)) - \lambda Q_\theta^c(s,a)$. Recall that $\theta = [(\theta_1)^T, \ldots, (\theta_N)^T]^T$ and $\log(\pi_\theta(a|s)) = \sum_{i \in \mathcal{N}} \log(\pi_{i,\theta_i}(a_i|o_i(s)))$. The gradient of $L(\theta, \lambda)$ w.r.t. $\theta_i$ takes the form

$$\nabla_{\theta_i} L(\theta, \lambda) = \mathbb{E}_{s \sim d_\theta, a \sim \pi_\theta}[A_\theta^\lambda(s,a)\nabla_{\theta_i} \log(\pi_{i,\theta_i}(a_i|o_i(s)))], \tag{21}$$

which completes the proof.

## C   DISCUSSIONS ABOUT ASSUMPTIONS

**Assumption 1.** This assumption is standard in early works on actor-critic algorithms with function approximation (Bhatnagar et al., 2009; Suttle et al., 2019). Note that the first part of this assumption is reasonable due to that entropy regularization penalizes overly deterministic policies.

**Assumption 2.** The conditions on the weight matrices $\{W_t\}_{t \geq 0}$ in this assumption are widely considered in existing works on decentralized MARL (Zhang et al., 2018; Chen et al., 2022; Hu et al., 2024). In condition (i), the row stochasticity of $W_t$ requires each agent to make the weights assigned to the updates coming from its neighbors summing to one. The column stochasticity of $W_t$ is only required to hold on average. This allows us to incorporate various gossip types of communication schemes, such as the broadcast gossip scheme and the pairwise gossip scheme, for the networked multi-agent system (Bianchi & Jakubowicz, 2013). Condition (ii) is related to the connectivity of the communication topology, which holds for the random gossip schemes mentioned above if and only if the underlying communication graph is connected (Bianchi & Jakubowicz, 2013). Condition (iii) means that $W_t$, $r_{i,t+1}$ and $c_{i,t+1}$ are independent conditioned on the past. This is common for practical multi-agent systems, since the random communication link failures and the gossip schemes are usually independent of the past and irrelevant to the rewards as well as the costs received by the agents (Zhang et al., 2018).

**Assumption 3.** This assumption can be easily satisfied as the reward and cost functions are typically designed manually and can be bounded within limited state and action spaces.

**Assumption 4.** This assumption can also be naturally satisfied if we properly select the features for the linear critics.

**Assumption 5.** This assumption has been used in existing safe reinforcement learning algorithms which enjoy convergence based on multi-timescale stochastic approximation theory (Borkar, 2005; Bhatnagar, 2010). Note that the condition in the last sentence of this assumption will be employed to analyze parameter consensus (Zhang et al., 2018; Chen et al., 2022; Hu et al., 2024).

**Assumption 6.** In the first part of this assumption, the stability requirement on the critic parameters $\omega_{i,t}^z, \forall z \in \{r, c\}$ can be relaxed if the lower boundedness of the nonzero elements in $W_t$ can be ensured (Zhang et al., 2018), and this assumption can also be satisfied empirically through using the clip trick which can constrain the parameters of NNs within certain ranges. The second part of the assumption is commonly used in decentralized MARL works (Zhang & Zavlanos, 2019; Chen et al., 2022; Hu et al., 2024), which can be satisfied in practice when the policy parameter space is large.

# D  PROOF OF THEOREM 3

We denote $r_t = [r_{1,t}, \ldots, r_{N,t}]^T$, $c_t = [c_{1,t}, \ldots, c_{N,t}]^T$, $\omega_t^z = [(\omega_{1,t}^z)^T, \ldots, (\omega_{N,t}^z)^T]^T$, and $\delta_t^z = [\delta_{1,t}^z, \ldots, \delta_{N,t}^z]^T$ with $\delta_{i,t}^z$ defined in (6) for any $z \in \{r, c\}$. Then, we can rewrite (6) in a compact form, represented by

$$\omega_{t+1}^z = (W_t \otimes I)(\omega_t^z + \beta_{\omega,t} y_{t+1}^z), \quad \forall z \in \{r, c\}, \tag{22}$$

where $I \in \mathbb{R}^{K_z \times K_z}$, $y_{t+1}^z = [\delta_{1,t}^z(\phi_t^z)^T, \ldots, \delta_{N,t}^z(\phi_t^z)^T]^T \in \mathbb{R}^{K_z N}$ and $\phi_t^z = \phi^z(s_t, a_t) \in \mathbb{R}^{K_z}$. Let us define an average operator $\langle \cdot \rangle : \mathbb{R}^{Nq} \to \mathbb{R}^q$ for any positive integer $q$, which satisfies

$$\langle \chi \rangle = \frac{1}{N}(\mathbb{1}^T \otimes I)\chi = \frac{1}{N}\sum_{i \in \mathcal{N}}\chi_i \tag{23}$$

for any $\chi = [\chi_1^T, \ldots, \chi_N^T]^T$ with $\chi_i \in \mathbb{R}^q$. Then, we have $\omega_t^z = \mathbb{1} \otimes \langle \omega_t^z \rangle + \omega_{\perp,t}^z$, where $\mathbb{1} \otimes \langle \omega_t^z \rangle$ is the agreement component in $\omega_t^z$, and $\omega_{\perp,t}^z$ is the disagreement component in $\omega_t^z$. We first show that all $\omega_{i,t}^z$ will achieve consensus through proving that $\lim_{t\to\infty} \omega_{\perp,t}^z = 0$ a.s. for any $z \in \{r, c\}$.

**Consensus analysis.** Let $\mathcal{F}_{t,1} = \sigma(r_\tau, c_\tau, \omega_\tau^r, \omega_\tau^c, s_\tau, a_\tau, W_{\tau-1}, \tau \le t)$ be an increasing $\sigma$-algebra up to time $t$. Let $J = \frac{1}{N}(\mathbb{1}\mathbb{1}^T \otimes I)$, then it holds that $J\omega_t^z = \mathbb{1} \otimes \langle \omega_t^z \rangle, \forall z \in \{r, c\}$, based on which we can obtain

$$\begin{aligned}\omega_{\perp,t+1}^z &= (I - J)\omega_{t+1}^z = (I - J)(W_t \otimes I)(\mathbb{1} \otimes \langle \omega_t^z \rangle + \omega_{\perp,t}^z + \beta_{\omega,t} y_{t+1}^z) \\ &= (I - J)(W_t \otimes I)(\omega_{\perp,t}^z + \beta_{\omega,t} y_{t+1}^z), \end{aligned} \tag{24}$$

where the last equality holds due to that $W_t$ is row stochastic. Then, we have

$$\begin{aligned}&\mathbb{E}\left[\|\beta_{\omega,t+1}^{-1}\omega_{\perp,t+1}^z\|^2|\mathcal{F}_{t,1}\right] \\ &\le \frac{\beta_{\omega,t}^2}{\beta_{\omega,t+1}^2}\mathbb{E}\left[(\beta_{\omega,t}^{-1}\omega_{\perp,t}^z + y_{t+1}^z)^T(W_t^T(I - \mathbb{1}\mathbb{1}^T/N)W_t \otimes I)(\beta_{\omega,t}^{-1}\omega_{\perp,t}^z + y_{t+1}^z)|\mathcal{F}_{t,1}\right] \\ &\le \frac{\beta_{\omega,t}^2}{\beta_{\omega,t+1}^2}\tilde{\rho}\mathbb{E}\left[(\beta_{\omega,t}^{-1}\omega_{\perp,t}^z + y_{t+1}^z)^T(\beta_{\omega,t}^{-1}\omega_{\perp,t}^z + y_{t+1}^z)|\mathcal{F}_{t,1}\right], \\ &\le \frac{\beta_{\omega,t}^2}{\beta_{\omega,t+1}^2}\tilde{\rho}(\|\beta_{\omega,t}^{-1}\omega_{\perp,t}^z\|^2 + 2\|\beta_{\omega,t}^{-1}\omega_{\perp,t}^z\|\mathbb{E}[\|y_{t+1}^z\|^2|\mathcal{F}_{t,1}]^{\frac{1}{2}} + \mathbb{E}[\|y_{t+1}^z\|^2|\mathcal{F}_{t,1}]), \end{aligned} \tag{25}$$

where the first inequality holds since $W_t^T(I - \mathbb{1}\mathbb{1}^T/N)^T(I - \mathbb{1}\mathbb{1}^T/N)W_t = W_t^T(I - \mathbb{1}\mathbb{1}^T/N)W_t$, the second inequality holds due to conditions (ii) and (iii) in Assumption 2, and the last inequality holds based on the Cauchy–Schwarz inequality. Recall that

$$\begin{aligned}\delta_{i,t}^r &= r_{i,t+1} + \gamma((\phi_{t+1}^r)^T\omega_{i,t}^r - N\alpha\log(\pi_{i,\theta_{i,t}}(a_{i,t+1}|o_i(s_{t+1})))) - (\phi_t^r)^T\omega_{i,t}^r, \\ \delta_{i,t}^c &= c_{i,t+1} + \gamma(\phi_{t+1}^c)^T\omega_{i,t}^c - (\phi_t^c)^T\omega_{i,t}^c. \end{aligned} \tag{26}$$

From Assumptions 3 and 4, both $z_{i,t+1}$ and $\phi_t^z$ are uniformly bounded for any $z \in \{r, c\}$ and $t \ge 0$. Moreover, $\log(\pi_{i,\theta_{i,t}}(a_{i,t}|o_i(s_t)))$ is uniformly bounded for any $s_t \in \mathcal{S}$ and $a_{i,t} \in \mathcal{A}_i$ due to Assumption 1. Thus, given any $M_z > 0$, we obtain that $\mathbb{E}[\|y_{t+1}^z\|^2|\mathcal{F}_{t,1}] = \mathbb{E}\left[\sum_{i \in \mathcal{N}}\|\delta_{i,t}^z\phi_t^z\|^2|\mathcal{F}_{t,1}\right]$

is bounded on the set $\{\sup_{\tau \leq t} \|\omega_\tau^z\| \leq M_z\}$ for any $z \in \{r, c\}$. As a result, we can follow the proof of Lemma B.3 in Zhang et al. (2018) to obtain that $\lim_{t \to \infty} \omega_{\perp,t}^z = 0$ a.s. for any $z \in \{r, c\}$.

**Convergence analysis.** We now analyze the asymptotic behavior of $\langle \omega_t^z \rangle$ to establish the convergence of critic parameters. With the average operator $\langle \cdot \rangle$, we can rewrite (22) as

$$
\begin{aligned}
\langle \omega_{t+1}^z \rangle &= \langle (W_t \otimes I)(\omega_t^z + \beta_{\omega,t} y_{t+1}^z) \rangle \\
&= \langle \omega_t^z \rangle + \beta_{\omega,t} \langle (W_t \otimes I)(y_{t+1}^z + \beta_{\omega,t}^{-1} \omega_{\perp,t}^z) \rangle \\
&= \langle \omega_t^z \rangle + \beta_{\omega,t} \mathbb{E}[\langle \delta_t^z \rangle \phi_t^z | \mathcal{F}_{t,1}] + \beta_{\omega,t} \xi_{t+1}^z,
\end{aligned}
\tag{27}
$$

where $\xi_{t+1}^z = \langle (W_t \otimes I)(y_{t+1}^z + \beta_{\omega,t}^{-1} \omega_{\perp,t}^z) \rangle - \mathbb{E}[\langle \delta_t^z \rangle \phi_t^z | \mathcal{F}_{t,1}]$, and

$$
\begin{aligned}
\langle \delta_t^r \rangle &= \bar{r}_{t+1} + \gamma((\phi_{t+1}^r)^T \langle \omega_t^r \rangle - \alpha \log(\pi_{\theta_t}(a_{t+1}|s_{t+1}))) - (\phi_t^r)^T \langle \omega_t^r \rangle, \\
\langle \delta_t^c \rangle &= \bar{c}_{t+1} + \gamma(\phi_{t+1}^c)^T \langle \omega_t^c \rangle - (\phi_t^c)^T \langle \omega_t^c \rangle.
\end{aligned}
\tag{28}
$$

Based on (28), we have that $\mathbb{E}[\langle \delta_t^z \rangle \phi_t^z | \mathcal{F}_{t,1}]$ is Lipschitz continuous in $\langle \omega_t^z \rangle$ for any $z \in \{r, c\}$. Then, $\{\xi_{t+1}^z\}_{t \geq 0}$ is a martingale difference sequence since

$$
\begin{aligned}
&\mathbb{E}\left[ \langle (W_t \otimes I)(y_{t+1}^z + \beta_{\omega,t}^{-1} \omega_{\perp,t}^z) \rangle | \mathcal{F}_{t,1} \right] \\
&= \mathbb{E}\left[ \langle y_{t+1}^z + \beta_{\omega,t}^{-1} \omega_{\perp,t}^z \rangle | \mathcal{F}_{t,1} \right] = \mathbb{E}\left[ \langle y_{t+1}^z \rangle | \mathcal{F}_{t,1} \right] = \mathbb{E}[\langle \delta_t^z \rangle \phi_t^z | \mathcal{F}_{t,1}],
\end{aligned}
\tag{29}
$$

where the first equality holds due to conditions (i) and (iii) in Assumption 2. Moreover, we have

$$
\mathbb{E}[\|\xi_{t+1}^z\|^2 | \mathcal{F}_{t,1}] \leq 2\mathbb{E}\left[ \|y_{t+1}^z + \beta_{\omega,t}^{-1} \omega_{\perp,t}^z\|_{G_t}^2 | \mathcal{F}_{t,1} \right] + 2\|\mathbb{E}[\langle \delta_t^z \rangle \phi_t^z | \mathcal{F}_{t,1}]\|^2,
\tag{30}
$$

where $G_t = \frac{1}{N^2} W_t^T \mathbb{1}\mathbb{1}^T W_t \otimes I$, and $\| \cdot \|_{G_t}$ is the Euclidean norm weighted by $G_t$. Recall that the term $\log(\pi_{\theta_t}(a_{t+1}|s_{t+1}))$ is uniformly bounded from Assumption 1. Thus, following similar steps in the proof of Theorem 4.6 in Zhang et al. (2018), for any $M_z > 0$, there exists $L_z < \infty$, such that

$$
\mathbb{E}[\|\xi_{t+1}^z\|^2 | \mathcal{F}_{t,1}] \leq L_z(1 + \|\langle \omega_t^z \rangle\|^2)
\tag{31}
$$

on the set $\{\sup_{t \to \infty} \|\omega_\tau^z\| \leq M_z\}$, $\forall z \in \{r, c\}$. For the critic recursion (27), the associated ODEs take the form

$$
\begin{aligned}
\langle \dot{\omega}^r \rangle &= (\Phi^r)^T D_\theta^{s,a} \left( \bar{R} + \gamma P_\theta(\Phi^r \langle \omega^r \rangle - \Omega_\theta) - \Phi^r \langle \omega^r \rangle \right), \\
\langle \dot{\omega}^c \rangle &= (\Phi^c)^T D_\theta^{s,a} \left( \bar{C} + \gamma P_\theta(\Phi^c \langle \omega^c \rangle) - \Phi^c \langle \omega^c \rangle \right),
\end{aligned}
\tag{32}
$$

which can be further rewritten as

$$
\begin{aligned}
\langle \dot{\omega}^r \rangle &= (\Phi^r)^T D_\theta^{s,a}(\gamma P_\theta - I)\Phi^r \langle \omega^r \rangle + (\Phi^r)^T D_\theta^{s,a}(\bar{R} - \gamma P_\theta \Omega_\theta), \\
\langle \dot{\omega}^c \rangle &= (\Phi^c)^T D_\theta^{s,a}(\gamma P_\theta - I)\Phi^c \langle \omega^c \rangle + (\Phi^c)^T D_\theta^{s,a} \bar{C}.
\end{aligned}
\tag{33}
$$

Note that $(\Phi^z)^T D_\theta^{s,a}(\gamma P_\theta - I)\Phi^z$, $\forall z \in \{r, c\}$ is negative definite based on Assumption 4 (Bhatnagar, 2010). As a result, the ODEs in (32) are globally asymptotically stable. Let $\omega_\theta^r$ and $\omega_\theta^c$ be the equilibria for $\langle \omega^r \rangle$ and $\langle \omega^c \rangle$, respectively, such that

$$
\begin{aligned}
(\Phi^r)^T D_\theta^{s,a} \left( \bar{R} + \gamma P_\theta(\Phi^r \omega_\theta^r - \Omega_\theta) - \Phi^r \omega_\theta^r \right) &= (\Phi^r)^T D_\theta^{s,a} \left( \mathcal{T}_\theta^r[\Phi^r \omega_\theta^r] - \Phi^r \omega_\theta^r \right) = 0, \\
(\Phi^c)^T D_\theta^{s,a} \left( \bar{C} + \gamma P_\theta \Phi^c \omega_\theta^c - \Phi^c \omega_\theta^c \right) &= (\Phi^c)^T D_\theta^{s,a} \left( \mathcal{T}_\theta^c[\Phi^c \omega_\theta^c] - \Phi^c \omega_\theta^c \right) = 0.
\end{aligned}
\tag{34}
$$

Note that the sequence $\{\omega_t^z\}_{t \geq 0}$, $\forall z \in \{r, c\}$ is bounded a.s. from Assumption 6, so is $\{\langle \omega_t^z \rangle\}_{t \geq 0}$. Based on Theorem D.2 in Zhang et al. (2018), it holds $\lim_{t \to \infty} \langle \omega_t^z \rangle = \omega_\theta^z$ a.s. for any $z \in \{r, c\}$. Recall that $\lim_{t \to \infty} \omega_{i,t}^z - \langle \omega_t^z \rangle = 0$ a.s. Therefore, we have $\lim_{t \to \infty} \omega_{i,t}^z = \omega_\theta^z$ a.s. with $\omega_\theta^z$ defined in (34) for any $i \in \mathcal{N}$ and $z \in \{r, c\}$, which completes the proof.

# E   PROOF OF THEOREM 4

Since the sequence $\{\omega_{i,t}^z\}_{t \geq 0}$ converges to $\omega_\theta^z$ at the faster timescale for any $i \in \mathcal{N}$ and $z \in \{r, c\}$, we consider the following actor update step:

$$
\tilde{\theta}_{i,t+1} = \theta_{i,t} + \beta_{\theta,t} N \eta_{i,t,\theta_t}^{\lambda_i} \psi_{i,t,\theta_t}, \quad \theta_{i,t+1} = \sum_{j \in \mathcal{N}_{i,t}} w_t(i,j) \tilde{\theta}_{j,t+1}.
\tag{35}
$$

We further rewrite (35) in a compact form, represented by

$$\theta_{t+1} = (W_t \otimes I)(\theta_t + \beta_{\theta,t} y_{t,\theta_t}^{\lambda}), \tag{36}$$

where $\theta_t = [\theta_{1,t}^T, \ldots, \theta_{N,t}^T]^T$ and $y_{t,\theta_t}^{\lambda} = [N\eta_{1,t,\theta_t}^{\lambda_1} \psi_{1,t,\theta_t}^T, \ldots, N\eta_{N,t,\theta_t}^{\lambda_N} \psi_{N,t,\theta_t}^T]^T$. We first show that the actor parameters of all agents will achieve consensus asymptotically.

**Consensus analysis.** With the average operator $\langle \cdot \rangle$ defined in (23), we have $\theta_t = \mathbb{1} \otimes \langle \theta_t \rangle + \theta_{\perp,t}$, where $\mathbb{1} \otimes \langle \theta_t \rangle$ is the agreement component in $\theta_t$, and $\theta_{\perp,t}$ is the disagreement component in $\theta_t$. Recall that $\eta_{i,t,\theta_t}^{\lambda_i} = (\phi_t^r)^T \omega_{\theta_t}^r - \alpha \log(\pi_{[\theta_{i,t}]}(a_t|s_t)) - \lambda_i(\phi_t^c)^T \omega_{\theta_t}^c$. Then, due to the boundedness of feature vectors based on Assumption 4, we have that $(\phi_t^z)^T \omega_{\theta_t}^z$ is uniformly bounded provided that $\omega_{\theta_t}^z$ is the MSPBE minimizer associated to $\theta_t$ for any $z \in \{r, c\}$. Note that $\log(\pi_{[\theta_{i,t}]}(a_t|s_t))$ is uniformly bounded based on Assumption 1. For any $s_t \in \mathcal{S}$ and $a_{i,t} \in \mathcal{A}_i$, we have that $\psi_{i,t,\theta_t} = \nabla_{\theta_i} \log(\pi_{i,\theta_{i,t}}(a_{i,t}|o_i(s_t)))$ is bounded as it is continuous over a compact set based on Assumptions 1 and 6. Note that both the state and action spaces are discrete. Hence, the boundedness of $y_{t,\theta_t}^{\lambda}$ can be ensured. Following similar lines as in the consensus analysis in Appendix D, we can obtain that $\lim_{t \to \infty} \theta_{\perp,t} = 0$ a.s., so that $\lim_{t \to \infty} \theta_{i,t} = \langle \theta_t \rangle$ a.s. for any $i \in \mathcal{N}$.

**Convergence analysis.** Let $\mathcal{F}_{t,2} = \sigma(\theta_\tau, W_\tau, \tau \leq t)$ be an increasing $\sigma$-algebra up to time $t$. Then, it holds

$$\langle \theta_{t+1} \rangle = \langle (W_t \otimes I)(\theta_t + \beta_{\theta,t} y_{t,\theta_t}^{\lambda}) \rangle = \langle \theta_t \rangle + \beta_{\theta,t} \langle (W_t \otimes I)(y_{t,\theta_t}^{\lambda} + \beta_{\theta,t}^{-1} \theta_{\perp,t}) \rangle$$

$$= \langle \theta_t \rangle + \beta_{\theta,t} \mathbb{E}_{s_t \sim d_{[\langle \theta_t \rangle]}, a_t \sim \pi_{[\langle \theta_t \rangle]}}[\langle y_{t,[\langle \theta_t \rangle]}^{\lambda} \rangle | \mathcal{F}_{t,2}] + \beta_{\theta,t} \zeta_{t+1,1}^{\theta} + \beta_{\theta,t} \zeta_{t+1,2}^{\theta}, \tag{37}$$

where $\zeta_{t+1,1}^{\theta}$ and $\zeta_{t+1,2}^{\theta}$ take the forms

$$\zeta_{t+1,1}^{\theta} = \langle (W_t \otimes I)(y_{t,\theta_t}^{\lambda} + \beta_{\theta,t}^{-1} \theta_{\perp,t}) \rangle - \mathbb{E}_{s_t \sim d_{[\langle \theta_t \rangle]}, a_t \sim \pi_{[\langle \theta_t \rangle]}}[\langle y_{t,\theta_t}^{\lambda} \rangle | \mathcal{F}_{t,2}],$$

$$\zeta_{t+1,2}^{\theta} = \mathbb{E}_{s_t \sim d_{[\langle \theta_t \rangle]}, a_t \sim \pi_{[\langle \theta_t \rangle]}}[\langle y_{t,\theta_t}^{\lambda} \rangle - \langle y_{t,[\langle \theta_t \rangle]}^{\lambda} \rangle | \mathcal{F}_{t,2}]. \tag{38}$$

Following similar lines to the proof of Lemma 1 in Bhatnagar (2010), we can prove that $\omega_\theta^r$ and $\omega_\theta^c$ are continuously differentiable in $\theta$. Hence, we have $\lim_{t \to \infty} \zeta_{t+1,2}^{\theta} = 0$ a.s. due to $\lim_{t \to \infty} \theta_t - [\langle \theta_t \rangle] = 0$ a.s. Following the similar analysis for $\xi_{t+1}^z$ in Appendix D, we have

$$\mathbb{E}[\langle (W_t \otimes I)(y_{t,\theta_t}^{\lambda} + \beta_{\theta,t}^{-1} \theta_{\perp,t}) \rangle | \mathcal{F}_{t,2}] = \mathbb{E}[\langle y_{t,\theta_t}^{\lambda} + \beta_{\theta,t}^{-1} \theta_{\perp,t} \rangle | \mathcal{F}_{t,1}] = \mathbb{E}[\langle y_{t,\theta_t}^{\lambda} \rangle | \mathcal{F}_{t,1}], \tag{39}$$

which shows that $\{\zeta_{t+1,1}^{\theta}\}_{t \geq 0}$ is a martingale difference sequence. Recall that $\{y_{t,\theta_t}^{\lambda}\}_{t \geq 0}$ is bounded. It holds that $\{\zeta_{t+1,1}^{\theta}\}_{t \geq 0}$ is bounded as well. Denote $M_t = \sum_{\tau=0}^{t} \beta_{\theta,\tau} \zeta_{\tau+1,1}^{\theta}$, such that $\{M_t\}_{t \geq 0}$ is a martingale sequence. Thus, it holds that $\sum_{t=0}^{\infty} \|M_{t+1} - M_t\|^2 = \sum_{t=1}^{\infty} \|\beta_{\theta,t} \zeta_{t+1,1}^{\theta}\|^2 < \infty$ by Assumption 5. Along the similar lines as the proof of Theorem 4.7 in Zhang et al. (2018), based on the Kushner-Clark lemma, $\langle \theta_t \rangle$ converges a.s. to a point in the set of asymptotically stable equilibria of the ODE

$$\dot{\theta} = \mathbb{E}_{s_t \sim d_{[\hat{\theta}]}, a_t \sim \pi_{[\hat{\theta}]}} \left[ \langle y_{t,[\hat{\theta}]}^{\lambda} \rangle \right] = \mathbb{E}_{s_t \sim d_{[\hat{\theta}]}, a_t \sim \pi_{[\hat{\theta}]}} \left[ \sum_{i \in \mathcal{N}} \eta_{i,t,[\hat{\theta}]}^{\lambda_i} \psi_{i,t,[\hat{\theta}]} \right]. \tag{40}$$

Recall that $\lim_{t \to \infty} \theta_{i,t} = \langle \theta_t \rangle$ a.s. for any $i \in \mathcal{N}$. Thus, each $\theta_{i,t}$ will converge to this point a.s., which completes the proof.

# F    PROOF OF THEOREM 5

Note that the dual variables update at the slowest timescale from Assumption 5. Based on Theorems 3 and 4, we consider a variant of (8) by respectively replacing $\omega_{i,t}^c$ and $\pi_{[\theta_{i,t}]}$ with $\omega_{[\hat{\theta}_{\bar{\lambda}_t}]}^c$ and $\pi_{[\hat{\theta}_{\bar{\lambda}_t}]}$, $\forall i \in \mathcal{N}$, which is expressed as

$$\tilde{\lambda}_{i,t+1} = \Gamma_\lambda \left[ \lambda_{i,t} - \beta_{\lambda,t} \mathbb{E}_{\tilde{s} \sim \rho, \tilde{a} \sim \pi_{[\hat{\theta}_{\bar{\lambda}_t}]}} [b - Q^c(\tilde{s}, \tilde{a}; \omega_{[\hat{\theta}_{\bar{\lambda}_t}]}^c)] \right], \quad \lambda_{i,t+1} = \sum_{j \in \mathcal{N}_{i,t}} w_t(i,j) \tilde{\lambda}_{j,t+1}. \tag{41}$$

Recall that $\bar{\lambda}_t = [\lambda_{1,t}, \ldots, \lambda_{N,t}]^T$. The compact form of (41) is

$$\bar{\lambda}_{t+1} = W_t \Gamma_\lambda \left[ \bar{\lambda}_t + \beta_{\lambda,t} \tilde{y}_{t,\bar{\lambda}_t} \right], \tag{42}$$

where $\tilde{y}_{t,\bar{\lambda}_t} = \mathbb{1} \otimes \mathbb{E}_{\tilde{s}\sim\rho, \tilde{a}\sim\pi_{[\hat{\theta}_{\bar{\lambda}_t}]}} [Q^c(\tilde{s}, \tilde{a}; \omega^c_{[\hat{\theta}_{\bar{\lambda}_t}]}) - b]$. By abuse of notation, the operator $\Gamma_\lambda$ here is used to project any element of a vector onto the set $[0, \lambda_{\max}]$.

**Consensus analysis.** With the average operator $\langle \cdot \rangle$ defined in (23), we have $\bar{\lambda}_t = \mathbb{1} \otimes \langle \bar{\lambda}_t \rangle + \bar{\lambda}_{\perp,t}$, where $\mathbb{1} \otimes \langle \bar{\lambda}_t \rangle$ is the agreement component in $\bar{\lambda}_t$, and $\bar{\lambda}_{\perp,t}$ is the disagreement component in $\bar{\lambda}_t$. Given the projection operator $\Gamma_\lambda$, we have $\Gamma_\lambda \left[ \bar{\lambda}_t + \beta_{\lambda,t} \tilde{y}_{t,\bar{\lambda}_t} \right] = \bar{\lambda}_t + \beta_{\lambda,t} \tilde{y}_{t,\bar{\lambda}_t} + \beta_{\lambda,t} \tilde{y}^p_{t,\bar{\lambda}_t}$, where $\beta_{\lambda,t} \tilde{y}^p_{t,\bar{\lambda}_t}$ is the vector of the shortest Euclidean length required to take $\bar{\lambda}_t + \beta_{\lambda,t} \tilde{y}_{t,\bar{\lambda}_t}$ back to the set $[0, \lambda_{\max}]^N$ if it doesn't belong to this set (Kushner & Yin, 1997). Recall that $\omega^c_{[\hat{\theta}_{\bar{\lambda}_t}]}$ is the MSPBE minimizer by Theorem 3, and the feature vectors are uniformly bounded by Assumption 4. Thus, the boundedness of $\tilde{y}_{t,\bar{\lambda}_t}$ can be guaranteed. It is worth noting that $\bar{\lambda}_t \in [0, \lambda_{\max}]^N$ due to that $W_t$ is row stochastic for any $t \geq 0$. As a consequence, we can obtain that $\tilde{y}^p_{t,\bar{\lambda}_t}$ is bounded since it is the projection term. Let $y_{t,\bar{\lambda}_t} = \tilde{y}_{t,\bar{\lambda}_t} + \tilde{y}^p_{t,\bar{\lambda}_t}$, then we can rewrite (42) as $\bar{\lambda}_{t+1} = W_t[\bar{\lambda}_t + \beta_{\lambda,t} y_{t,\bar{\lambda}_t}]$. Following similar lines as in the consensus analysis in Appendix D, we obtain that $\lim_{t\to\infty} \bar{\lambda}_{\perp,t} = 0$ a.s., i.e., $\lim_{t\to\infty} \lambda_{i,t} = \langle \bar{\lambda}_t \rangle$ a.s. for any $i \in \mathcal{N}$.

**Convergence analysis.** Let $\mathcal{F}_{t,3} = \sigma\left( \bar{\lambda}_\tau, W_\tau, \tau \leq t \right)$ be an increasing $\sigma$-algebra up to time $t$. Then, it holds

$$\langle \bar{\lambda}_{t+1} \rangle = \langle W_t \Gamma_\lambda [\bar{\lambda}_t + \beta_{\lambda,t} \tilde{y}_{t,\bar{\lambda}_t}] \rangle. \tag{43}$$

Let $\tilde{y}_{t,\bar{\lambda}_t} = [\tilde{y}_{1,t,\bar{\lambda}_t}, \ldots, \tilde{y}_{N,t,\bar{\lambda}_t}]^T$ and $\tilde{y}^p_{t,\bar{\lambda}_t} = [\tilde{y}^p_{1,t,\bar{\lambda}_t}, \ldots, \tilde{y}^p_{N,t,\bar{\lambda}_t}]^T$. In the following analysis, it is assumed that the projection term $\tilde{y}^p_{t,\bar{\lambda}_t}$ satisfies one of the following conditions at each time $t \geq 0$: (i) $\tilde{y}^p_{i,t,\bar{\lambda}_t} < 0, \forall i \in \mathcal{N}$. (ii) $\tilde{y}^p_{i,t,\bar{\lambda}_t} > 0, \forall i \in \mathcal{N}$. (iii) $\tilde{y}^p_{i,t,\bar{\lambda}_t} = 0, \forall i \in \mathcal{N}$. Note that this assumption is mild since $\tilde{y}_{1,t,\bar{\lambda}_t} = \cdots = \tilde{y}_{N,t,\bar{\lambda}_t}$, and all $\lambda_{i,t}$ will achieve consensus. Since $W_t$ is row stochastic by Assumption 2, we can further obtain that

$$\langle \bar{\lambda}_{t+1} \rangle = \langle W_t \Gamma_\lambda [\bar{\lambda}_t + \beta_{\lambda,t} \tilde{y}_{t,\bar{\lambda}_t}] \rangle = \Gamma_\lambda \left[ \langle W_t (\bar{\lambda}_t + \beta_{\lambda,t} \tilde{y}_{t,\bar{\lambda}_t}) \rangle \right], \tag{44}$$

based on which we have

$$\begin{aligned}
\langle \bar{\lambda}_{t+1} \rangle &= \Gamma_\lambda \left[ \langle \bar{\lambda}_t \rangle + \beta_{\lambda,t} \langle W_t (\tilde{y}_{t,\bar{\lambda}_t} + \beta_{\lambda,t}^{-1} \bar{\lambda}_{\perp,t}) \rangle \right] \\
&= \Gamma_\lambda \left[ \langle \bar{\lambda}_t \rangle + \beta_{\lambda,t} \langle \tilde{y}_{t,[\langle\bar{\lambda}_t\rangle]} \rangle + \beta_{\lambda,t} \zeta^\lambda_{t+1,1} + \beta_{\lambda,t} \zeta^\lambda_{t+1,2} \right],
\end{aligned} \tag{45}$$

where $\zeta^\lambda_{t+1,1}$ and $\zeta^\lambda_{t+2,2}$ take the forms

$$\begin{aligned}
\zeta^\lambda_{t+1,1} &= \langle W_t (\tilde{y}_{t,\bar{\lambda}_t} + \beta_{\lambda,t}^{-1} \bar{\lambda}_{\perp,t}) \rangle - \langle \tilde{y}_{t,\bar{\lambda}_t} \rangle, \\
\zeta^\lambda_{t+2,2} &= \langle \tilde{y}_{t,\bar{\lambda}_t} \rangle - \langle \tilde{y}_{t,[\langle\bar{\lambda}_t\rangle]} \rangle.
\end{aligned} \tag{46}$$

Recall that both $\pi_\theta$ and $\omega^c_\theta$ are continuous in $\theta$. We can obtain that $\tilde{y}_{t,\bar{\lambda}_t}$ is continuous in $\bar{\lambda}_t$ based on Assumption 7. As a result, it holds $\lim_{t\to\infty} \zeta^\lambda_{t+1,2} = 0$ a.s. due to that $\lim_{t\to\infty} \bar{\lambda}_t - [\langle\bar{\lambda}_t\rangle] = 0$ a.s. Based on Assumption 2, we can obtain that

$$\mathbb{E}[\langle W_t (\tilde{y}_{t,\bar{\lambda}_t} + \beta_{\lambda,t}^{-1} \bar{\lambda}_{\perp,t}) \rangle | \mathcal{F}_{t,3}] = \mathbb{E}[\langle \tilde{y}_{t,\bar{\lambda}_t} \rangle | \mathcal{F}_{t,3}] = \langle \tilde{y}_{t,\bar{\lambda}_t} \rangle, \tag{47}$$

which shows that $\{\zeta^\lambda_{t+1,1}\}_{t\geq 0}$ is a martingale difference sequence. Following the similar analysis for $\zeta^\theta_{t+1,1}$ in Appendix E, we can obtain that $\{\zeta^\lambda_{t+1,1}\}_{t\geq 0}$ is bounded. As a consequence, $\langle \bar{\lambda}_t \rangle$ converges a.s. to a point in the set of asymptotically stable equilibria of the ODE

$$\dot{\lambda} = \hat{\Gamma}_\lambda \left[ \langle \tilde{y}_{t,[\lambda]} \rangle \right] = \hat{\Gamma}_\lambda \left[ \mathbb{E}_{\tilde{s}\sim\rho, \tilde{a}\sim\pi_{[\hat{\theta}_{[\lambda]}]}} [Q^c(\tilde{s}, \tilde{a}; \omega^c_{[\hat{\theta}_{[\lambda]}]}) - b] \right]. \tag{48}$$

Recall that $\lim_{t\to\infty} \lambda_{i,t} = \langle \bar{\lambda}_t \rangle$ a.s. for any $i \in \mathcal{N}$. Thus, each $\lambda_{i,t}$ will converge to this point a.s., which completes the proof.

## G  PROOFS OF PROPOSITIONS 1 AND 2

**Proof of Proposition 1.** Recall that $\hat{\Lambda} = \{\lambda : \hat{\Gamma}_\lambda[\mathbb{E}_{\tilde{s}\sim\rho,\tilde{a}\sim\pi_{[\hat{\theta}_{[\lambda]}]}}[Q^c(\tilde{s},\tilde{a};\omega^c_{[\hat{\theta}_{[\lambda]}]}) - b]] = 0, \lambda \in [0,\lambda_{\max})\}$. Using a contradiction argument, suppose that $\mathbb{E}_{\tilde{s}\sim\rho,\tilde{a}\sim\pi_{[\hat{\theta}_{[\lambda^*]}]}}[Q^c(\tilde{s},\tilde{a};\omega^c_{[\hat{\theta}_{[\lambda^*]}]})] > b$ for some $\lambda^* \in \hat{\Lambda}$. Then, we have

$$
\begin{aligned}
&\hat{\Gamma}_\lambda[\mathbb{E}_{\tilde{s}\sim\rho,\tilde{a}\sim\pi_{[\hat{\theta}_{[\lambda^*]}]}}[Q^c(\tilde{s},\tilde{a};\omega^c_{[\hat{\theta}_{[\lambda^*]}]}) - b]] \\
&= \lim_{\eta\to 0^+} \frac{\Gamma_\lambda\left[\lambda^* + \eta(\mathbb{E}_{\tilde{s}\sim\rho,\tilde{a}\sim\pi_{[\hat{\theta}_{[\lambda^*]}]}}[Q^c(\tilde{s},\tilde{a};\omega^c_{[\hat{\theta}_{[\lambda^*]}]}) - b])\right] - \lambda^*}{\eta} \\
&= \lim_{\eta\to 0^+} \frac{\lambda^* + \eta(\mathbb{E}_{\tilde{s}\sim\rho,\tilde{a}\sim\pi_{[\hat{\theta}_{[\lambda^*]}]}}[Q^c(\tilde{s},\tilde{a};\omega^c_{[\hat{\theta}_{[\lambda^*]}]}) - b]) - \lambda^*}{\eta} \\
&= \mathbb{E}_{\tilde{s}\sim\rho,\tilde{a}\sim\pi_{[\hat{\theta}_{[\lambda^*]}]}}[Q^c(\tilde{s},\tilde{a};\omega^c_{[\hat{\theta}_{[\lambda^*]}]}) - b] > 0,
\end{aligned}
\tag{49}
$$

where the second equality holds for sufficiently small $\eta > 0$ since $\lambda^* \in [0,\lambda_{\max})$, and the inequality further indicates that $\lambda^* \notin \hat{\Lambda}$. Hence, we prove the proposition by contradiction.

**Proof of Proposition 2.** Given the condition $\mathbb{E}_{\tilde{s}\sim\rho,\tilde{a}\sim\pi_{[\hat{\theta}_{[\lambda^*]}]}}\left[Q^c(\tilde{s},\tilde{a};\omega^c_{[\hat{\theta}_{[\lambda^*]}]})\right] < b$, it is apparent that the limit of (49) equals zero when $\lambda^* = 0$. Moreover, for the case $\lambda^* \in (0,\lambda_{\max}]$, we can follow the similar lines as the proof of Proposition 1 to obtain that the limit of (49) is negative under the condition $\mathbb{E}_{\tilde{s}\sim\rho,\tilde{a}\sim\pi_{[\hat{\theta}_{[\lambda^*]}]}}\left[Q^c(\tilde{s},\tilde{a};\omega^c_{[\hat{\theta}_{[\lambda^*]}]})\right] < b$. Note that this result contradicts the fact that $\lambda^* \in \Lambda$, such that the proposition is proved.

## H  SIMULATION RESULTS OF THE THEORETICAL DECENTRALIZED ALGORITHM IN SECTION 3

We now empirically demonstrate the effectiveness of our theoretical decentralized primal-dual actor-critic algorithm (6)-(8) on a toy experiment.

**The environment.** Consider an environment with $N$ agents, where $N$ is even. The local state of each agent $i \in \mathcal{N}$ is $s_i = \cos(\frac{i-1}{N-1}\pi)$. All agents have the same local action space $\mathcal{A}_i = \{0,1\}$. After the joint action $a = (a_1,\ldots,a_N)$ is executed in the environment, the global state $s = (s_1,\ldots,s_N)$ transitions to a terminal state. The agents share the same reward function, defined by

$$
R_i(s,a) = \sum_{k=1}^{N/2}\mathbb{I}_{\{a_k=1\}} - \sum_{k=N/2+1}^{N}\mathbb{I}_{\{a_k=1\}}, \quad \forall i \in \mathcal{N},
\tag{50}
$$

where $\mathbb{I}$ is the indicator function. It is obvious that the largest reward $\frac{N}{2}$ can be obtained if $a_i = 1$ for $i \leq \frac{N}{2}$, and $a_i = 0$ for $i > \frac{N}{2}$. Define the local observation of each agent $i \in \mathcal{N}$ as $o_i = (s_i, s_1,\ldots,s_N)$. Then, this environment can be cast as a homogeneous MG (Chen et al., 2022). In our setting, the agents further have the same cost function, defined by

$$
C_i(s,a) = \sum_{k=1}^{N}\mathbb{I}_{\{a_k=1\}}, \quad \forall i \in \mathcal{N}.
\tag{51}
$$

Note that the cost function (51) satisfies condition (ii) in Definition 1. Hence, our environment can be cast as a homogeneous constrained MG. In our experiment, $N$ is set as 10, and the threshold $b$ in (1) is set as 4. In this case, the optimal reward is 4, i.e., one of the first five agents should choose the action 0.

**Experimental setting.** Our algorithm is compared against two baselines. The first baseline is the centralized version of our algorithm, which employs a centralized unit to directly train the parameter $\hat{\theta}$ for the joint policy $\pi_{[\hat{\theta}]}$ based on Theorem 1. The second baseline is taken from Hu et al. (2024), which is decentralized, and also employs the entropy regularization mechanism for policy learning.

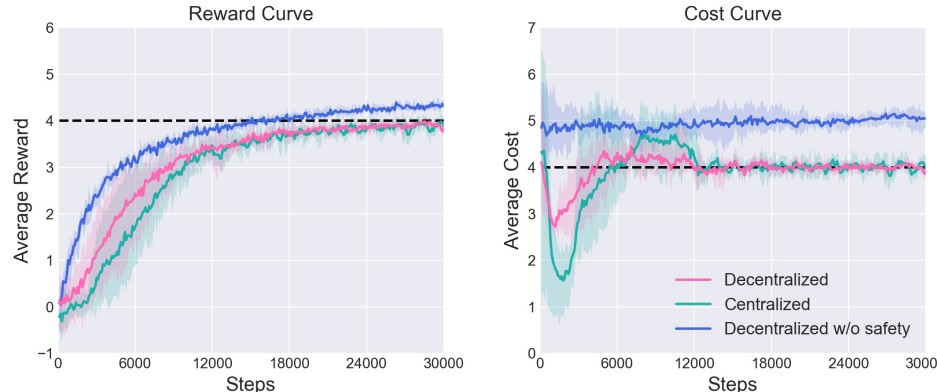

Figure 3: Learning curves of three algorithms on the toy experiment, where the dotted line in the left subfigure indicates the optimal reward for our problem, and the dotted line in the right subfigure indicates the threshold $b$.

Nevertheless, this baseline only aims to maximize the reward, without taking the constraint into consideration. Following Chen et al. (2022), the feature vector of the linear critics is designed as $\phi^z(s, a) = \text{concat}\left[\{s_i, \text{one\_hot}(a_i)\}_{i \in \mathcal{N}}\right], \forall z \in \{r, c\}$, which is shared by all the three algorithms. The actor is parameterized as a liner function, which generates the distribution over two actions by the softmax function. At each iteration $t$, we generate $\mathcal{G}_t$ by randomly placing 18 communication links among agents. Following Zhang et al. (2018), the element $w_t(i, j)$ in the weight matrix $W_t$ is determined by

$$w_t(i, j) = \frac{1}{1 + \max\{d_t(i), d_t(j)\}}, \quad \forall (i, j) \in \mathcal{E}_t,$$

$$w_t(i, i) = 1 - \sum_{j \in \mathcal{N}_{i,t}} w_t(i, j), \quad \forall i \in \mathcal{N}, \tag{52}$$

where $d_t(i) = |\mathcal{N}_{i,t}|$ is the degree of agent $i$. The expectation in (8) is estimated using Monte Carlo samples. In addition, the Adam optimizer is employed to update parameters with adaptive learning rates, and $\alpha$ is set as 0.01.

**Results.** The learning curves of three algorithms are represented in Fig. 3. The blue curves indicate that the decentralized algorithm from Hu et al. (2024) achieves the highest reward after training but fails to meet the safety constraint. This can be attributed to the fact that the algorithm only tries to maximize the reward without considering the constraint. On the contrary, the other two algorithms can achieve the optimal reward of the constrained problem while satisfying the safety constraint after training, which is attributed to the primal-dual training method. The simulation results empirically verify the convergence conclusion in Section 4.

# I APPENDIX FOR THE PRACTICAL ALGORITHM IN SECTION 5

## I.1 ANALYSIS ON THE LOSS FUNCTION (14)

Let $G_\theta^{\lambda_i}(s_t, a_t) = \alpha \log(\pi_\theta(a_t|s_t)) - Q^r(s_t, a_t; \omega_i^r) + \lambda_i Q^c(s_t, a_t; \omega_i^c)$, then we have

$$\nabla_{\theta_i} J_\pi(\theta_i) = \nabla_{\theta_i} \left[ \mathbb{E}_{s_t \sim \mathcal{B}_i, a_t \sim \pi_\theta}[G_\theta^{\lambda_i}(s_t, a_t)] \right]$$

$$= \mathbb{E}_{s_t \sim \mathcal{B}_i} \left[ \sum_{a_t \in \mathcal{A}} \nabla_{\theta_i}(G_\theta^{\lambda_i}(s_t, a_t)\pi_\theta(a_t|s_t)) \right]$$

$$= \mathbb{E}_{s_t \sim \mathcal{B}_i} \left[ \sum_{a_t \in \mathcal{A}} (\nabla_{\theta_i} G_\theta^{\lambda_i}(s_t, a_t))\pi_\theta(a_t|s_t) + G_\theta^{\lambda_i}(s_t, a_t)(\nabla_{\theta_i}\pi_\theta(a_t|s_t)) \right]. \tag{53}$$

Realize that

$$\sum_{a_t \in \mathcal{A}} (\nabla_{\theta_i} G_\theta^{\lambda_i}(s_t, a_t)) \pi_\theta(a_t|s_t) = \alpha \sum_{a_t \in \mathcal{A}} (\nabla_{\theta_i} \log(\pi_{\theta_i}(a_{i,t}|o_{i,t}))) \pi_\theta(a_t|s_t) = 0, \qquad (54)$$

where the last equality holds due to (B.3) in Zhang et al. (2018). As a result, we have

$$\nabla_{\theta_i} J_\pi(\theta_i) = \mathbb{E}_{s_t \sim \mathcal{B}_i} \left[ \sum_{a_t \in \mathcal{A}} G_\theta^{\lambda_i}(s_t, a_t)(\nabla_{\theta_i} \pi_\theta(a_t|s_t)) \right]$$

$$= \mathbb{E}_{s_t \sim \mathcal{B}_i} \left[ \sum_{a_t \in \mathcal{A}} G_\theta^{\lambda_i}(s_t, a_t)\pi_\theta(a_t|s_t)(\nabla_{\theta_i} \log(\pi_{\theta_i}(a_{i,t}|o_{i,t}))) \right]$$

$$= \mathbb{E}_{s_t \sim \mathcal{B}_i, a_t \sim \pi_\theta} \left[ G_\theta^{\lambda_i}(s_t, a_t)\nabla_{\theta_i} \log(\pi_{\theta_i}(a_{i,t}|o_{i,t})) \right]. \qquad (55)$$

Note that the only difference between $-\nabla_{\theta_i} J_\pi(\theta_i)$ and the local actor parameter update direction in (7) is the state distribution. Realize that it is a common trick in DRL to use experiences stored in the replay buffer for updating current policy parameters (Lillicrap et al., 2016).

## I.2 PSEUDOCODE

We now present the pseudocode of our decentralized algorithm DPDAC-ER in Algorithm 1.

---

**Algorithm 1** Decentralized Primal-Dual Actor-Critic with Entropy Regularization (DPDAC-ER)

---

1: Initialize $\{\omega_i^r\}_{i=1}^N$, $\{\omega_i^c\}_{i=1}^N$, $\{\theta_i\}_{i=1}^N$, $\{\lambda_i\}_{i=1}^N$, $\{\alpha_i\}_{i=1}^N$.
2: Set $\mathcal{B}_i \leftarrow \emptyset$, $\bar{\omega}_i^z \leftarrow \omega_i^z$ for each $i \in \mathcal{N}$ and $z \in \{r, c\}$.
3: **for** each iteration **do**
4:     **for** each environment step **do**
5:         Each agent $i \in \mathcal{N}$ samples $a_{i,t} \sim \pi_{i,\theta_i}(o_{i,t})$.
6:         Update state $s_{t+1} \sim P(\cdot|s_t, a_t)$.
7:         Each agent $i \in \mathcal{N}$ obtains $s_{t+1}$, $r_{i,t+1}$ and $c_{i,t+1}$.
8:         Each agent $i \in \mathcal{N}$ updates its replay buffer $\mathcal{B}_i \leftarrow \mathcal{B}_i \cup \{(s_t, a_t, r_{i,t+1}, c_{i,t+1}, s_{t+1})\}$.
9:     **end for**
10:     **for** each gradient step **do**
11:         **for** each agent $i \in \mathcal{N}$ **do**
12:             Sample a batch of data $\mathcal{D}_i$ from $\mathcal{B}_i$.
13:             $\omega_i^z \leftarrow \omega_i^z - l_Q \hat{\nabla}_{\omega_i^z} J_Q^z(\omega_i^z)$ for each $z \in \{r, c\}$.
14:             $\theta_i \leftarrow \theta_i - l_\pi \hat{\nabla}_{\theta_i} J_\pi(\theta_i)$.
15:             $\lambda_i \leftarrow (\lambda_i - l_\lambda \hat{\nabla}_{\lambda_i} J_D(\lambda_i))_+$.
16:             $\alpha_i \leftarrow (\alpha_i - l_\alpha \hat{\nabla}_{\alpha_i} J(\alpha_i))_+$.
17:             $\bar{\omega}_i^z \leftarrow \tau \omega_i^z + (1-\tau)\bar{\omega}_i^z$ for each $z \in \{r, c\}$.
18:         **end for**
19:     **end for**
20:     **for** each consensus step **do**
21:         **for** each agent $i \in \mathcal{N}$ **do**
22:             $\tilde{\omega}_i^z \leftarrow \sum_{j \in \mathcal{N}_{i,m}} w_m(i,j)\omega_j^z$ for each $z \in \{r, c\}$.
23:             $\tilde{\theta}_i \leftarrow \sum_{j \in \mathcal{N}_{i,m}} w_m(i,j)\theta_j$.
24:             $\tilde{\lambda}_i \leftarrow \sum_{j \in \mathcal{N}_{i,m}} w_m(i,j)\lambda_j$.
25:         **end for**
26:         **for** each agent $i \in \mathcal{N}$ **do**
27:             $\omega_i^z \leftarrow \tilde{\omega}_i^z$ for each $z \in \{r, c\}$.
28:             $\theta_i \leftarrow \tilde{\theta}_j$.
29:             $\lambda_i \leftarrow \tilde{\lambda}_i$.
30:         **end for**
31:     **end for**
32: **end for**

---

# J EXPERIMENTS

## J.1 SIMULATION ENVIRONMENTS

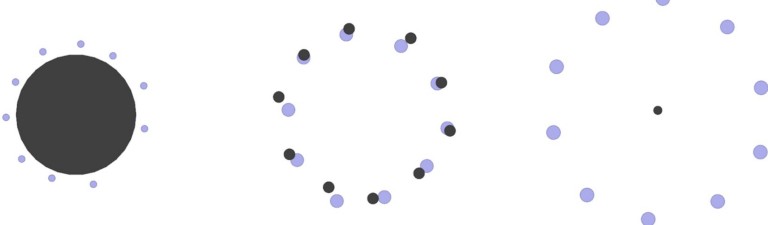

Figure 4: Snapshots of three task environments at the end of episodes based on the policies trained by DPDAC-ER, where every purple disk represents an agent and every black disk represents a landmark. **Left**: the Aggregation task. **Middle**: the Swapping task. **Right**: the Formation task.

Note that our algorithm DPDAC-ER can learn desirable safe policies. In the Aggregation task, all agents distribute themselves along the circumference of the hazardous area to maximize the reward while ensuring safety. In the Swapping task, all agents move slowly to the target positions to comply with the velocity saturation constraint. In the Formation task, all agents form a circular formation with a center near the landmark. Videos demonstrating the coordination performance of the learned safe and unsafe policies are available at https://github.com/ICLR2025anonymous/DPDAC-ER/.

## J.2 IMPLEMENTATION DETAILS

Table 1: Comparison of DPDAC-ER with four baselines.

|  | Safe | Decentralized | Entropy | Number of Critics (Reward) | Number of Critics (Cost) | Number of Actors |
|---|---|---|---|---|---|---|
| DPDAC-ER (ours) | ✓ | ✓ | ✓ | $N$ | $N$ | $N$ |
| DPDAC (ours) | ✓ | ✓ |  | $N$ | $N$ | $N$ |
| MASAC-Lag | ✓ |  | ✓ | 1 | 1 | 1 |
| DAC-ER |  | ✓ | ✓ | $N$ | $N$ | $N$ |
| MASAC |  |  | ✓ | 1 | 1 | $N$ |

The differences between our algorithm DPDAC-ER and four baselines are summarized in Table 1. For fair comparison, all algorithms share the same critic and actor NN structures. Realize that the dimension of the global state increases rapidly as the number of agents grows, leading to severe scalability issues for MARL algorithms that use conventional NNs such as multi-layer perceptrons (MLPs) (Liu et al., 2019). To this end, we employ the graph neural network (GNN)-based critic with the same hyperparameters as in Hu et al. (2024) as separate critics for both the reward and the cost in our algorithm. In this case, it is more convenient for each agent $i \in \mathcal{N}$ to directly store $o_t = (o_{1,t}, \ldots, o_{N,t})$ rather than $s_t$ in its replay buffer $\mathcal{B}_i$, which can be achieved due to the observability of the global state and the permutation preserving property in homogeneous constrained MGs. Note that for environments like Safe Multi-Agent MuJoCo (Gu et al., 2023), we can still store $s_t$ in $\mathcal{B}_i$ and use MLPs to construct critic NNs, as the observation of an individual agent is the concatenation of the state of robots and a one-hot vector. In our implementation, the double Q-learning trick is used for the critic NN for rewards. To handle continuous action spaces, we design the actor NN as a Gaussian policy, which includes two hidden layers and two linear layers[2]. Here, each hidden layer consists of 128 neurons, with ReLU as its activation function. For all the decentralized algorithms, the communication graph $\mathcal{G}_t$ is generated through randomly placing 18 communication links among

---

[2]https://github.com/pranz24/pytorch-soft-actor-critic

agents. Note that there exist at most 45 links when all-to-all communication is allowed. Therefore, the communication condition considered in our experiment is relatively mild. All agents perform the gradient and consensus update steps once after each episode ends, where the weight matrix is determined by (52). The hyperparameters of DPDAC-ER in three tasks are given in Table 2. Note that $\bar{b}$ is the expected (undiscounted) cost threshold, such that the threshold $b$ in our problem (3) is approximated by $\frac{(1-\gamma^L)\bar{b}}{(1-\gamma)L}$ (Ray et al., 2019; Yang et al., 2021), where $L$ denotes the episode length. In addition, the hyperparameters for the baselines are also finely tuned. In our experimental results, the reported return and cumulative cost are undiscounted.

Table 2: Hyperparameter Settings of DPDAC-ER.

| Hyperparameters | Aggregation | Swapping | Formation |
|---|---|---|---|
| Number of training episodes $M$ | 20000 | 20000 | 20000 |
| Episode length $L$ | 25 | 25 | 25 |
| Undiscounted cost threshold $\bar{b}$ | 0 | 0 | 2 |
| Learning rate $\lambda_Q$ | 5e-3 | 5e-3 | 5e-3 |
| Learning rate $\lambda_\pi$ | 3e-4 | 4e-4 | 4e-4 |
| Learning rate $\lambda_\lambda$ | 3e-5 | 2e-5 | 3e-5 |
| Learning rate $\lambda_\alpha$ | 3e-4 | 4e-4 | 4e-4 |
| Discount factor $\gamma$ | 0.95 | 0.95 | 0.95 |
| Target smoothing coefficient $\tau$ | 0.01 | 0.01 | 0.01 |
| Initial value of $\log \lambda_i$ | 0.5 | 0.5 | 0.5 |
| Initial value of $\log \alpha_i$ | 0 | 0 | 0 |
| Target entropy $\mathcal{H}_0$ | -2 | -2 | -2 |
| Buffer size $|\mathcal{B}_i|$ | 500000 | 500000 | 500000 |
| Batch size $|\mathcal{D}_i|$ | 256 | 256 | 256 |

## J.3 ABLATION ON THE COMMUNICATION NETWORK

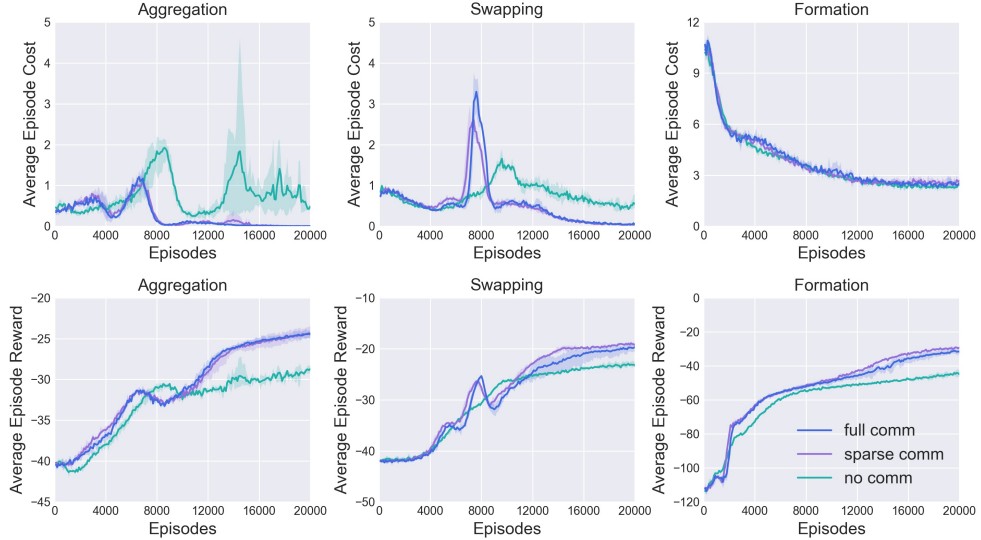

Figure 5: Learning curves of DPDAC-ER in three different communication scenarios.

## J.4 ABLATION ON THE CONSTRAINT

In the Swapping task, DPDAC-ER is evaluated under three undiscounted cost thresholds: 0, 5, and 10. In the Formation task, DPDAC-ER is evaluated under three undiscounted cost thresholds: 0, 2, and 5. The learning curves of DPDAC-ER under different cost thresholds can be found in Fig. 6.

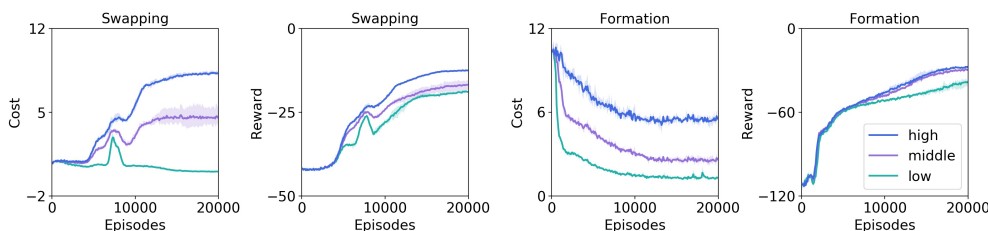

Figure 6: Learning curves of DPDAC-ER under different cost thresholds.

## J.5 ABLATION ON THE LOCAL OBSERVATION

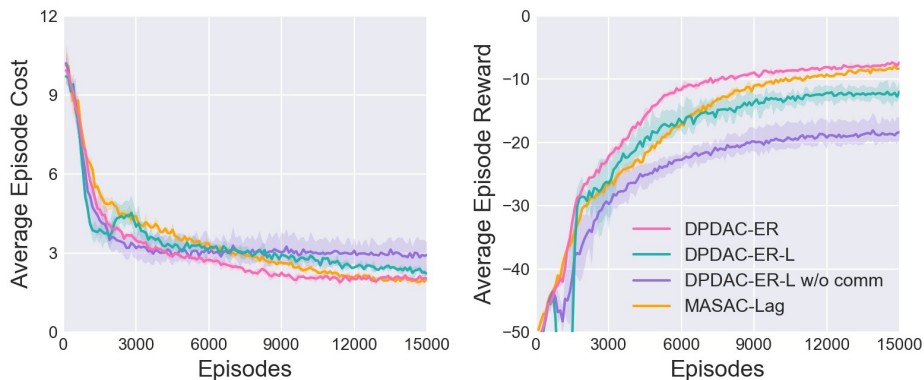

Figure 7: Learning curves of four algorithms in the simplified Formation task.

Under the local observation setting, the global state information is not available to agents, and each agent $i \in \mathcal{N}$ can only obtain its local observation $o_{i,t}$ during training. To deal with this problem, existing decentralized algorithms (Qu et al., 2022; Ying et al., 2023b; Chen et al., 2022) leverage the communication network $\mathcal{G}_t$ to enable agents to exchange local state-action (observation-action) pairs with their neighbors. Denote $\mathcal{C}_{i,t} \subseteq \mathcal{N}_{i,t}$ as the set containing all neighbors for observation-action communication for each agent $i \in \mathcal{N}$. Then, the critic NNs in our algorithm can be reformulated as $Q^z(o_{i,t}^{co}, a_{i,t}^{co}), \forall z \in \{r, c\}$, where $o_{i,t}^{co} = \{o_{k,t}\}_{k \in \bar{\mathcal{C}}_{i,t}}$ and $a_{i,t}^{co} = \{a_{k,t}\}_{k \in \bar{\mathcal{C}}_{i,t}}$ with $\bar{\mathcal{C}}_{i,t} = \{i\} \cup \mathcal{C}_{i,t}$, and the replay buffer takes the form $\mathcal{B}_i = \{(o_{i,t}^{co}, a_{i,t}^{co}, r_{i,t+1}, c_{i,t+1}, o_{i,t+1}^{co})\}_t$. Based on the notations above, the loss function (13) for the critics is modified as

$$J_Q^z(\omega_i^z) = \mathbb{E}_{(o_{i,t}^{co}, a_{i,t}^{co}, r_{i,t+1}, c_{i,t+1}, o_{i,t+1}^{co}) \sim \mathcal{B}_i} \left[ (Q^z(o_{i,t}^{co}, a_{i,t}^{co}; \omega_i^z) - y_i^z)^2 \right], \quad \forall z \in \{r, c\}, \qquad (56)$$

where $y_i^r = r_{i,t+1} + \gamma(Q^r(o_{i,t+1}^{co}, a_{i,t+1}^{co}; \bar{\omega}_i^r) - N\alpha \log(\pi_{i,\theta_i}(a_{i,t+1}|o_{i,t+1})))$ and $y_i^c = c_{i,t+1} + \gamma Q^c(o_{i,t+1}^{co}, a_{i,t+1}^{co}; \bar{\omega}_i^c)$. The loss function (14) for the actor is modified as

$$J_\pi(\theta_i) = \mathbb{E}\left[ \alpha \log(\pi_{i,\theta_i}(a_{i,t}|o_{i,t})) - Q^r(o_{i,t}^{co}, a_{i,t}^{co}; \omega_i^r) + \lambda_i Q^c(o_{i,t}^{co}, a_{i,t}^{co}; \omega_i^c) \right], \qquad (57)$$

where $o_{i,t}^{co} \sim \mathcal{B}_i$ and $a_{k,t} \sim \pi_{k,\theta_k}(\cdot|o_{k,t})$ for any $k \in \bar{\mathcal{C}}_{i,t}$. Recall that each agent $i \in \mathcal{N}$ estimates $\pi_{k,\theta_k}$, where $k \in \mathcal{C}_{i,t}$, using its local policy $\pi_{i,\theta_i}$ based on the policy consensus. Finally, the loss function (15) for the dual variable is modified as

$$J_D(\lambda_i) = \mathbb{E}_{(o_{i,t}^{co}, a_{i,t}^{co}) \sim \mathcal{B}_i} \left[ \lambda_i(b - Q^c(o_{i,t}^{co}, a_{i,t}^{co}; \omega_i^c)) \right]. \qquad (58)$$

Denote DPDAC-ER-L as the modified version of DPDAC-ER under the local observation setting, which uses the loss functions (56)-(58).

In our ablation experiment, we demonstrate the effectiveness of DPDAC-ER-L on a simplified version of the Formation task. In this task, there exist 5 agents, and the mean position of all agents is not available to any agent. In our implementation of DPDAC-ER-L, we set $\mathcal{C}_{i,t} = \emptyset$, such that each agent only uses its local observation-action pair $(o_{i,t}, a_{i,t})$ to estimate global action-value functions.

We model the critic NN $Q^z(o_{i,t}, a_{i,t})$ as an MLP, $\forall z \in \{r, c\}$, which contains two hidden layers with 128 neurons per layer. We compare DPDAC-ER-L with DPDAC-ER and MASAC-Lag, which can use global state information during training. In addition, we evaluate DPDAC-ER-L in the no communication scenario. In this case, the agents learn to update their policies independently. The learning curves can be found in Fig. 7. There is no doubt that DPDAC-ER and MASAC-Lag have the best learning performance, which converge to safe policies with the highest return. This result attributes to that the global state information is available to all agents in these algorithms. It can also be observed that when no communication is available, DPDAC-ER-L performs significantly worse than the other algorithms in terms of reward, which also has the worst learning stability. On the contrary, when sparse communication is available among agents, the performance of DPDAC-ER-L improves significantly in terms of both reward and cost, demonstrating the effectiveness of our algorithm. We leave the evaluation of DPDAC-ER-L with $\mathcal{C}_{i,t} \neq \emptyset$ on more challenging safe MARL tasks as our future work.

### J.6 ADDITIONAL EXPERIMENTS

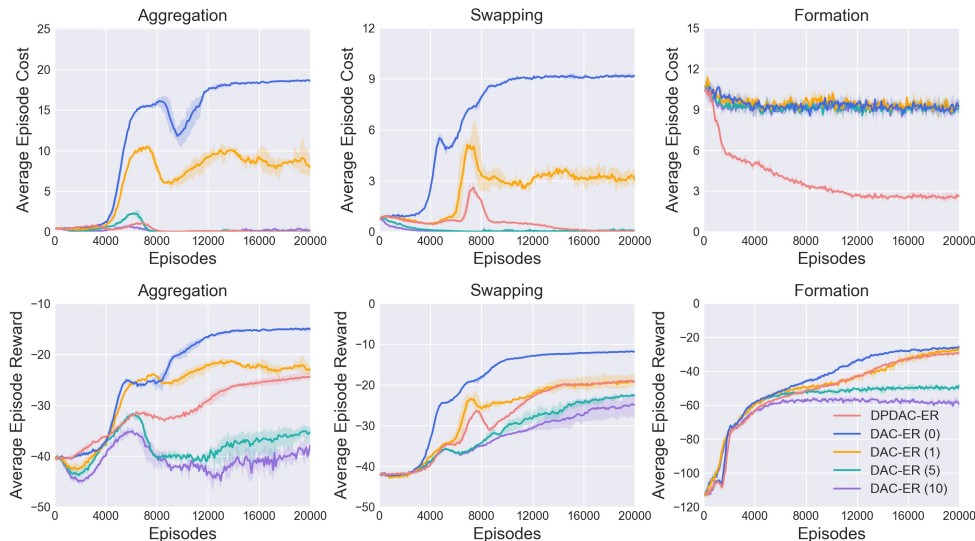

Figure 8: Learning curves of DPDAC-ER and DAC-ER under different $w$.

To demonstrate the effectiveness of our algorithm in balancing reward maximization and constraint satisfaction, a variant of DAC-ER is considered as a new baseline which uses the reward shaping mechanism to deal with the constraints. Specifically, we replace the reward $r_{i,t}$ with $r_{i,t} - wc_{i,t}$ in DAC-ER, where $w \geq 0$. It can be found in Fig. 8 that safe policies can be learned by DAC-ER with large $w$ in the Aggregation task and the Swapping task. However, the safe policies learned by DAC-ER yield significantly lower rewards compared to those learned by our algorithm. Note that DAC-ER fails to learn safe policies in the Formation task across different values of $w$, with its learning curve for rewards appearing to diverge when $w = 10$. These simulation results demonstrate the importance of deal with constraints independently in safe RL.

Finally, we create a customized 3D environment for the Formation task to further evaluate the sample efficiency of our algorithm. It can be found in Fig. 9 that the learning performance of our algorithm is preserved when the dimension of the space increases. However, the learning performance of DP-DAC declines significantly, which shows the importance of incorporating the entropy regularization mechanism in our algorithm.

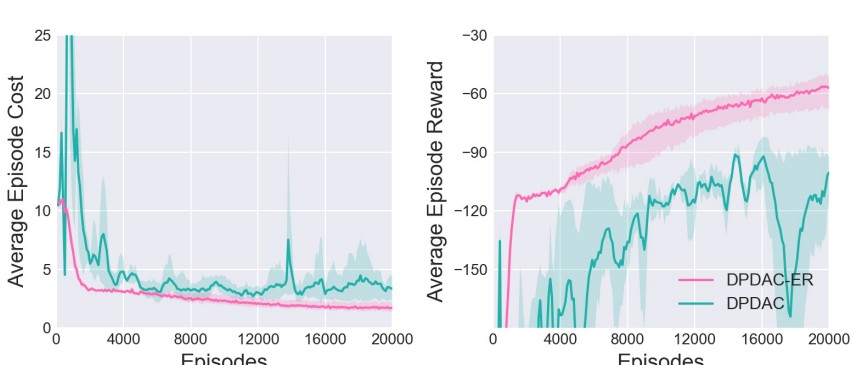

Figure 9: Learning curves of DPDAC-ER and DPDAC in a customized 3D environment.

