# OpenReview forum: "Decentralized primal-dual actor-critic with entropy regularization for safe multi-agent reinforcement learning"
_ICLR.cc/2025/Conference — Submitted to ICLR 2025_

### Official Review · Reviewer_bAPC · 2024-11-01

**Soundness:** 3
**Presentation:** 3
**Contribution:** 2
**Rating:** 3
**Confidence:** 3

**Summary:**

This paper considers a safe multi-agent MDP scenario and proposes a decentralized MARL algorithm to solve this problem. It assumes a homogenous constrained MG setup, which results in the fact that independent policies are enough. The paper then considers a communication graph structure over which agents communicate. The paper shows the convergence of the proposed approach.

**Strengths:**

1. The paper considers a really important problem of constrained multi-agent RL problem.

2. The paper proposes algorithms that have a provable convergence guarantee.

3. The paper shows convergence under different communication graph structures.

**Weaknesses:**

1. What is the main motivation behind considering joint reward and the joint cost as permutation preserving? Can the paper provide some real examples that satisfy the above assumption?

2. The paper considers the constraint where the expected cumulative cost across the agents is below a certain threshold. In general, what types of problems have such constraints?

3. Assumption 5 states that it is a three-time scale approximation which might be very slow in convergence.

4. The paper did not bound the sub-optimality gap and even considered a stronger assumption for convergence. For example, Theorem 4 states that for a given dual-variable, the actor policy converges. Then, the paper assumes Assumptions 7 which states that the convergent points of the actor policy parameters are continuous. It seems that it is a very strong assumption.
Nevertheless, it did not prove the sub-optimality gap. It may converge to some local solution.

Post Rebuttal Comments:

While constrained MG setup has not been studied much, a few works are along this line. The paper did not provide any solid motivations behind this work. This paper considers a special case where the agents are homogenous. The authors did not answer on a satisfactory level why the results obtained in CMDP setup will not transfer in this setting. While a decentralized algorithm has been proposed, again, this is based on the unconstrained case. There is no satisfactory answer what are the challenges in extending those results to the constrained case apart from the standard ones (like tuning dual variable or so). Hence, even though the results might be novel, the results do not have much impact at this present form to the literature.

**Questions:**

1. What are the technical novelties? In particular, unconstrained decentralized MARL is well-studied, distributed algorithms under communication graph structure are also well-studied. Finally, the two-time scale algorithms are also well-studied. This paper seems to combine all these approaches. Hence, the technical novelties are not at all clear.

2. From the simulation results, it seems that the performance is similar (or, negligible improvement) compared to MASAC-Lag approach. In fact, for Formation scenario MASAC-Lag has a smaller cost and a better reward. What is the value of the constraint ($b$) for the simulation?

---

> ### Author Response · Authors · 2024-11-22
> **Response Part I**
>
> We thank the reviewer for appreciating our work. We hope our responses below provide further clarity.
>
> **[Response to Weakness 1]** Thanks for your questions. This property is motivated by the homogeneity of agents. For instance, consider a permutation $M$ that swaps a pair of agents $(i, j)$, where agent $i$ ($j$) executes the local action $a_j$ ($a_i$) based on the local state $s_j$ ($s_i$) after permutation. Due to the homogeneity of agents, agent $i$ ($j$) obtains the local reward $r_j$ ($r_i$) and the local cost $c_j$ ($c_i$) after permutation, which results in the permutation preserving property for the joint reward and the joint cost. In the revised manuscript, an illustrative example of our homogeneous constrained MG model is given in Appendix A.1, and some practical examples are given in Appendix A.2.
>
> **[Response to Weakness 2]** Thanks for your question. For the indicator cost function satisfying $c_{i,t+1} = 1$ when agent $i$ is unsafe and $c_{i,t+1} = 0$ otherwise, our constraint $E_{s \sim \rho, \pi}[\sum_{t=0}^\infty \gamma^t \bar c_{t+1}] \le 0$ (by setting $b = 0$) is satisfied when $E_{s \sim \rho, \pi}[\sum_{t=0}^\infty \gamma^t c_{i,t+1}] \le 0$ for any $i \in N$. These individual constraints are consistent with those considered in existing safe MARL works [R1], [R2]. Note that the indicator cost function setting is considered in the Aggregation task and the Swapping task in our experiments. Additionally, several physical constraints considering the joint state of agents can be incorporated into our problem setting. For example, in the economic dispatch problem for smart grids, the power outputs of all generators should meet the requirement of the total power demand. In collaborative transportation tasks, the total upward force exerted by all agents must exceed the weight of the cargo.
>
> **[Response to Weakness 3]** Thanks for your comment. Based on primal-dual optimization theory, the policy update in primal-dual-based safe RL algorithms typically requires a relatively larger learning rate, while the dual variable update requires a smaller one, which could lead to a slower convergence speed. However, combining this approach with deep RL architectures seems to mitigate efficiency concerns, as various primal-dual-based safe RL algorithms have demonstrated effectiveness within an acceptable number of episodes [R1], [R3], [R4]. Note that existing decentralized safe MARL algorithms also use the primal-dual method [R2], [R5]. Realize that these algorithms face severe learning efficiency problems in continuous safe MARL tasks (see the related work section in the manuscript for details). Our practical algorithm is developed based on the deep RL architectures, which preserves the decentralized training setting. Note that the entropy regularization mechanism is employed here considering exploration issues in continuous spaces. Therefore, while our theoretical algorithm may have a slow convergence rate, we enhance its efficiency through practical modifications, which takes a step towards the application of decentralized safe MARL algorithms in challenging safe multi-agent tasks.

---

> > ### Author Response · Authors · 2024-11-22
> > **Response Part II**
> >
> > **[Response to Weakness 4]** Thanks for your careful reading and comments. By regarding the multi-agent system as a large but singe-agent system, our safe MARL problem (1) can be regarded as a safe RL problem. Therefore, we can directly obtain the results on the sub-optimality gap in our problem using existing theoretical results for primal-dual-based safe RL methods [R6]. Specifically, for general policy classes, there is no duality gap in our problem provided that the joint reward is uniformly bounded and there exists a strictly feasible policy for (1) (see Theorem 3 in [R6]). For parameterized policies, the duality gap can be bounded by a function which is linear with the approximation error $\varepsilon$ of the parameterization (see Theorem 4 in [R6]). We now explain the theoretical results in this work. Based on multi-timescale stochastic approximation analysis [R7] (Lemma 1, page 66), [R8] (page 18, arXiv version), the update of the dual variables in our algorithm can be viewed to track the ODE $\dot \lambda_i(t) \equiv 0$ due to the condition $\beta_{\lambda,t} = o(\beta_{\theta,t})$ in Assumption 5, which allows us to consider the dual variables as constants when analyzing the faster update of policy parameters. In Theorem 4, we show that the update of policy parameters tracks the ODE (11) in the manuscript. Realize that the RHS of (11) is continuous in $\bar \lambda = [\lambda_1, \dots, \lambda_N]$. Thus, Assumption 7 is made, which is not strict. Note that this assumption is also employed in [R9]. Finally, as an actor-critic algorithm, our algorithm is not guaranteed to converge to a global maximum [R6], [R8], [R9]. Note that existing decentralized safe MARL algorithms also converge to stationary points, which are not globally optimal. Nevertheless, the simulation results in our (toy and multi-robot coordination) experiments show that our algorithm can learn desirable safe policies.
> >
> > **[Response to Question 1]** Thanks for your variable questions. In the decentralized training setting, it becomes a huge challenge to update the dual variables for the agents. Recall the centralized constraint $E_{s \sim \rho}[V_{\theta}^c(s)] = E_{s \sim \rho, a \sim \pi_{\theta}} [Q_{\theta}^c(s,a)] \le b$ in our problem. Even though the global value function $Q_{\theta}^c$ can be estimated using existing decentralized MARL algorithms, the gradient of the Lagrangian $L(\theta, \lambda)$ w.r.t $\lambda$ cannot be calculated by any individual agent as it has no access to the joint policy due to the decentralized setting. In this work, one of the technical novelties is the design of the decentralized dual variable update step (formula (8)), where each agent maintains a local estimate of the dual variable, denoted by $\lambda_i$. Theoretical convergence of the dual variables are then established (see Theorem 5). Compared with existing decentralized primal-dual methods [R2], [R5], the constraint satisfaction of the learned policy is further analyzed (see Propositions 1 and 2). Moreover, to improve learning efficiency of our theoretical algorithm, we further develop practical loss functions under the deep RL training setting, which enables our algorithm to deal with challenging continuous safe MARL tasks (see Section 5 and Appendix I for details).
> >
> > **[Response to Question 2]** Thanks for your question. MASAC-Lag is a centralized-training algorithm which assumes to have access to all information (reward, cost, state, action) for policy learning. On the contrary, in our decentralized algorithm, each agent only has access to the local reward and cost information, which uses much less information than the centralizer in MASAC-Lag for policy learning. The simulation results show that our algorithm has comparable performance to MASAC-Lag, which demonstrates the effectiveness of our algorithm. In addition, the values of $b$ in all the three tasks as well as other experimental settings can be found in Appendix J.2. Note that we set $b=0$ for both the Aggregation and Swapping tasks, as agents can remain safe throughout each episode once the safe policy is learned and applied. In the Formation task, the cost is designed as the distance between the agents’ mean position and the landmark’s position, which is shared by all agents. Let $\bar b$ be the threshold for the undiscounted accumulated distance over the course of the episode. Then, we set $b = \frac{(1 - \gamma^L)\bar b}{(1- \gamma)L}$, where $L$ is the episode length. Note that the episodic cost cannot be zero, even when the safest policy is applied, as the distance is positive at the beginning of each episode.

---

> > > ### Author Response · Authors · 2024-11-22
> > > **Response Part III**
> > >
> > > References
> > >
> > > [R1] Gu S, Kuba J G, Chen Y, et al. Safe multi-agent reinforcement learning for multi-robot control[J]. Artificial Intelligence, 2023, 319: 103905.
> > >
> > > [R2] Lu S, Zhang K, Chen T, et al. Decentralized policy gradient descent ascent for safe multi-agent reinforcement learning[C]//Proceedings of the AAAI conference on artificial intelligence. 2021, 35(10): 8767-8775.
> > >
> > > [R3] Ray A, Achiam J, Amodei D. Benchmarking safe exploration in deep reinforcement learning[J]. arXiv preprint arXiv:1910.01708, 2019, 7(1): 2.
> > >
> > > [R4] Yang Q, Simão T D, Tindemans S H, et al. WCSAC: Worst-case soft actor critic for safety-constrained reinforcement learning[C]//Proceedings of the AAAI Conference on Artificial Intelligence. 2021, 35(12): 10639-10646.
> > >
> > > [R5] Ying D, Zhang Y, Ding Y, et al. Scalable primal-dual actor-critic method for safe multi-agent rl with general utilities[J]. Advances in Neural Information Processing Systems, 2024, 36.
> > >
> > > [R6] Paternain S, Calvo-Fullana M, Chamon L F O, et al. Safe policies for reinforcement learning via primal-dual methods[J]. IEEE Transactions on Automatic Control, 2022, 68(3): 1321-1336.
> > >
> > > [R7] Borkar V S, Borkar V S. Stochastic approximation: a dynamical systems viewpoint[M]. Cambridge: Cambridge University Press, 2008.
> > >
> > > [R8] Zhang K, Yang Z, Liu H, et al. Fully decentralized multi-agent reinforcement learning with networked agents[C]//International conference on machine learning. PMLR, 2018: 5872-5881.
> > >
> > > [R9] Bhatnagar S. An actor–critic algorithm with function approximation for discounted cost constrained Markov decision processes[J]. Systems \& Control Letters, 2010, 59(12): 760-766.

---

> > > > ### Comment · Reviewer_bAPC · 2024-11-25
> > > >
> > > > I thank the authors for their efforts. I believe that the technical contributions are limited since one can apply the results of CMDP and the decentralized MARL algorithms (as the agents are homogenous) to obtain convergence. Hence, I would keep the score.

---

> > > > > ### Author Response · Authors · 2024-11-25
> > > > > **Response**
> > > > >
> > > > > Dear Reviewer,
> > > > >
> > > > > Thanks for your feedback. We sincerely appreciate the time and effort you have dedicated to reviewing our paper, and we fully respect your comments on this work.
> > > > >
> > > > > It would be greatly appreciated if the reviewer could take a look at the sentences below:
> > > > >
> > > > > **Contribution to Safe MARL problems:** We develop a new constrained MG model, and provide theoretical justifications for the policy sharing mechanism in safe MARL for the first time.
> > > > >
> > > > > **Contribution to theoretical Safe MARL algorithms:** We design a decentralized primal-dual actor-critic algorithm, which solves the decentralized dual variable update problem and provide its theoretical convergence for the first time.
> > > > >
> > > > > **Contribution to practical Safe MARL algorithms:** We propose a practical off-policy decentralized safe MARL algorithm for the first time, whose effectiveness is demonstrated through comparison with solid centralized-training baselines (supported by more experiments in the revised manuscript).
> > > > >
> > > > > Our contribution to safe MARL is built upon all three parts mentioned. We really hope you could change your mind. Thanks!
> > > > >
> > > > > Yours faithfully,
> > > > >
> > > > > Authors

---

### Official Review · Reviewer_rR6P · 2024-11-03

**Soundness:** 2
**Presentation:** 2
**Contribution:** 2
**Rating:** 5
**Confidence:** 3

**Summary:**

This paper studies the decentralized, safe multi-agent reinforcement learning problem. The paper proposes a homogeneous, constrained Markov game framework to formulate the problem and an off-policy algorithm to solve it. The paper also proves the convergence of the on-policy version of the proposed algorithm under several assumptions. The proposed approach is validated in a safety-aware continuous swarm robot task in a simulation environment.

**Strengths:**

1. The study of the safety problem of multi-agent reinforcement learning is well-motivated.
2. The paper is well-structured.

**Weaknesses:**

1. Problem Formulation isn't sound to me: In 2.2 Problem formulation, the authors assume the availability of the global state and agents' joint actions, which violates the problem setting of the decentralized multi-agent. In decentralized multi-agent RL, the local agent should only have access to the information of its local neighborhood.

2. The novelty of the paper is limited. 1) problem formulation 2.2, policy gradient derivation, and the convergence proof seem like straightforward extensions of the work [1][4]. 2) The primal-dual method for multi-agent safe RL is also studied in [2-3].

3. The proposed approach is only demonstrated in simple robotic swarming tasks in simulation, which is a bit insufficient for validating the effectiveness of the proposed algorithm. From the attached video, I also noticed that in the safe swapping scenario, there were two instances where collisions occurred between agents. It seems the learned policy is not safe enough.

4. I personally feel like the paper is a bit hard to follow due to the heavy notations.

[1] Zhang, Kaiqing, et al. "Fully decentralized multi-agent reinforcement learning with networked agents." International conference on machine learning. PMLR, 2018.

[2] Ying, Donghao, et al. "Scalable primal-dual actor-critic method for safe multi-agent rl with general utilities." Advances in Neural Information Processing Systems 36 (2024)

[3] Paternain, Santiago, et al. "Safe policies for reinforcement learning via primal-dual methods." IEEE Transactions on Automatic Control 68.3 (2022): 1321-1336.

[4] Zhang, Kaiqing, Zhuoran Yang, and Tamer Başar. "Multi-agent reinforcement learning: A selective overview of theories and algorithms." Handbook of reinforcement learning and control (2021): 321-384.

**Questions:**

How do you set the value of b in the loss function equation (15)?

---

> ### Author Response · Authors · 2024-11-22
> **Response Part I**
>
> Thank you very much for the time and efforts spent on reviewing our paper. We have tried to address all your concerns. Please refer to the following responses for details.
>
> **[Response to Weakness 1]** We totally agree with your comments on decentralized MARL, where agents partially observe the environment (state, action, reward and cost) and share information with neighbors through a sparse network. Nevertheless, it still remains a huge challenge to design decentralized algorithms considering both theoretical convergence and learning efficiency under this setting. Fortunately, recent decentralized MARL algorithms have empirically demonstrated potential in addressing practical MARL tasks within a fully decentralized setting [R1], [R2], which are introduced in the related work section. Currently, there are two categories of decentralized algorithms: communication for parameter information and communication for state-action information. In the former category, the availability of the global state and action is usually assumed due to the coupled state transition function. Each agent updates its local policy based on the individual reward and cost information as well as the parameter information from its neighbors under practical communication conditions. In the latter category, the assumption on the availability of the global state and action is removed under an additional assumption on the spatial correlation decay property of weakly coupled MDPs. In these algorithms, each agent needs local state and action pairs from its $k$-hop neighbors when updating its local policy. However, existing algorithms in this category are on-policy, which only consider discrete spaces. Note that the spatial correlation decay property does not hold in many multi-robot coordination task. Furthermore, the communication condition is not robust, which cannot deal with link failures. Our work is more pertinent to the former, such that the global observability of the state and the joint action is assumed when providing convergence analysis on our theoretical algorithm. Nevertheless, it is also discussed in the manuscript that the mechanism of sharing local state-action pairs can be effectively combined into our practical algorithm to enable decentralized training without the global state and action information. An experiment considering limited environment information is also conducted in this work. More details can be found in Appendix J.5.
>
> **[Response to Weakness 2]** Thanks for your comments. In this work, we solve a safe MARL problem under the decentralized training setting, where agents aim to maximize the team-average return and the joint policy’s entropy, while satisfying safety constraints associated to the cumulative team-average cost. Compared with existing decentralized MARL works [R3], [R4], the constraint in this work is centralized, which requires the joint policy and cost function information. Further explanation of our constraint can be found in our response to Weakness 2 for Reviewer bAPC. Even though existing decentralized MARL methods enable any individual agent here to approximate the dual function in this work in a decentralized manner. It remains a huge challenge to design decentralized update formulas for the dual variable $\lambda$ associated to the centralized constraint. One of the technical novelties of this work is the decentralized update design for the dual variable, where theoretical convergence is established based on multi-timescale stochastic approximation theory. Please note that existing decentralized safe MARL works stopped at the theoretical analysis stage [R3], [R4]. Our work further proposes a practical decentralized algorithm which is closely related to our theoretical algorithm. Compared with [R3], [R4], our algorithm can deal with continuous spaces and achieve off-policy training to improve sample efficiency. As mentioned in the response to Weakness 1, our algorithm can also be adapted under the local observability setting. In addition, our work is different from [R5]. In fact, the zero duality gap was discussed in [R5], where the theoretical results also hold for our problem (see the response to Weakness 4 for Reviewer bAPC for more details). However, the algorithm proposed in [R5] is not specific (see Algorithm 2 in [R5]), and no theoretical convergence analysis was provided.

---

> > ### Author Response · Authors · 2024-11-22
> > **Response Part II**
> >
> > **[Response to Weakness 3]** Thanks for your comment. It has already been stated in the manuscript that the cost function in the Swapping task is designed for the velocity saturation constraint. When agent $i$ collides with other agents, it will receive a smaller reward. More details can be found in the code that we have shared. You can find that the agents trained with our algorithm move much slower than those trained with other algorithms that don't take safety into consideration, such that the velocity saturation constraint can be met. In fact, achieving collision avoidance for large-scale multi-robot systems using model-free RL remains a challenging issue. Representative works in this field carefully design the reward function to enable agents to plan collision-free trajectories [R6], [R7]. Although these works do not require a system model of the agents, they typically assume the agents follow a first-order dynamics model. Note that the collision avoidance problem for multi-agent systems with complex system dynamics have also been studied in the control community [R8]. However, the system model of agents should be known in advance. Recall that the agents in MPE follow a discrete second-order dynamics model, which requires significant effort to design safe MARL algorithms capable of addressing the collision avoidance problem even under centralized training. In the revised manuscript, our proposed algorithm is further compared with a reward-shaping-based baseline, and is evaluated on a customized 3D environment (more details can be found in Appendix J.6). We would like to thank the reviewer again for this valuable comment. Our future work will focus on addressing more challenging safe MARL tasks using our algorithm.
> >
> > **[Response to Weakness 4]** Thank you for your feedback. We have made every effort to eliminate redundant notations in the manuscript to enhance readability. However, as with existing works in this field [R3], [R4], employing numerous notations is unavoidable for the theoretical analysis of our algorithm.
> >
> > **[Response to Question 1]** The determination of the threshold $b$ has already been discussed in Appendix J.2. Specifically, let $L$ be the episode length, and $\bar b$ be the real-world cost threshold, which can either be an upper bound on the number of constraint violations or have a specific physical meaning for certain tasks. Then, the threshold $b$ in our problem (3) is approximated by $b = \frac{(1 - \gamma^L)\bar b}{(1- \gamma)L}$. In the Aggregation task and the Swapping task in our experiment, we consider indicator cost functions satisfying $c_{i,t+1} = 1$ when agent $i$ is unsafe, and $c_{i,t+1} = 0$ otherwise. We set $\bar b = 0$ to ensure safety of all agents throughout the entire trajectory, such that $b = 0$. In the Formation task, a team cost is shared by all agents, which is the distance between the agents' mean position and a target position. Note that this distance cannot be zero at the beginning of the episode. We set $\bar b = 2$ in our experiment based on our environment setting, such that $b = \frac{(1 - \gamma^L)2}{(1- \gamma)L}$.
> >
> > **References**
> >
> > [R1] Chen D, Li Y, Zhang Q. Communication-Efficient Actor-Critic Methods for Homogeneous Markov Games[C]//The Tenth International Conference on Learning Representations (ICLR 2022). 2022.
> >
> > [R2] Hu Y, Fu J, Wen G, et al. Distributed entropy-regularized multi-agent reinforcement learning with policy consensus[J]. Automatica, 2024, 164: 111652.
> >
> > [R3] Ying D, Zhang Y, Ding Y, et al. Scalable primal-dual actor-critic method for safe multi-agent rl with general utilities[J]. Advances in Neural Information Processing Systems, 2024, 36.
> >
> > [R4] Lu S, Zhang K, Chen T, et al. Decentralized policy gradient descent ascent for safe multi-agent reinforcement learning[C]//Proceedings of the AAAI conference on artificial intelligence. 2021, 35(10): 8767-8775.
> >
> > [R5] Paternain S, Calvo-Fullana M, Chamon L F O, et al. Safe policies for reinforcement learning via primal-dual methods[J]. IEEE Transactions on Automatic Control, 2022, 68(3): 1321-1336.
> >
> > [R6] Long P, Fan T, Liao X, et al. Towards optimally decentralized multi-robot collision avoidance via deep reinforcement learning[C]//2018 IEEE international conference on robotics and automation (ICRA). IEEE, 2018: 6252-6259.
> >
> > [R7] Han R, Chen S, Wang S, et al. Reinforcement learned distributed multi-robot navigation with reciprocal velocity obstacle shaped rewards[J]. IEEE Robotics and Automation Letters, 2022, 7(3): 5896-5903.
> >
> > [R8] Hu Y, Fu J, Wen G. Decentralized robust collision-avoidance for cooperative multirobot systems: A gaussian process-based control barrier function approach[J]. IEEE Transactions on Control of Network Systems, 2022, 10(2): 706-717.

---

> ### Comment · Reviewer_rR6P · 2024-11-24
> **Response to authors**
>
> Dear authors,
>
> Thanks for your response; I appreciate your efforts in answering the questions and preparing more experiments. The revised paper has improved a lot and resolved most of my questions. I have some extra questions:
>
> 1. Although the availability of the global state and action in decentralized multiagent reinforcement learning is assumed in many published literature [R1, R2], I'm still unconvinced by this setting. Can you give practical examples to explain the motivation of this setting? Since the global state and action are available to all agents, why can't local reward and policy information be available globally, too?
> 2. At line 184, the policy set $\mathcal{\Theta} $ is compact and context. What is the intuition behind it?
> 3. At line 215, what is the intuition behind $K_z  \ll |S| \times |A|$?
>
> Thanks for reading this, and I am looking forward to your response.
>
> Best
>
> Reviewer
>
> [R1] Zhang, Kaiqing, et al. "Fully decentralized multi-agent reinforcement learning with networked agents." International conference on machine learning. PMLR, 2018.
>
> [R2] Ye, Lintao, et al. "Resilient Multi-Agent Reinforcement Learning With Function Approximation." IEEE Transactions on Automatic Control (2024).

---

> > ### Author Response · Authors · 2024-11-25
> > **Response**
> >
> > Thanks for your positive feedback. Please refer to the following responses to your additional questions.
> >
> > **[Response to Question 1]** Thanks for your very valuable question. Consider an environment where $N$ players (viewed as agents) collaborate online to accomplish a game task. The global state and action of the game environment are clearly visible on the screen and accessible to all agents. Each agent's reward may vary depending on the role assigned to it and whether it successfully completes its subtask. Note that both the local policy and the reward information of an individual agent cannot be obtained by the other agents if they not communicate through the internet. We can also take the data collection task (shown Appendix A.2 in the revised manuscript) as another example. The UAV can obtain the remaining data information of the Internet of Things devices through some fixed wireless communication channels (the communication cost is low). In addition, it can determine the relative positions of the other UAVs based on sensors (such as Radar). Combined with the GPS signal it obtained, each UAV can know the global information in this task (the actions can be estimated based on positions). The reward of each UAV differs, which depends on the amount of data collected by the UAV in practice, which cannot be observed by the other UAVs. Again, the local policy is maintained and used to control an individual UAV, which cannot be obtained by the other UAVs.
> >
> > **[Response to Question 2]** Thanks for your very valuable question. We have removed the assumption on the convexity of the joint policy parameter space $\Theta$, as our original problem is not a convex problem. On the contrary, the compactness of this set is a comment assumption for completeness [R1], [R2], which is practical by considering this set as a ball with an extremely large but finite radius.
> >
> > **[Response to Question 3]** Thanks for your question. When the state and actor spaces are large (but finite), the scalability issue of using the $Q$ table occurs, where there are $|S| \times |A|$ elements in the $Q$ table. This inspires us to use approximators to replace $Q$ table when learning policies [R3], where the dimension of the parameters of the approximators is significantly smaller than the total number of the elements in the $Q$ table. Note that when the spaces are continuous, we are unable to build such a $Q$ table as the elements of both the state and the action are infinite.
> >
> >
> > Thank you again for the time and efforts spent on reviewing our paper.
> >
> > Yours faithfully,
> >
> > Authors
> >
> > **References**
> >
> > [R1] Shalabh Bhatnagar, Richard S. Sutton, Mohammad Ghavamzadeh, and Mark Lee. Natural actor-critic algorithms. Automatica, 45(11):2471-2482, 2009.
> >
> > [R2] Zhang K, Yang Z, Liu H, et al. Fully decentralized multi-agent reinforcement learning with networked agents [C]. International conference on machine learning. PMLR, 2018: 5872-5881.
> >
> > [R3] Richard S. Sutton and Andrew G. Barto. Reinforcement learning: An introduction. MIT press, 2018.

---

> > > ### Comment · Reviewer_rR6P · 2024-11-25
> > > **Response**
> > >
> > > Dear authors,
> > >
> > > Thanks for your timely response, which resolves my questions. I increased the score to 5.
> > >
> > > Still, I think the work's limitation is its lack of novelties and strong demonstrations on public benchmarks. At last, I suggest the authors demonstrate the proposed method in a more sophisticated environment, for instance [R1][R2] in the future to make the paper stronger.
> > >
> > > Best
> > >
> > > Reviewer
> > >
> > > [R1] https://sites.google.com/view/aij-safe-marl/
> > >
> > > [R2] https://github.com/PKU-Alignment/safety-gymnasium

---

> > > > ### Author Response · Authors · 2024-11-25
> > > > **Response**
> > > >
> > > > Dear Reviewer,
> > > >
> > > > So many thanks for increasing your score. The development of decentralized safe MARL is still at the early stage, and so many existing theoretical works only evaluate their performance on toy experiments. In this work, beyond the theoretical contributions, we have indeed made many efforts to narrow the learning performance gap between decentralized safe MARL algorithms and centralized-training safe MARL algorithms.
> > > >
> > > > We sincerely appreciate the time and effort you have dedicated to reviewing our paper.
> > > >
> > > > Yours faithfully,
> > > >
> > > > Authors

---

### Official Review · Reviewer_93J6 · 2024-11-03

**Soundness:** 3
**Presentation:** 2
**Contribution:** 1
**Rating:** 3
**Confidence:** 3

**Summary:**

This paper considers the decentralized safe multi-agent reinforcement learning (MARL) problem in homogeneous multi-agent systems. To address the problem, the paper formalizes it as a homogeneous constrained Markov game and proposes an on-policy decentralized primal-dual actor-critic algorithm. Theoretical results are provided on the convergence of the algorithm, and experimental results show the efficacy of the algorithm in 3 multi-particle environments.

**Strengths:**

The writing of the paper is clear. The paper considers an important problem. Both theoretical results and empirical results are provided. 3 ablation studies are provided.

**Weaknesses:**

1. The problem setting only considers finite state and action spaces.

2. The proposed algorithm is standard. Seems that the only difference compared with MAPPO-Lagrangian [1] is that the update steps are decentralized by the parameter aggregation.

3. There is some ambiguity in the presentation of the algorithm (see Questions for details).

4. The convergence analysis depends on many assumptions, some of which are not realistic. For example, 1) the critic is approximated by linear critic functions; 2) the stepsize of the critic is much larger than the stepsize of the actor, which is also much larger than the stepsize of $\lambda$. None of these assumptions can be realized in a practical algorithm.

5. The proposed practical algorithm violates the assumptions of the convergence analysis.

6. The experiments are only performed in Multi-agent Particle environments, which is easy. It is difficult to demonstrate the performance of the proposed algorithm only in these easy environments.

7. Two of the baselines considered in the experiments do not consider the safety constraint at all. Therefore, these two baselines cannot justify the claims of the paper. I suggest the authors add costs to the reward functions such that the rewards become $r' = r - w c$, where $w$ is the weight, and then choose different $w$ to create several baselines.

**Minors:**

1. The weight matrix $C_t = [c_t(i, j)]_{N\times N}$ uses the same notation as the cost. I suggest using another notation to avoid confusion.


**References:**

[1] Gu, Shangding, et al. "Safe multi-agent reinforcement learning for multi-robot control." Artificial Intelligence 319 (2023): 103905.

**Questions:**

1. The paper assumes the observation function is bijective, and therefore the agents have global observation. Then why is the observation function needed? What is the observation function in the environments in the experiments?

2. How is the weight matrix $C_t$ updated? Is it learned or handcrafted?

3. Why is the practical algorithm in Section 5 an off-policy one while all the previous analyses are based on the on-policy one?

4. Is the theoretical analysis valid on the practical algorithm?

5. The environments in the experiments have continuous action space. Does this violate Assumption 1?

6. What is the cost threshold for the Formation environment?

---

> ### Author Response · Authors · 2024-11-22
> **Response Part I**
>
> Thank you very much for your appreciation of our work. Many thanks for your time and efforts in reviewing this paper. Please refer to the following responses for details.
>
> **[Response to Weakness 1]** Thanks for your comment. The setting of finite state and action spaces is standard, which forms the basis for many significant theoretical results on actor-critic methods [R1], [R2], [R3]. One of the theoretical contribution of this work is a convergent decentralized primal-dual actor-critic algorithm for safe MARL, where we leverage existing analytical analysis tools to obtain our convergence results. Hence, we also assume that the state and action spaces are discrete. However, based on several practical modifications, our algorithm can be directly applied to continuous safe MARL tasks (see Section 5 for details). Please note that the famous RL algorithm SAC [R4] was proposed similarly. In [R4], a soft policy iteration algorithm was proposed at first, which is convergent under the tabular setting. Then, SAC was proposed whose loss functions are developed based on soft policy iteration.
>
> **[Response to Weakness 2]** Thanks for your interesting comment. Both our practical algorithm and MAPPO-Lag use the primal-dual optimization framework, and are designed to solve continuous safe MARL problems under the fully cooperative setting. However, the essential difference between our algorithm and MAPPO-Lag is the training fashion. As mentioned in the related work section, similar to existing MARL algorithms (e.g., MADDPG, MAPPO), MAPPO-Lag is also a centralized-training-based algorithm. This means it requires centralized or all-to-all communication to gather information (e.g., observations, actions, rewards, costs, and policies) from all agents for policy learning when addressing safe MARL tasks beyond simulation, which demands high communication resources. To overcome the difficulty, decentralized MARL have been investigated recently [R2], [R3]. Decentralized MARL algorithms aim to solve the problem considered by centralized-training-based algorithms over sparse communication networks to meet practical communication conditions. Our algorithm is decentralized, which eliminates the need for centralized or all-to-all communication required by MAPPO-Lag. Please note that it is not straightforward to design decentralized algorithms by simply combining parameter updates in centralized-training-based algorithms with parameter aggregation, as the information obtained by each agent is local. The difference between centralized and decentralized MARL algorithms, along with our contributions to decentralized safe MARL, is discussed in the Introduction and Related Work sections.
>
> **[Response to Weakness 4]** Thanks for your comment. In fact, the convergence of our theoretical algorithm is obtained under standard assumptions which are commonly used in existing decentralized MARL algorithms [R2], [R3], [R5]. This point of view is also supported by Reviewer b1tN (see the Summary for details). Compared to existing decentralized MARL works without safety [R2], [R3], [R5], the theoretical analysis of our algorithm is more challenging due to the decentralized update of the dual variables. Note that the authors are also clear that these standard assumptions may limit applications of our method when dealing with practical safe MARL tasks. Similar to SAC, practical loss functions for the update of the critic, the actor and the dual variable are designed based on our theoretical algorithm, which allows us to employ deep neural networks during training and reuse samples based on the experience replay mechanism. Moreover, the learning rates of the critic, actor and the dual variable can be tuned depending on specific tasks. Additionally, we are unclear about the reviewer's comment regarding the learning rate. For deep RL algorithms without safety, we always set a relatively larger learning rate for the critic for finding correct parameter update directions for the actor. In primal-dual-based safe RL algorithms (e.g., MAPPO-Lag), the learning rate for the dual variable is typically the smallest to ensure learning stability. Therefore, the learning rate setting in our experiment aligns with our theoretical assumptions to some extent.

---

> > ### Author Response · Authors · 2024-11-22
> > **Response Part II**
> >
> > **[Response to Weakness 5]** Thanks for your comment. We believe this comment on our work is unjustified. Apart from SAC which is technically modified from soft policy iteration, many state-of-the-art RL methods are technical modifications from theoretical algorithms which is not practical. For example, PPO is a technical implementation of TRPO. Note that both DDPG and TD3 are modified from a theoretical actor-critic algorithm using deterministic policy gradient. Note that the SOTA safe MARL algorithm MAPPO-Lag which you mention in Weakness 2 is a technical implementation of a safe multi-agent policy iteration algorithm. These practical algorithms typically lack theoretical convergence guarantees due to the use of complex neural networks and task-specific hyperparameters, such that their performance is always empirically validated in RL tasks.
> >
> > **[Response to Weakness 6]** Thanks for your comment. As mentioned in the response to Weakness 4, our theoretical contribution to decentralized safe MARL is not insignificant. In addition, we have taken a big step towards the application of decentralized safe MARL methods in continuous safe multi-agent coordination tasks compared with existing decentralized safe MARL works. In the revised manuscript, our proposed algorithm is further compared with a reward-shaping-based baseline (as suggested by you), and is also evaluated on a customized 3D environment (as suggested by Reviewer b1tN). Please refer to Appendix J.6 for more details.
> >
> > **[Response to Weakness 7]** Thanks for your very constructive suggestion. We have compared our algorithm with the suggested baselines. Detailed discussions are provided in Appendix J.6.
> >
> > **[Response to Minor]** Thanks for your careful reading. In the revised manuscript, $C$ and $c_t$ in $C_t = [c_t(i,j)]_{N \times N}$ have been replaced by $W$ and $w_t$, respectively.
> >
> > **[Response to Question 1]** Thanks for your question. Individual observation plays an important role in our algorithm design. In this work, we study safe MARL problems with a team of homogeneous agents. Similar to existing decentralized MARL works [R2], [R3], [R5], the global observability of the state and action is assumed here. The individual observation function enables each agent $i$ to have different observation $o_i$ provided the same state $s$, which allows the agents to generate different actions when they are equipped with the same observation-based policy. Based on the proposed homogeneous constrained Markov game model (Definition 1), it is proven that sharing of the observation-based policy doesn't harm the optimality of our safe MARL problem, which inspires us to incorporate the policy parameter consensus step into our algorithms. Based on the consensus of the policy parameters, each agent is able to estimate the policy of the other agents using its local policy. In our theoretical algorithm design, this setting enables the agents to perform the decentralized dual variable update. In our practical algorithm design, this setting enables us to incorporate the experience replay mechanism, as the local actions of the other agents can be predicted by the agent using its own policy. The observation of each agent contains its position and velocity vectors, the relative position vector of both the other agents and landmarks, and some additional task-specific information, which can be found in the code that we have shared.

---

> > > ### Author Response · Authors · 2024-11-22
> > > **Response Part III**
> > >
> > > **[Response to Question 2]** Similar to existing decentralized MARL works [R2], [R3], [R5], $C_t$ (now $W_t$) is handcrafted (see (52) in Appendix H for details). As explained in the response to Question 1, to achieve parameter consensus during training, the design of $C_t$ should satisfy Assumption 2, where detailed explanation on this assumption can be found in Appendix C.
> > >
> > > **[Response to Question 3]** As stated in the responses before, our practical algorithm is developed based on practical modifications on our theoretical algorithm. Note that it will take expensive sampling costs when training policies for multi-agent systems in continuous spaces. To improve sample efficiency, we employ the off-policy training architecture here, which is similar to SAC and DDPG. Note that existing decentralized safe MARL algorithms are on-policy. Therefore, the development of this off-policy decentralized safe MARL algorithm represents a technical contribution to the field of decentralized safe MARL.
> > >
> > > **[Response to Question 4]** The convergence of our theoretical algorithm has already been evaluated on a toy experiment. Please see Appendix H for details.
> > >
> > > **[Response to Question 5]** Our practical algorithm is not dependent on the discrete space assumption. An explanation of this perspective can be found in the responses above.
> > >
> > > **[Response to Question 6]** The determination of the threshold $b$ has already been discussed in Appendix J.2. Specifically, let $L$ be the episode length, and $\bar b$ be the real-world cost threshold, which can either be an upper bound on the number of constraint violations or have a specific physical meaning for certain tasks. Then, the threshold $b$ in our problem (3) is approximated by $b = \frac{(1 - \gamma^L)\bar b}{(1- \gamma)L}$. In the Aggregation task and the Swapping task in our experiment, we consider indicator cost functions satisfying $c_{i,t+1} = 1$ when agent $i$ is unsafe, and $c_{i,t+1} = 0$ otherwise. We set $\bar b = 0$ to ensure safety of all agents throughout the entire trajectory, such that $b = 0$. In the Formation task, a team cost is shared by all agents, which is the distance between the agents' mean position and a target position. Note that this distance cannot be zero at the beginning of the episode. We set $\bar b = 2$ in our experiment based on our environment setting, such that $b = \frac{(1 - \gamma^L)2}{(1- \gamma)L}$.
> > >
> > > **References**
> > >
> > > [R1] Shalabh Bhatnagar, Richard S. Sutton, Mohammad Ghavamzadeh, and Mark Lee. Natural actor-critic algorithms. Automatica, 45(11):2471–2482, 2009.
> > >
> > > [R2] Zhang K, Yang Z, Liu H, et al. Fully decentralized multi-agent reinforcement learning with networked agents[C]//International conference on machine learning. PMLR, 2018: 5872-5881.
> > >
> > > [R3] Hu Y, Fu J, Wen G, et al. Distributed entropy-regularized multi-agent reinforcement learning with policy consensus[J]. Automatica, 2024, 164: 111652.
> > >
> > > [R4] Haarnoja T, Zhou A, Abbeel P, et al. Soft actor-critic: Off-policy maximum entropy deep reinforcement learning with a stochastic actor[C]//International conference on machine learning. PMLR, 2018: 1861-1870.
> > >
> > > [R5] Chen D, Li Y, Zhang Q. Communication-Efficient Actor-Critic Methods for Homogeneous Markov Games[C]//International Conference on Learning Representations.

---

> > > > ### Comment · Reviewer_93J6 · 2024-11-25
> > > >
> > > > Thanks very much for the detailed reply and the additional experiments. Some of my concerns have been addressed, so I have raised my score to 5. I did not increase to a higher score for the following reasons.
> > > >
> > > > 1. From the current presentation and the authors' rebuttal, the novelty of the work still seems limited and the proposed algorithm seems incremental. This is agreed upon by all reviewers. If the authors believe that the contribution of the proposed algorithm is not straightforward, I would encourage the authors to state clearly what key theoretical problem is addressed mathematically, instead of arguing that "our theoretical contribution is not insignificant".
> > > > 2. The limited empirical evaluations are supported by other reviewers as well. What are the obstacles that impede the authors from testing their algorithm with more complex benchmarks?
> > > > 3. In [Response to Weakness 5], the authors replied that my comment was unjustified. However, it seems that the authors have justified this weakness in their reply. My comment was "The proposed practical algorithm violates the assumptions of the convergence analysis.", and the authors replied that other works also contain theoretical algorithms that are not practical.
> > > > 4. The gap between the theoretical analysis and the practical algorithm seems to be large, which is also raised by other reviewers.

---

> > > > > ### Author Response · Authors · 2024-11-26
> > > > > **Response**
> > > > >
> > > > > Dear Reviewer,
> > > > >
> > > > > Thanks so much for raising your score on this work. We sincerely appreciate your additional valuable comments on this work. We have provided a statement on the theoretical contribution of this work (below our global response). Detailed responses to your additional comments are given as follows:
> > > > >
> > > > > **[Response to Comment 1]** We believe the reviewer will agree that since safe RL and RL are closely related, many safe RL methods seem incremental, which use existing RL methods for policy optimization. Many reviewers including you think the contribution of this work is limited, as we use the classic primal-dual method to learn safe policies. Nevertheless, as shown in the statement, it has remained a huge challenge to combine the primal-dual method with decentralized MARL before our task. In the decentralized setting, the joint policy is unavailable to any individual agent, while the constraint is associated to the joint policy. Compared with unconstrained decentralized MARL, an important mathematical problem in decentralized safe MARL is how to update dual variables associated to centralized constraints in a decentralized fashion. None of the existing decentralized safe MARL methods can deal with this problem under the "true" decentralized setting (please refer to our statement). Our work tackles this problem mathematically based on homogeneous multi-agent systems, and it remains a challenge to consider this problem for heterogeneous multi-agent systems. In the statement, we also explain that the analytical tools used in existing decentralized MARL methods cannot be employed in this work, as a projection operator is included in the local update step of the dual variable. We are sorry for using the sentence "our theoretical contribution is not insignificant" in our original response. We provide detailed reasons this time.
> > > > >
> > > > > **[Response to Comment 2]** Thanks for your valuable question. Based on the comments from you and another reviewer, we have provided more experiments to evaluate our algorithm in the revised paper, including introducing another baselines and a 3D environment. When dealing with more challenging safe MARL tasks, more technical tricks should be included. Taking the multi-robot coordination tasks considering collision avoidance as an example, advanced centralized-training MARL methods may use recurrent neural networks and graph neural networks to build the critic and actor, which may also use some practical tricks such as the policy distillation to obtain stable learning performance. In their ablation experiments, the learning performance of the proposed method without these tricks is terrible. We are unable to provide theoretical justifications on these technical tricks which make the RL method work at present. Even though our algorithm can be combined with these tricks, please allow us to evaluate it on more challenging tasks with the combination of these tricks as our future work. Again, compared with existing decentralized safe MARL works, we have taken a step towards the application of decentralized safe MARL algorithms through the combination with deep RL.
> > > > >
> > > > > **[Response to Comments 3 and 4]** Based on your comments, we have collected all (including newly added) comments on our practical algorithm: Weakness 3 by Reviewer rR6P, Weaknesses 5, 6, Questions 3, 4 by Reviewer 93J6 (you), Weakness 1(c), Question 6 by Reviewer b1tN, and Weakness 4 by Reviewer Futv. None of the rest of reviewers raised concerns on the relationship between our practical algorithm and the theoretical algorithm. On the contrary, we are praised by many reviewers on performing the empirical results based on our practical algorithm, which is supported by Summary and Strength 3 by Reviewer Futv, Summary and Strength 2 by Reviewer Hjad, Summary and Strength 4 by Reviewer b1tN. We have provided many examples in our original response to explain that practical algorithms does not obey the assumptions for the theoretical algorithm due to the incorporation of technical tricks. This is not a weakness of our algorithm, right? One target of this work is to further develop a practical version of our theoretical algorithm to improve learning performance of decentralized safe MARL algorithms in continuous safe MARL tasks. We never stated that our practical algorithm is theoretically convergent, which is similar to existing practical algorithms.
> > > > >
> > > > > Thank you again for the time and efforts spent on reviewing our paper.
> > > > >
> > > > > Yours faithfully,
> > > > >
> > > > > Authors

---

### Official Review · Reviewer_b1tN · 2024-11-04

**Soundness:** 3
**Presentation:** 2
**Contribution:** 2
**Rating:** 3
**Confidence:** 4

**Summary:**

This paper presents a decentralized primal-dual actor-critic algorithm for safe multi-agent reinforcement learning (MARL). Initially, the authors formulate the studied problem as a homogeneous constrained Markov game with the addition of entropy regularization to enhance exploration. Then, an on-policy algorithm which can be trained in a decentralized manner, is presented based on policy gradient, using local gradient and consensus steps for updating the policies. Convergence guarantees for this algorithm are also provided under standard assumptions. Subsequently, an off-policy version of this method is provided using standard deep RL techniques. Finally, a comparison with other methods/variations on three multi-robot tasks is demonstrated which shows the advantages of the proposed algorithm.

**Strengths:**

The strengths of this paper can be summarized as:

1. The safe MARL problem is formally formulated as a homogeneous constrained Markov game. Entropy regularization is also incorporated for facilitating exploration.

2. An on-policy decentralized primal-dual actor-critic method is presented for addressing this constrained Markov game. The included constraint is handled through the introduction of a dual variable, with the new goal now being to find a saddle point of the Lagrangian. Training takes place in a decentralized manner under the assumption that the global state and joint action can be observed by all agents.

3. Convergence guarantees are provided for the on-policy method based on commonly used assumptions in the safe MARL literature.

4. A practical off-policy variant of the method is presented for enhancing its performance and increasing sampling efficiency.

5. The performance of the method is verified on three multi-robot tasks. The proposed DPDAC-ER method achieves similar performance to MASAC-Lag. It is also shown to respect the constraints in most cases in contrast with MASAC and DAC-ER that do not account for constraints. Finally, the importance of entropy regularization is highlighted since the performance of DPDAC is significantly inferior compared to DPDAC-ER.

**Weaknesses:**

The weaknesses/limitations of this work are outlined as follows:

1. The proposed approach appears to be a bit incremental compared to prior works presented in [R1, R2]. In [R1] a decentralized actor-critic algorithm was presented for homogeneous MGs without the additions of constraints or entropy regularization that the current submission is addressing. Subsequently, [R2] presented a decentralized actor-critic algorithm that also incorporated entropy regularization for encouraging exploration. Hence, to the reviewer’s best understanding, the main difference between [R2] and the current submission is the inclusion of the constraint and the related dual variable. In particular,

   a) The on-policy algorithm presented in Section 4 of [R2] is essentially the same with the one in Section 3 of the current paper, without the addition of the dual update related to the constraint. In fact, the updates in Eq. (9) and (10) in [R2] are the same with the ones in Eq. (6) and (7) of the current paper.

   b) The convergence analysis provided in Section 4 is very similar to the one in Section 5 in [R2]. The main difference lies in the inclusion of the dual variable, which is handled through the multi-timescale stochastic approximation theory, assuming the dual variable to be fixed when studying the convergence of the actor and the critic.

   c) While the off-policy variant presented in Section 5 appears to be a practical alternative, this is not really a novel contribution as the suggested modifications (actor/critic neural networks and replay buffer) are common tricks in the deep RL literature.

     Furthermore, primal-dual algorithms for handling constraints in safe MARL have already been quite popular in the literature; see for example [R3, R4, R5].

2. The authors claim to be using only observation-based policies. Nevertheless, the assumption in item (iii) of Definition 1 that each $o_i$ is a bijective function, implies that every observation corresponds to a unique state. It is unclear whether this is an indirect assumption of full observability.

3. In the experiments, the three presented problems (Aggregation, Swapping and Formation) are quite simple multi-robot tasks on a 2D space. Exploring more complex tasks with more constraints or tasks in 3D space could further justify the importance of the proposed approach and how it can be advantageous compared to the other methods.

4. In one of the presented tasks (Formation), it is unclear whether the algorithm can provide solutions satisfying the constraint, although this seems a quite simple constraint to meet. See question 7 for more details.



[R1] Chen, D., Li, Y., & Zhang, Q. (2022). Communication-Efficient Actor-Critic Methods for Homogeneous Markov Games. In The Tenth International Conference on Learning Representations (ICLR 2022).

[R2] Hu, Y., Fu, J., Wen, G., Lv, Y., & Ren, W. (2024). Distributed entropy-regularized multi-agent reinforcement learning with policy consensus. Automatica, 164, 111652.

[R3] Paternain, S., Calvo-Fullana, M., Chamon, L. F., & Ribeiro, A. (2022). Safe policies for reinforcement learning via primal-dual methods. IEEE Transactions on Automatic Control, 68(3), 1321-1336.

[R4] Ying, D., Zhang, Y., Ding, Y., Koppel, A., & Lavaei, J. (2024). Scalable primal-dual actor-critic method for safe multi-agent rl with general utilities. Advances in Neural Information Processing Systems, 36.

[R5] Lu, S., Zhang, K., Chen, T., Başar, T., & Horesh, L. (2021, May). Decentralized policy gradient descent ascent for safe multi-agent reinforcement learning. In Proceedings of the AAAI conference on artificial intelligence (Vol. 35, No. 10, pp. 8767-8775).

**Questions:**

1. The authors are strongly encouraged to elaborate on the differences between their submission and [R2], as well as on the novelty of the contribution of this paper in general; see weakness (1).

2. Please provide a clarification on the point raised in weakness (2), regarding whether full observability is indirectly assumed in this problem setup.

3. The convergence analysis relies on the multi-scale approximation theory, essentially assuming that the critic and actor converge under fixed dual variables. The authors are encouraged to elaborate on why this is a reasonable assumption.

4. In the convergence analysis, in line 823, it is mentioned that ‘Note that both the state and action spaces are discrete.’ Does this suggest that the convergence analysis is not applicable to the continuous action spaces? The authors are encouraged to provide a clarification given that the paper is supposed to be addressing continuous action spaces.

5. The variable $\rho$ is defined as the ‘initial state distribution’ in line 156. But $\rho$ also appears in line 743 (as part of Eq. (25)), which in that context may be related to the spectral norm?

6. Can the proposed algorithm be successfully applied to more complex tasks? The authors could consider for example, more “obstacles” that all agents need to avoid, or tasks in 3D space. It would also be interesting to comment on the advantages of entropy regularization in such tasks, since it could be expected that proper exploration would be even more critical in such scenarios.

7. In Figures 1, 4 and 5, the average episode costs for the formation task never reach the threshold values. In fact, in Fig. 5, with threshold values 0.0, 2.0, 5.0, the cost of the resulting policies is around 2.0, 4.0 and 6.0, respectively. Does this indicate that the proposed algorithm cannot really satisfy the constraint for this task? It is also interesting how increasing the threshold gives “more unsafe” policies that violate even the additionally relaxed threshold.

---

> ### Author Response · Authors · 2024-11-22
> **Response Part I**
>
> Thank you very much for the time and efforts spent on reviewing our paper. Thank you for the great summary on this work. We have tried to address all your concerns. Please refer to the following responses for details.
>
> **[Response to Weakness 1(a)]** Thanks for your careful reading and constructive comments. Our work is developed based on existing works on decentralized MARL [R1], [R2], where the design of decentralized parameter update steps for both the critic and actor is inspired by those in [R2]. This enables each agent to obtain a decentralized estimate of the global value function associated to the optimized joint policy provided a fixed dual variable. Nevertheless, designing a decentralized update step for the dual variable associated with centralized constraints and establishing convergence for the entire decentralized algorithm have remained significant challenges prior to our work. Under the fully decentralized setting, the joint policy of all agents is not available to any individual agent, such that the gradient of the Lagrangian $L(\theta, \lambda)$ (formula (3) in the manuscript) w.r.t. $\lambda$ cannot be calculated by any individual agent. It is worth pointing out that it still remains an open problem to design decentralized algorithms considering centralized constraints for heterogeneous multi-agent systems. In this work, the decentralized update step for the dual variable is designed based on the policy consensus among agents, where each agent estimates the other agents' policies with its local policy. Further explanation of our centralized constraint can be found in our response to Weakness 2 for Reviewer bAPC.
>
> **[Response to Weakness 1(b)]** Thanks for your comments. In this work, the convergence of the dual variables cannot be directly established using the analysis tools from [R1] and [R2], due to the incorporation of the projection operator in the local updates of the dual variables. Through the analysis on the projection terms, the convergence of the dual variables is established (Theorem 5). In addition, through the proof of Theorem 5, the reviewers can also observe that designing convergent decentralized algorithms for heterogeneous agents (i.e., agents with different policies) is very challenging.
>
>
> **[Response to Weakness 1(c)]** Thanks for you comment. The development of our practical algorithm is inspired by the famous deep RL algorithm SAC [R3]. In [R3], a theoretical algorithm named soft policy iteration was proposed at first, which only considers finite state and action spaces. SAC was then proposed based on the theoretical algorithm for challenging continuous RL tasks, where the loss function for the critic was designed based on the TD error, and the loss function for the actor was designed based on the policy improvement step in soft policy iteration. Even though the modifications (actor/critic neural networks and replay buffer) in SAC were also developed before, we believe the key contribution here is turning the theoretical algorithm into practical loss functions. Note that our practical algorithm is developed similarly, which turns the parameter update formulas in the theoretical algorithm into practical loss functions. It is also worth pointing out that since each agent has not acess to the joint policy in the decentralized setting, existing works are limited to the on-policy training setting. In this work, the policy consensus of this work is leveraged by each agent to esimate the other agents' policy with its local policy. Finally, we show the difference between our work and [R4], [R5], [R6]. Both [R4] and [R5] proposed convergent decentralized safe MARL algorithms. Nevertheless, these algorithms are on-policy, and rely on the vanilla policy gradient for policy improvement. Note that they were only validated in environments with discrete spaces, with some even utilizing Q-tables during training [R4]. Our practical algorithm can deal with continuous tasks, and is sample-efficient by reusing samples, which takes a step towards the applications of decentralized safe MARL methods in practical safe multi-agent tasks. [R6] investigated the sub-optimality gap in primal-dual safe RL methods. However, no theoretically convergent algorithm was proposed in [R6].

---

> > ### Author Response · Authors · 2024-11-22
> > **Response Part II**
> >
> > **[Response to the Weakness 2]** It has already been stated in the related section and Section 2.2 that the global state and action information is available to each agent. Note that this assumption is common in existing works on decentralized MARL [R1], [R2], [R5]. In Definition 1, it is assumed that each agent $i$ has a unique observation function $o_i$ which can map a common state into different observations. As a result, the agents can behave differently when they employ the same observation-based policy. It is further shown in Theorem 1 that the optimality of our safe MARL problem can be preserved when all agents share the same observation-based policy. Inspired by this result, the policy consensus step is incorporated into our decentralized algorithm design. In our experiment, we also consider the local observation setting, where the global state and action information is not available to agents. Please refer to Appendix J.5 for more details.
> >
> > **[Response to the Weakness 3]** Thanks for your very helpful comments. In the revised manuscript, our proposed algorithm is further evaluated on a customized 3D environment (as suggested by you), and is also compared with a reward-shaping-based baseline (as suggested by Reviewer 93J6). Please refer to Appendix J.6 for more details.
> >
> > **[Response to the Weakness 4]** Thanks for your careful reading. In the Formation task, the cost is designed as the distance between the agents’ mean position and the landmark’s position, which is shared by all agents. Note that the episodic cost cannot be zero, even when the safest policy is applied, as the distance is positive at the beginning of each episode. Nevertheless, it can be found in the attached video that the mean position of the agents approaches the target position as quickly as possible based on the policy trained by our algorithm.
> >
> > **[Response to the Question 1]** Thanks for your comments. Compared with [R2], we consider the decentralized MARL problem under centralized constraints. Before designing our algorithm, a constrained Markov game model is formulated which extends the model proposed in [R1] under the safe MARL setting. Optimality of the decentralized safe MARL problem based on the proposed model is theoretically analyzed, where policy sharing among agents can preserve optimality and safety. Based on this observation, a theoretical decentralized primal-dual algorithm is proposed. Compared with [R2], the local policy update is modified based on the policy gradient for the Lagrangian function, and a novel decentralized dual variable update step is designed. Convergence of the proposed decentralized algorithm is established. Note that the analysis tools in [R2] cannot be directly employed in our analysis (see the response to Weakness 1(b)). Finally, a practical decentralized safe MARL algorithm is proposed whose loss functions are theoretically derived from the theoretical algorithm. This practical algorithm bridges our theoretical distributed MARL algorithm and DRL. Compared with existing decentralized safe MARL methods, we can tackle continuous safe MARL tasks. Please refer to the related work section for more details.
> >
> > **[Response to the Question 2]** The response to this comment can be found in the response to Weakness 2. In addition, the authors want to express that one advantage of employing observation-based policies is that our algorithm can be adapted to local observation settings conveniently. Please refer to Appendix J.5 for more details.
> >
> > **[Response to the Question 3]** Based on multi-timescale stochastic approximation analysis, the update of the dual variables in our algorithm can be viewed to track the ODE $\dot \lambda_i(t) \equiv 0$ due to the condition $\beta_{\lambda,t} = o(\beta_{\theta,t})$ in Assumption 5, which allows us to consider the dual variables as constants when analyzing the faster update of policy parameters. Please refer to [R6] (Lemma 1, page 66) and [R7] (page 18, arXiv version) for similar analyses.
> >
> > **[Response to the Question 4]** Thanks for your valuable suggestion. It is possible to obtain similar convergence results of our algorithm in continuous spaces [R8]. However, the theoretical analysis would require introducing numerous additional notations, such as operators over matrices of functions and inner products for matrices of functions, which could harm the readability of this work. It is worth noting that one reviewer has already mentioned the abundance of notations in the manuscript. To this end, the discrete space setting is used in our theoretical algorithm design. Note that our practical algorithm can deal with continuous tasks.

---

> > > ### Author Response · Authors · 2024-11-22
> > > **Response Part III**
> > >
> > > **[Response to the Question 5]** Thanks for your correction. In the revised manuscript, $\rho$ in (25) has been replaced by $\tilde \rho$.
> > >
> > > **[Response to the Question 6]** The response to this comment can be found in the response to Weakness 3.
> > >
> > > **[Response to the Question 7]** The response to the first part of the comments can be found in the response to Weakness 4. Evaluating safe RL algorithms under different safety thresholds is also called the sensitivity analysis in safe RL [R9]. In experiments, a larger threshold results in a smaller dual variable during training (see the loss function for the dual variable). Thus, in the policy update, the agents will pay more attention to reward optimization, which results in more unsafe policies.
> > >
> > > **References**
> > >
> > > [R1] Chen D, Li Y, Zhang Q. Communication-Efficient Actor-Critic Methods for Homogeneous Markov Games[C]//International Conference on Learning Representations.
> > >
> > > [R2] Hu Y, Fu J, Wen G, et al. Distributed entropy-regularized multi-agent reinforcement learning with policy consensus[J]. Automatica, 2024, 164: 111652.
> > >
> > > [R3] Haarnoja T, Zhou A, Abbeel P, et al. Soft actor-critic: Off-policy maximum entropy deep reinforcement learning with a stochastic actor[C]//International conference on machine learning. PMLR, 2018: 1861-1870.
> > >
> > > [R4] Ying D, Zhang Y, Ding Y, et al. Scalable primal-dual actor-critic method for safe multi-agent rl with general utilities[J]. Advances in Neural Information Processing Systems, 2024, 36.
> > >
> > > [R5] Lu S, Zhang K, Chen T, et al. Decentralized policy gradient descent ascent for safe multi-agent reinforcement learning[C]//Proceedings of the AAAI conference on artificial intelligence. 2021, 35(10): 8767-8775.
> > >
> > > [R6] Borkar V S, Borkar V S. Stochastic approximation: a dynamical systems viewpoint[M]. Cambridge: Cambridge University Press, 2008.
> > >
> > > [R7] Zhang K, Yang Z, Liu H, et al. Fully decentralized multi-agent reinforcement learning with networked agents[C]//International conference on machine learning. PMLR, 2018: 5872-5881.
> > >
> > > [R8] Zhang K, Yang Z, Basar T. Networked multi-agent reinforcement learning in continuous spaces[C]//2018 IEEE conference on decision and control (CDC). IEEE, 2018: 2771-2776.
> > >
> > > [R9] Gu S, Kuba J G, Chen Y, et al. Safe multi-agent reinforcement learning for multi-robot control[J]. Artificial Intelligence, 2023, 319: 103905.

---

> > > > ### Comment · Reviewer_b1tN · 2024-12-03
> > > >
> > > > Thank you very much for your detailed response. Here are my key points after going over the revised version of your paper and the responses:
> > > >
> > > > - Overall, my main concern remains regarding the incremental nature of this work compared to [R1] and [R2]. While I understand the importance of incorporating a decentralized dual update step, it is hard to argue that the additions/modifications of this work are sufficient to meet the novelty standards of this conference.
> > > > - I really appreciate the added experiment with the 3D formation task. While this experiment provides a good indication about the performance of the proposed algorithm, the authors should try to have a more extensive experimental evaluation on complex environments. I strongly encourage the authors to present a greater variety of such tasks in future submissions.
> > > > - I understand that based on Assumption 5, you are able to assume that the critic and actor converge under fixed dual variables, which significantly facilitates the convergence analysis. While I see that this type of analysis has been used in a prior paper, my question on why this is a reasonable assumption has not been addressed.
> > > > - I understand that a convergence analysis for continuous spaces might have been much harder, but still this is the problem this paper is supposed to tackle. As a result, it is hard to convince the reader regarding the value of a convergence analysis on a setup that is substantially different than the one the work is supposed to address.
> > > >
> > > > To summarize, while this paper faces an important challenge in safe MARL, it falls short to meet the required standards regarding novelty, breadth of experimental results and relevance of some theoretical statements. As a result, I believe that my initial score (5, weak reject) is a fair evalution, and I will keep my score unchanged.

---

### Official Review · Reviewer_Hjad · 2024-11-04

**Soundness:** 2
**Presentation:** 2
**Contribution:** 2
**Rating:** 5
**Confidence:** 3

**Summary:**

This paper studies safe multi-agent reinforcement learning problem. The authors present a decentralized primal-dual actor-critic algorithm and analyze its convergence properties for *homogeneous*, *entropy regularized* and constrained Markov games with finite state and action spaces under linear function approximation while practical version adopting neural network approximation has also been discussed.

**Strengths:**

- The paper studies an important problem.
- Detailed numerical examples are provided.
- Claims are supported with theoretical analysis.

**Weaknesses:**

- The results are presented in a convoluted way such that it is challenging to make connections between different parts. For example, homogeneous MGs are defined yet not used in the convergence analysis except the one related to conditioning policies on limited observation.
- There are no justifications for the assumptions 1-6 made. Justification is needed especially for the non-standard ones.
- There are no discussions about the key ideas of the convergence analysis. For example, how important is the role of the entropy regularization or homogeneity of MGs in the convergence analysis.
- There should be more qualitative discussions about the algorithm presented.

**Questions:**

I have the following concerns:
- A similar problem has already been addressed in [Lu et al. 2021; Ying et al. 2023b]. These existing works have been criticized for addressing discrete action spaces while this paper also presents results for discrete cases. Therefore, the last sentence in page 1 is misleading. Furthermore, these works have been criticized to keep track of global policy for privacy-related issues. However, it is not explicit whether the homogenous Markov game structure plays an important role not needing to keep track of global policy. For example, the algorithm uses the product of the same policy, with the copy operator, as if all other agents follow the same local policy independently. Can the authors provide more intuition/explanation on this regard? Indeed, it is also confusing that theorem statements do not include the homogeneous MG assumption except Theorem 1 discussing limited observations. Does the paper consider homogeneous MGs just for limited observations?
- The limited observation extension is confusing since agents can keep track of, e.g., $Q^z(s_t,a_t;\omega_{i,t}^z)$, whereas condition their policy on $o_i(s_t)$. The former implies that they can observe $s_t$. The first line at page 5 also says "Once a tuple $(s_t,...)$" is collected". Therefore, such a limitation for policies is not clear. Furthermore, why the policy at line 237 is not conditioned on $o_i(s_t)$.
- Can the authors discuss the key technical ideas used in the proofs so that we can identify the technical novelty of the convergence analysis?
- There is a brief discussion about the duality gap at page 4. I think this is a major limitation and should be discussed in detail. For example, the convergence results are based on multi-time scale stochastic approximation theory, which necessitates global asymptotic stability for fast dynamics for any fixed parameter of the slow dynamics. Can the authors elaborate on whether this holds? Indeed, is this the reason to incorporate entropy regularization? Furthermore, is Assumption 6 related to this concern? Can the authors provide justification for Assumption 6?
- Related to the concern above, why do we consider entropy regularization especially because it is stated as a *huge* challenge at line 163. Furthermore, the entropy of the joint policy is the sum of the entropies of the local policies since joint policy is the product of the local policies, i.e., they are independent. The non-linear coupling of local policies does not sound. Can the authors provide more explanation in this regard?
- The update rule for $\omega$ is confusing since the temporal difference term at line 223 has the entropy regularization scaled by $\gamma$. However, at line 152, there is no such scaling for the regularization term.

---

> ### Author Response · Authors · 2024-11-22
> **Response Part I**
>
> Thank you very much for your appreciation of our work. Many thanks for your time and efforts in reviewing this paper. Please refer to the following responses for details.
>
> **[Response to Weakness 1]** Thanks for your careful reading and constructive comments. In fact, all parts of the manuscript are closely interconnected. In Section 2, the homogeneous constrained MG model is introduced, where the different observation functions allow agents to behave differently under the same state $s_t$ when they are equipped with the same observation-based local policy. Note that sharing the observation-based policy among agents may harm the optimality of the safe MARL problem. This problem is tackled in Section 3. It is shown in Theorem 1 that (observation-based) policy sharing among agents preserve the optimality of our safe MARL problem in homogeneous constrained MGs. This inspires us to incorporate the (decentralized) policy consensus step in our theoretical algorithm design, which is also shown in Section 3. After the convergence analysis in Section 4, a practical decentralized algorithm is proposed in Section 5 to bridge our theoretical algorithm and DRL, whose effectiveness in dealing with continuous safe MARL tasks is shown in Section 6. In addition, similar to existing works on decentralized MARL [R1], [R2], [R3], the observation function is assumed to be bijective in this work. It is stated in the related work section and is also shown in Section 2.2 that the global state is available to each agent. We also discuss the local observation setting in the manuscript (see Appendix J.5 for details), where the observation-action communication mechanism can be incorporated, such that each agent can replace the state and the joint action information with locally obtained information in the critic design.
>
> **[Response to Weakness 2]** Thanks for your comments. The justifications for our assumptions have already been given in Appendix C. These assumptions are standard for the convergence analysis of the proposed theoretical algorithm, which can be found in existing theoretical works on single-agent actor-critic [R4], [R5], [R6] and multi-agent actor-critic [R1], [R2], [R7].
>
> **[Response to Weakness 3]** Thanks for your valuable comments. It is stated in the response to Weakness 1 that the optimality analysis for our homogeneous constrained MG model inspires us to incorporate the policy consensus update in our algorithm. Therefore, the policy parameter consensus should be proved in our theoretical analysis. Please refer to the consensus analysis in the proof of Theorem 4. Regarding the entropy regularization setting, the local update step for the critic is designed based on the Bellman function associated to the entropy-regularized value functions [R8] (see (2.7)-(2.8) in [R8]). Moreover, the local update step for the actor is designed based on both the vanilla and the entropy-regularized policy gradients. In the convergence analysis of the critics, the convergence of the critic parameters associated to the reward should be established independently, as the temporal-difference error under the entropy regularization setting is different (see the operators in (9) in the manuscript). In addition, in the convergence analysis of both the critic and policy parameters, the boundedness of the terms associated to the entropy regularizer should also be analyzed. More details can be found in the proof of Theorems 3 and 4.
>
> **[Response to Weakness 4]** Thanks for your constructive comment. In our practical algorithm, the memory complexity of each agent is $O(H(2P + Q + 3))$, where $H$ denotes the upper bound on the number of the stored experiences, $P$ and $Q$ denote the dimensions of the state and the joint action, respectively. Additionally, the reward, cost, and done signals are also stored. Note that the memory complexity of storing the parameters of both the critic NNs and the actor NN is not included in the memory complexity mentioned above, which is due to that the storage of the replay buffer typically consumes the most memory in practice. The memory complexity of our algorithm is comparable to that of existing practical decentralized MARL algorithms [R1], [R2], which don't take safety into consideration.

---

> > ### Author Response · Authors · 2024-11-22
> > **Response Part II**
> >
> > **[Response to Question 1]** Thanks for your careful reading. In Section 5, a practical decentralized algorithm is developed by bridging the theoretical algorithm and DRL, which allows us to deal with continuous safe MARL tasks under the decentralized training setting (also see the third point of our contributions). The motivation of designing the practical algorithm can be found in the response to Weakness 1(c) for Reviewer b1tN. Discussions on the reasons for establishing the convergence of our theoretical algorithm in discrete spaces can be found in our response to Question 4 for Reviewer b1tN. We then express the difference between our work and existing decentralized safe MARL works [R3], [R9]. We have already provided reasons in the related work section that the algorithms proposed in [R3], [R9] have troubles in dealing with continuous tasks. The vanilla policy gradient method is employed in [R3], which always struggles in finding high-quality policies in continuous spaces. This point is also supported through our experiments (see the poor performance of DPDAC which also employs the vanilla policy gradient method). The algorithm proposed in [R9] inevitably faces challenges in estimating local state-action occupancy measures in continuous spaces. On the contrary, our algorithm represents comparable performance to solid centralized-training-based baselines in continuous tasks. It has also been stated in the related work section that [R3] requires each agent to have a local copy of the global policy structure, which will result in severe scalability issues when the scale of agents increases. On the contrary, each agent in our algorithm uses the copy operator to estimate the joint policy with the local policy based on the policy consensus step and Theorem 1, which is scalable. Finally, the authors want to stress that the observation is not limited in this work, as the observation function is bijective. An extension of our practical algorithm under the local observation setting can be found in Appendix J.5.
> >
> > **[Response to Question 2]** Thanks for your comments. In our setting, each agent $i$ is able to observe the environment state $s_t$. The state is then mapped into an individual observation $o_i(s_t)$ based on the observation function $o_i$, which is used as an input for the local observation-based policy. Thus, it is not surprising that each agent is able to collect the state during training. In addition, the joint policy joint at line 237 (now 240) has already been defined at line 170 (now 172), which is conditioned on the observations. Thanks.
> >
> > **[Response to Question 3]** Thanks for your valuable questions. In the decentralized training setting, it becomes a huge challenge to update the dual variables for the agents. Recall the centralized constraint $E_{s \sim \rho}[V_{\theta}^c(s)] = E_{s \sim \rho, a \sim \pi_{\theta}} [Q_{\theta}^c(s,a)] \le b$ in our problem. Even though the global value function $Q_{\theta}^c$ can be estimated using existing decentralized MARL algorithms, the gradient of the Lagrangian $L(\theta, \lambda)$ w.r.t $\lambda$ cannot be calculated by any individual agent as it has no access to the joint policy due to the decentralized setting. In this work, one of the technical novelties is the design of the decentralized dual variable update step (formula (8)), where each agent maintains a local estimate of the dual variable, denoted by $\lambda_i$. Through the analysis on the projection terms in the local updates of the dual variables, theoretical convergence of the dual variables is then established (see Theorem 5 and its proof). Compared with existing decentralized primal-dual methods [R3], [R9], the constraint satisfaction of the learned policy is further analyzed (see Propositions 1 and 2). Another technical novelty is the derivation of the loss functions based on the proposed theoretical algorithm (see Section 5 and Appendix I for details). It is also worth pointing out that in the decentralized training, it becomes very challenging to design off-policy algorithms due to the lack of the global reward, cost and policy information. In our theoretical algorithm, each agent can observe the actions of the other agents to calculate the local TD error, which doesn't require specific policy information due that all actions are sampled in an on-policy fashion. When modifying our algorithm based on the deep RL training architecture, the next action batch cannot be used as the actions were sampled using past policies. Based on the policy consensus, we sample the other agents' actions using the local policy.

---

> > > ### Author Response · Authors · 2024-11-22
> > > **Response Part III**
> > >
> > > **[Response to Question 4]** Thanks for your very valuable questions. By regarding the multi-agent system as a large but singe-agent system, our safe MARL problem (1) can be regarded as a safe RL problem, such that we can directly obtain the results on the sub-optimality gap in our problem using existing theoretical results for primal-dual-based safe RL methods [R10]. As a result, the optimal policy for the safe MARL problem can be approximately obtained by finding the global optimum of the dual problem. Unfortunately, like existing actor-critic methods [R1]–[R7], [R9], our algorithm converges to stationary points associated with the local optimum of the dual problem. Thus, the duality gap is not discussed in detail in the manuscript. Instead, the convergence of our (both theoretical and empirical) algorithm is evaluated empirically. Based on the comments here, we guess the reviewer is pointing at some policy-gradient-based primal-dual methods for safe RL [R11], [R12], which enjoy global convergence guarantees with convergence rates for the optimality gap. We truly respect these works, as their theoretical contributions to safe RL are solid. However, there are some key differences between these works and ours. In our work, the exact policy gradients are not available, such that agents build critic networks to estimate the value functions. On the contrary, the policy gradient information is available to the RL agent in [R11], [R12]. Our work assumes the general parameterized policy class, while the soft-max parameterization is considered in [R12] and the policy class without parameterization is considered in [R11]. We believe the policy class considered in this work is more general. In addition, as previously mentioned, one technical novelty of our theoretical algorithm is its adaptability and compatibility with DRL, whereas few convincing simulation results were provided in [R11] and [R12]. It would be greatly appreciated if the reviewer could introduce some works which enjoy global convergence under the actor-critic and the general policy parameterization settings. Finally, Assumption 6 is not related to the entropy regularization setting. Please refer to Appendix C for the discussion on Assumption 6. (Please allow us to explain the entropy regularization setting in the response to your next question.)
> > >
> > > **[Response to Questions 5 and 6]** Thanks for your questions. The entropy regularization mechanism has been widely employed to empirically improve sample-efficiency of RL algorithms in high-dimensional spaces [R13] and theoretically obtain global optimality for policy gradient methods [R12]. Note that the entropy-regularized value function takes different forms across these works. Motivated by the weakness of existing decentralized safe MARL methods in dealing with the continuous safe multi-agent tasks, we propose entropy-regularized decentralized safe MARL methods to improve exploration efficiency by maximizing the policy's entropy. Similar to [R13], the entropy-regularized value function takes the form $V^r_{\pi}(s) = E\left[\sum_{t=0}^{\infty} \gamma^t (\bar r_{t+1} + \alpha \mathcal{H}(\pi(\cdot|s_t))) \vert s_0 = s \right]$, where $\mathcal{H}(\pi(\cdot|s)) = -\sum_{a \in \mathcal{A}} \pi(a|s) \log(\pi(a|s))$ is the entropy functional. Note that the entropy-regularized value function is different from that in [R12] (see formulas (7)-(8)), which is not the product of the local policies. Under the decentralized setting, the joint policy $\pi$ is not available to each agent, while the entropy functional in the objective function nonlinearly couples the local policy information of all agents, making the design of the decentralized algorithm more challenging. Finally, the design of the critic update step is inspired by the Bellman equation for the entropy-regularized value function. Note that the following Bellman equations hold: $Q_{\theta}^r(s, a) = \bar R(s, a) + \gamma E_{s'}[V_{\theta}^r(s')]$, $V_{\theta}^r(s) = E_{a \sim \pi_\theta}[Q_{\theta}^r(s, a) - \alpha \log(\pi_\theta(a|s))]$ [R8]. We can obtain that $Q_{\theta}^r(s, a) = \bar R(s, a) + \gamma E_{s', a' \sim \pi_\theta}[Q_{\theta}^r(s', a') - \alpha \log(\pi_\theta(a'|s'))]$. We believe your question can be answered by this equation.

---

> > > > ### Author Response · Authors · 2024-11-22
> > > > **Response Part IV**
> > > >
> > > > **References**
> > > >
> > > > [R1] Chen D, Li Y, Zhang Q. Communication-Efficient Actor-Critic Methods for Homogeneous Markov Games[C]//International Conference on Learning Representations.
> > > >
> > > > [R2] Hu Y, Fu J, Wen G, et al. Distributed entropy-regularized multi-agent reinforcement learning with policy consensus[J]. Automatica, 2024, 164: 111652.
> > > >
> > > > [R3] Lu S, Zhang K, Chen T, et al. Decentralized policy gradient descent ascent for safe multi-agent reinforcement learning[C]//Proceedings of the AAAI conference on artificial intelligence. 2021, 35(10): 8767-8775.
> > > >
> > > > [R4] Shalabh Bhatnagar, Richard S. Sutton, Mohammad Ghavamzadeh, and Mark Lee. Natural actor-critic algorithms. Automatica, 45(11):2471–2482, 2009.
> > > >
> > > > [R5] Bhatnagar S. An actor–critic algorithm with function approximation for discounted cost constrained Markov decision processes[J]. Systems \& Control Letters, 2010, 59(12): 760-766.
> > > >
> > > > [R6] Wesley Suttle, Zhuoran Yang, Kaiqing Zhang, and Ji Liu. Stochastic convergence results for regularized actor-critic methods. arXiv preprint arXiv:1907.06138, 2019.
> > > >
> > > > [R7] Zhang K, Yang Z, Liu H, et al. Fully decentralized multi-agent reinforcement learning with networked agents[C]//International conference on machine learning. PMLR, 2018: 5872-5881.
> > > >
> > > > [R8] Cayci S, He N, Srikant R. Convergence of Entropy-Regularized Natural Policy Gradient with Linear Function Approximation[J]. SIAM Journal on Optimization, 2024, 34(3): 2729-2755.
> > > >
> > > > [R9] Ying D, Zhang Y, Ding Y, et al. Scalable primal-dual actor-critic method for safe multi-agent rl with general utilities[J]. Advances in Neural Information Processing Systems, 2024, 36.
> > > >
> > > > [R10] Paternain S, Calvo-Fullana M, Chamon L F O, et al. Safe policies for reinforcement learning via primal-dual methods[J]. IEEE Transactions on Automatic Control, 2022, 68(3): 1321-1336.
> > > >
> > > > [R11] Ding D, Zhang K, Başar T, et al. Convergence and optimality of policy gradient primal-dual method for constrained Markov decision processes[C]//2022 American Control Conference (ACC). IEEE, 2022: 2851-2856.
> > > >
> > > > [R12] Ying D, Ding Y, Lavaei J. A dual approach to constrained markov decision processes with entropy regularization[C]//International Conference on Artificial Intelligence and Statistics. PMLR, 2022: 1887-1909.
> > > >
> > > > [R13] Haarnoja T, Zhou A, Abbeel P, et al. Soft actor-critic: Off-policy maximum entropy deep reinforcement learning with a stochastic actor[C]//International conference on machine learning. PMLR, 2018: 1861-1870.

---

> > > > > ### Comment · Reviewer_Hjad · 2024-11-25
> > > > >
> > > > > I thank the authors for the detailed responses. However, the responses are very convoluted by referring to other responses, different parts of the paper and appendices, and various references. I could not find the strong answers addressing my concerns so that I can champion this paper for acceptance against the other reviewers' evaluations and ratings. Therefore, I will keep my rating unchanged.

---

> > > > > > ### Author Response · Authors · 2024-11-26
> > > > > > **Response**
> > > > > >
> > > > > > Dear Reviewer,
> > > > > >
> > > > > > Thank you for your feedback. We feel sorry that our responses have not fully addressed your concerns. Based on the comments from other reviewers, we have provided a detailed statement of the theoretical contributions of this work, which might improve your understanding of this work. However, we are unable to provide further detailed responses to the unresolved issues you have raised based on your latest comments. Could you kindly let us know which specific parts of this work remain unclear to you?
> > > > > >
> > > > > > We sincerely appreciate the time and effort you have dedicated to reviewing this work, and we are eager to address your concerns.
> > > > > >
> > > > > > Yours faithfully,
> > > > > >
> > > > > > Authors

---

### Official Review · Reviewer_Futv · 2024-11-04

**Soundness:** 2
**Presentation:** 3
**Contribution:** 2
**Rating:** 3
**Confidence:** 4

**Summary:**

The paper proposed a primal-dual algorithm for decentralized multi-agent safe RL in homogeneous constrained MDPs. The policy is shared by all the agents. The authors proposed an on-policy primal-dual algorithm and proved the asymptotic convergence. A practical off-policy algorithm is also proposed. The empirical results verify the effectiveness of the algorithm.

**Strengths:**

1. The writing and presentation are good and easy to read.
2. The theoretical part is technically correct.
3. The empirical results are presented clearly and verifies the algorithm performance.

**Weaknesses:**

1. The novelty is limited. The convergence proof is not new and can be found in many existing works in the references. The proof on homogeneous multi-agent MDP is just a repeated version and does not bring any novel theoritical insights.
2. The problem setting is confusing. The authors claim communications in the problem formulation, but use local policy and global observations in the later derivations.
3. The assumption of stochastic-optimization-based convergence proof is not practical. It is difficult to make the parameter set of neural networks bounded and do projected gradient descent.
4. The practical algorithm part is also not novel, the off-policy is a commonly seen algorithm in practical RL.

Minor questions:
1. The notation is confusing, especially the permutation notations in Definition 1.

**Questions:**

1. The communication graph setting is not clear. How to get global observation with only local communications? Does the local policy and reward function include neighborhoods?
2. What case is homogeneous MDP applicable to the real world? Is it an important question since the optimal policy structure is so simple that all agents share the same optimal policy?
3. What theoretical questions did you solve when you try to prove the convergence, based on previous stochastic optimization-based convergence proof?

---

> ### Author Response · Authors · 2024-11-22
> **Response Part I**
>
> Thank you very much for the time and efforts spent on reviewing our paper. We have tried to address all your concerns. Please refer to the following responses for details.
>
> **[Response to Weakness 1]** Thanks for your comments. The policy sharing mechanism plays an important role in MARL, which improves scalability of MARL algorithms significantly. It has been studied in [R1] that the optimality of MARL problems can be preserved when agents share the same observation-based policy. However, these results cannot be applied to safe MARL directly as the satisfaction of constraints should be considered jointly. To this end, we extend the model proposed in [R1] under the safe MARL setting, and study the optimality and safety of safe MARL problems under the policy sharing setting (see Theorem 1). Theorem 1 justifies the use of the policy sharing mechanism in the safe MARL algorithm design for the first time, which is a very important contribution for safe MARL. We believe the reviewer will agree that any so-called 'novel' theoretical insight that fails to lead to a significant conclusion cannot truly be considered novel. In decentralized safe MARL, it has remained a huge challenge to deal with the update of the dual variable associated to centralized constraints before our work (please refer to the response to Question 1 for Reviewer bAPC for more details). In this work, a novel decentralized dual variable update step is designed, and the consensus and convergence of the dual variables which incorporate the projection operator in the local update are proven for the first time. Compared with existing decentralized safe MARL works [R2], [R3], the constraint satisfaction of the learned policies is further studied (see Propositions 1 and 2). These results constitute our second contribution. The third contribution of this work is the deign of our practical algorithm, whose novelty will be discussed in the response to Weakness 4.
>
> **[Response to Weakness 2]** Thanks for your comments. The problem setting of this work is clear. It has been stated in the related work section that our work belongs to one category of  decentralized works [R1], [R2], [R4], where each agent has access to the global state and action information, but the agent only knows its local reward and policy information. Here, the observation-based local policy setting is common, which has been widely used in CTDE-based MARL algorithms. Due to the availability of the global state, each agent can get the observation of the agents based on our homogeneous constrained MG model (see lines 242-243 in the revised manuscript). One advantage of using observation-based policies is that our algorithm can be naturally adapted under the local observation setting. Here, the global observation information becomes not available to agents. To obtain a good estimate of the global value functions, the agents can share its local observation-action pairs with its neighbors. Note that this trick has been used in another category of decentralized MARL works [R3]. More discussions on this regard can be found in the related work section and Appendix J.5 in the manuscript.
>
> **[Response to Weakness 3]** Thanks for your comment. The assumption on the boundedness of the policy parameters is standard, based on which we directly use the estimate of the policy gradient to locally update the policy parameters in our algorithms. Similar settings can be found in various theoretical RL works [R1], [R2], [R4], [R5], [R6]. From a practical perspective, the learning stability of the policy parameters can be improved by setting smaller learning rates. It is worth pointing out that the entropy regularization mechanism employed in this work can further improve the learning stability of our algorithm due to its excellent sample efficiency (see our simulation results), which makes this assumption practical.

---

> > ### Author Response · Authors · 2024-11-22
> > **Response Part II**
> >
> > **[Response to Weakness 4]** Thanks for your comment. On the contrary, the practical algorithm is one key contribution of this work compared with existing decentralized safe MARL works. The off-policy training setting has been widely employed in existing centralized-training-based MARL algorithms, such as MASAC, MATD3. Nevertheless, in the decentralized training, it becomes very challenging to design off-policy algorithms due to the lack of the global reward, cost and policy information. In our theoretical algorithm, each agent can observe the actions of the other agents to calculate the local TD error, which doesn't require specific policy information due that all actions are sampled in an on-policy fashion. When modifying our algorithm based on the deep RL training architecture, the next action batch cannot be used as the actions were sampled using past policies. Based on the policy consensus inspired by the first contribution, we sample the other agents' actions using the local policy. Moreover, a practical loss function is theoretically derived based on the theoretical update step. If you are an expert in deep RL, you would understand the difficulty of transforming a theoretical algorithm into a practical RL algorithm, especially under the decentralized training setting. Realize that existing decentralized algorithms are all on-policy, whose effectiveness are verified on simple tasks considering discrete spaces. On the contrary, our algorithm is evaluated on continuous tasks, and is compared with solid baselines. We believe our work takes a big step towards the application of decentralized safe MARL on challenging tasks. Thanks.
> >
> > **[Response to Minor questions]** The definition of the permutation notation has already been given in the last paragraph of Section 1. In the revised manuscript, an illustrative example of our homogeneous constrained MG model is proposed. Please refer to Appendix A.1 for more details. If the reviewer has any difficulties understanding the notations and settings used in this work, please let us know.
> >
> > **[Response to Question 1]** Thanks for your question. The communication graph setting considered in this work is both mild and general, making it applicable to practical environments with limited communication resources due to its sparsity, and robust to link failures. This type of communication graph setting is also widely considered in the control community [R7]. The reason for the availability of the global observation can be found in the response to Weakness 2. The local policy of each agent takes the individual observation as its input, where the neighbors' information can be incorporated in the observation (an example of the observation function can be found in Appendix A.1). Note that the individual observation is not the local state (which doesn't include neighbors' information), which makes our policy setting differ from that in scalable MARL works (another category of decentralized MARL works) [R3].
> >
> > **[Response to Question 2]** A lot of safety-aware multi-agent coordination scenarios containing homogeneous agents can be case as examples of our model, such as city monitoring, wireless data collection and coverage control. In these tasks, these robots are distributed across the environment to achieve a common goal collaboratively. These agents make decisions based on their observations. Due to that the agents are homogeneous and the observations of agents are different under the same state, the policy sharing mechanism have been empirically combined into MARL algorithms, which successfully solved many large-scale multi-agent coordination tasks [R8], [R9]. Recently, the authors in [R1] provided theoretical justifications on the policy sharing mechanism, where the optimality of the MARL problem for homogeneous multi-agent systems can be preserved when all agents share the same observation-based policy. In our work, one contribution is providing theoretical justifications on the policy sharing mechanism under the safe MARL setting. We theoretically show that the optimality of the safe MARL problem can also be preserved under the policy sharing setting (Theorem 1 in the manuscript). In Appendix A, a proof of Theorem 1 is given, an illustrative example of our proposed model is provided, and some practical tasks based on our model is further introduced. If the reviewer is still unfamiliar with MARL algorithms with observation-based policies, please let us know. We will try our best to help the reviewer to understand this important setting in MARL.

---

> > > ### Author Response · Authors · 2024-11-22
> > > **Response Part III**
> > >
> > > **[Response to Question 3]** Thanks for your question. As mentioned in the responses before, one key theoretical question we solved is the design of the decentralized dual variable update step in our theoretical algorithm. Unlike the convergence analysis for the actor and the critic, the projection term in the local update step for the dual variables should be carefully analyzed. Note that it still remains a huge challenge to design decentralized safe MARL algorithms considering centralized constraints for heterogeneous multi-agent systems, where the policy sharing mechanism cannot be applied. Another key theoretical question we addressed is the practical loss function design for our theoretical algorithm, which you overlooked. In Appendix I, we demonstrate how the local update step for the actor is transformed into a practical loss function from a theoretical perspective, which serves as the foundation for our practical algorithm. As a result, this work makes contributions to decentralized safe MARL from the both the theoretical and the practical perspectives.
> > >
> > > **References**
> > >
> > > [R1] Chen D, Li Y, Zhang Q. Communication-Efficient Actor-Critic Methods for Homogeneous Markov Games[C]//International Conference on Learning Representations.
> > >
> > > [R2] Lu S, Zhang K, Chen T, et al. Decentralized policy gradient descent ascent for safe multi-agent reinforcement learning[C]//Proceedings of the AAAI conference on artificial intelligence. 2021, 35(10): 8767-8775.
> > >
> > > [R3] Ying D, Zhang Y, Ding Y, et al. Scalable primal-dual actor-critic method for safe multi-agent rl with general utilities[J]. Advances in Neural Information Processing Systems, 2024, 36.
> > >
> > > [R4] Hu Y, Fu J, Wen G, et al. Distributed entropy-regularized multi-agent reinforcement learning with policy consensus[J]. Automatica, 2024, 164: 111652.
> > >
> > > [R5] Paternain S, Calvo-Fullana M, Chamon L F O, et al. Safe policies for reinforcement learning via primal-dual methods[J]. IEEE Transactions on Automatic Control, 2022, 68(3): 1321-1336.
> > >
> > > [R6] Yan Zhang and Michael M. Zavlanos. Distributed off-policy actor-critic reinforcement learning with policy consensus. In IEEE Conference on Decision and Control, pp. 4674–4679, 2019.
> > >
> > > [R7] Pascal Bianchi and Jeremie Jakubowicz. Convergence of a multi-agent projected stochastic gradient algorithm for non-convex optimization. IEEE Transactions on Automatic Control, 58(2):391–405,2013.
> > >
> > > [R8] Liu I J, Yeh R A, Schwing A G. PIC: permutation invariant critic for multi-agent deep reinforcement learning[C]//Conference on Robot Learning. PMLR, 2020: 590-602.
> > >
> > > [R9] Hu Y, Fu J, Wen G. Graph soft actor–critic reinforcement learning for large-scale distributed multirobot coordination[J]. IEEE transactions on neural networks and learning systems, 2023.

---

> > > > ### Comment · Reviewer_Futv · 2024-11-26
> > > >
> > > > Thanks for replying to my comments. Some of my comments have been addressed. Here are two major questions or weaknesses of the paper,
> > > >
> > > > 1. For the problem setting and the assumptions, my confusion is mainly on the communication graph setting. In practice, sharing parameters is usually much more complex than sharing the reward function. I also agree with other reviewers that, current papers have similar settings/assumptions does not necessarily mean the setting is reasonable.
> > > >
> > > > 2. For the theoretical insights, the authors did not give a strong answer. The update and projection of dual variables can be found in previous theoretical analyses, for example, [1]. The theoretical analysis is a straightforward extension of previous papers.
> > > >
> > > > Therefore, I will keep my score unchanged.
> > > >
> > > > [1] Chow, Yinlam, et al. "Risk-constrained reinforcement learning with percentile risk criteria." Journal of Machine Learning Research 18.167 (2018): 1-51.

---

> ### Author Response · Authors · 2024-11-26
> **Response**
>
> Dear Reviewer,
>
> Thanks for your feedback. In practical environments, the communication resources among agents are limited, i.e., there is no all-to-all or centralized communication among agents. Thus, existing centralized-training-based MARL algorithms cannot be applied. This limitation inspires the development of decentralized MARL, where agents only communication through a sparse network. This communication condition has already been used in early distributed control tasks. Again, due to limited communication, each agent in decentralized MARL has no access to the local policy, reward and cost information of other agents. Based on this setting, we provide answers to your major questions or weaknesses:
>
> 1. Even the agents can share local reward, it results in extremely high communication costs as all agents should communicate for the reward information at each time step. Additional, even though the agents can obtain the global reward at each time step, they are still unable to update the dual variable which is associated to the joint policy. As a result, our method of sharing local parameter information is practical in decentralized safe MARL. In our rebuttal, we have successfully solved the problem on the decentralized setting raised by Reviewer rR6P (comment 1), and we don't find any additional question on our problem setting in the latest comments.
>
> 2. As stated in the responses to your former comments, the dual variable associated to the centralized constraint cannot be updated due to the lack of the joint policy. We thank the reviewer for introducing [1], which provided convergence for the single-agent case. However, in the decentralized MARL, we believe one important issus in this work is how to obtain similar convergence results (which use perfect information) under the decentralized setting (which use imperfect information, e.g., local policy, reward, and cost information). The reviewer may also find that the first decentralized work (zhang et. al, ICML 2018) is developed on the results for sinlge-agent actor-critic, whose main contribution is achieving decentralized learning of the global value function and the policy. The main theoretical of this work is achieving decentralized dual variable update, where we obtain similar convergence results as [1] with imperfect information.
>
> As a result, we believe the reviewer may still have some misunderstandings about decentralized MARL. We feel sorry about this.
>
> We sincerely hope our response can solve your current concerns.
>
> Yours faithfully,
>
> Authors

---

### Author Response · Authors · 2024-11-22
**General Response**

We would like to express our sincere gratitude to the reviewers for reading our paper and providing valuable feedback. The manuscript has been revised based on the suggestions by the reviewers, and the revised parts are marked in **blue** in the revised manuscript. We briefly summarize the revised parts here:

**Contribution**: We have revised the description of this work's contributions.

**Model**: An illustrative example of our model proposed in Section 2 has been added in Appendix A.1, along with practical examples provided in Appendix A.2.

**Experiment**: A reward-shaping-based safe MARL baselines has been added (Appendix J.6). A customized 3D environment has been built to evaluate the performance of our algorithm (Appendix J.6).

Based on the comments by the reviewers, we explain the **contributions** and some **key settings** of this work. Please find our responses to other questions in the personalized rebuttals.

>**Contribution 1: A homogeneous constrained MG model and the theoretical justification for the policy sharing mechanism in safe MARL.**

The policy sharing mechanism has been widely employed in the MARL algorithm design to handle large-scale multi-agent policy learning problems with homogeneous agents. However, this mechanism lacks theoretical justification in safe MARL problems. This issue is tackled in this work for the first time. A homogeneous constrained MG model is proposed, based on which we theoretically show that both optimality and safety of the safe MARL problem can be preserved under the policy sharing setting. This result inspires us to incorporate the decentralized policy consensus step in our algorithm design.

>**Contribution 2: A theoretically convergent decentralized safe MARL algorithm.**

Due to the joint policy information contained in the centralized constraint, it has remained a huge challenge to design decentralized safe MARL algorithms, as the joint policy is not available to any individual agent. In this work, a novel decentralized dual variable update step is designed based on the policy parameter consensus. Compared with existing decentralized MARL works [R1], [R2],  consensus and convergence of the dual variables are established in this work through the additional analysis of the projection terms in the local update step.

>**Contribution 3: A practical off-policy decentralized safe MARL algorithm.**

Since the joint policy is not accessible to any individual agent, existing decentralized safe MARL algorithms [R3], [R4] rely on on-policy training, leading to low sample efficiency. In this work, a practical off-policy decentralized algorithm is proposed for the first time based on our theoretical algorithm and the consensus of the local policies, where the relationship between the practical algorithm and the theoretical algorithm is analyzed from the theoretical perspective.

>**Setting 1: Availability of the global state and action information.**

Following existing decentralized MARL works [R1], [R2], [R4], [R5], our work assumes each agent has access to the global state and action. However, our algorithm can be naturally adapted under the local observation setting by using the observation-action communication which is employed in another category of decentralized MARL works [R3]. Detailed discussions and experiment results can be found in Appendix J.5, and discussions on two categories of works can be found in the related work section.

>**Setting 2: Continuous state and action spaces.**

Our theoreical algorithm obtains convergence under discrete spaces. Similar convergence results can be obtained under continous spaces [R5]. Our practical algorithm can deal with continuous safe MARL tasks.

>**Setting 3: The bijective observation function.**

The observation function is bijective, such that we can consider the safe MARL problem within observation-based policy spaces (see the proof for Theorem 1). In addition, the unique observation allows each agent to behave differently under the same state, which inspires the use of policy sharing.

References:

[R1] Chen D, Li Y, Zhang Q. Communication-Efficient Actor-Critic Methods for Homogeneous Markov Games [C]. International Conference on Learning Representations.

[R2] Hu Y, Fu J, Wen G, et al. Distributed entropy-regularized multi-agent reinforcement learning with policy consensus [J]. Automatica, 2024, 164: 111652.

[R3] Ying D, Zhang Y, Ding Y, et al. Scalable primal-dual actor-critic method for safe multi-agent rl with general utilities [J]. Advances in Neural Information Processing Systems, 2024, 36.

[R4] Lu S, Zhang K, Chen T, et al. Decentralized policy gradient descent ascent for safe multi-agent reinforcement learning [C]. Proceedings of the AAAI conference on artificial intelligence. 2021, 35(10): 8767-8775.

[R5] Zhang K, Yang Z, Basar T. Networked multi-agent reinforcement learning in continuous spaces [C]. 2018 IEEE conference on decision and control (CDC). IEEE, 2018: 2771-2776.

---

> ### Author Response · Authors · 2024-11-26
> **Statement of Our Theoretical Contribution**
>
> Dear all reviewers,
>
> Thank you for raising concerns about the theoretical novelty of this work. Here, we provided a statement of our theoretical contributions through the comparison with existing decentralized MARL works:
>
> The authors in [R4] proposed the first decentralized safe MARL algorithm, where the constraints are **functions of the joint policy of all agents**. In their algorithm, each agent maintains a local copy of the parameters of the joint policy. When updating the dual variable of agent $i$, **all agents must sample trajectories based on the local copy of parameters maintained by agent $i$ to calculate gradients**, which in fact violates the decentralized training setting.
>
> The constraint in [R3] is a function of the **local state-action occupancy measure only**, which cannot be used to model many practical constraints which are closely related to the **joint policy**.
>
> As a result, **the safe MARL problem which considers constraints associated to the joint policy has not been tackled under the decentralized training setting** before our work.
>
> Our safe MARL problem is **general**, whose constraint is associated to the **joint policy**. This poses a huge challenge for agents which has no access to the joint policy to **update the dual variable based on the experiences sampled using its local policy**. The theoretical contributions of this work are the **design of the decentralized dual variable update step**, and the  **convergence results of the dual variables** base on multi-timescale stochastic analysis.
>
> The novelty of this work has also been questioned due to its similarity to some unconstrained decentralized MARL methods [R1], [R2]. To the best of our knowledge, RL and safe RL are closely related, and many representative safe RL and MARL methods are developed based on existing RL and MARL algorithms. For example, CPO (MACPO) is developed based on TRPO (HATRPO),  where only an additional condition for the constraint is developed based on the monotonic performance improvement theory.  Nevertheless, safe RL still plays an important role in RL as various constrainted should be strictly satisfied in real-world tasks. In addition, it has also been stated in the reponse that the convergence of the dual variable in our algorithm cannot be establised based on existing decentralized methods, as the projector is incorporated into the local update for the dual variable.
>
> We sincerely hope that the clarification of our theoretical contributions enhances your understanding of this work. We thank the reviewers once again for their professional and thoughtful evaluation of our paper.
>
> Yours faithfully,
>
> Authors

---

### Meta-Review · Area_Chair_w3jg · 2024-12-17

**Metareview:**

The contribution of this work is not enough to warrant acceptance at this time. The problem definition focuses on a very narrowly scoped sub-problem within multi-agent RL, and its algorithmic as well as analytical contributions are fairly limited. The reviewers are in agreement about this assessment.

**Additional Comments On Reviewer Discussion:**

See above.

---

### Decision · Program_Chairs · 2025-01-22

Reject